# Co-Regularization Enhances Knowledge Transfer in High Dimensions

**Shuo Shuo Liu**[1*]   **Haotian Lin**[1,2*]   **Matthew Reimherr**[1,2]   **Runze Li**[1]

[1]The Pennsylvania State University    [2]Amazon

## Abstract

Most existing transfer learning algorithms for high-dimensional models employ a two-step regularization framework, whose success heavily hinges on the assumption that the pre-trained model closely resembles the target. To relax this assumption, we propose a co-regularization process to directly exploit beneficial knowledge from the source domain for high-dimensional generalized linear models. The proposed method learns the target parameter by constraining the source parameters to be close to the target one, thereby preventing fine-tuning failures caused by significantly deviated pre-trained parameters. Our theoretical analysis demonstrates that the proposed method accommodates a broader range of sources than existing two-step frameworks, thus being more robust to less similar sources. Its effectiveness is validated through extensive empirical studies.

## 1   Introduction

Transfer learning (TL) is a technique that leverages knowledge from source domains to improve learning performance in a related but not necessarily identical target domain (Torrey and Shavlik, 2010). The success of TL usually relies on the philosophy that the pre-trained models over source domains carry informative knowledge and thus can benefit the target domain learning by fine-tuning the pre-trained models. This "model-based" two-step transfer framework is referred to as *hypothesis transfer learning* (HTL) in the community.

Recently, HTL has also drawn great attention and been studied in various high-dimensional models (Bastani, 2021; Li et al., 2022; Tian and Feng, 2023). These algorithms are usually established in the well-recognized pre-training and fine-tuning paradigm. Specifically, these algorithms pool the source datasets and obtain the pre-trained parameters via classical sparse learning techniques, such as Lasso regression (Tibshirani, 1996). The target parameter is then learned by minimizing regularized empirical risk minimization over the target data, with the regularization as the distance between the pre-trained parameter and the target parameter. The underlying assumption in these works to show provable learning gain is that the distance between pre-trained and target parameters is small. Later, Gu et al. (2024); Lin and Reimherr (2024b) revealed that the transfer learning gain dynamic of HTL further relies on the geometric angles and signal strength between source and target parameters, which thus raises the demand for obtaining a "good" pre-trained parameter as the lever.

Most of the aforementioned multi-source transfer learning methods indiscriminately fuse all source and/or target domains to obtain a pre-trained estimator for the target domain. However, this approach overlooks the fact that different source domains may vary in their relevance and contribution to the target task. Besides, since pre-trained parameters are optimized to perform well over the source

---

*Co-first author.

  This work does not relate to the authors' positions at Amazon.

  Correspondence to Shuo Shuo Liu (shuoliu.academic@gmail.com).

domain (Li et al., 2022), they are not guaranteed to be sufficiently similar to the target parameter, and thus, fine-tuning might fail to generalize well on the target domain. Therefore, the above observations raise the following question:

*How to better exploit the knowledge from sources that can enhance the learning of the target?*

In this work, we take a step towards addressing the aforementioned questions by proposing a multi-source integration transfer framework. We summarize our contributions as follows.

- For a given set of sources, we propose a co-learning process termed *Co-Regularization Transfer* (CoRT) for high-dimensional generalized linear models (GLMs). This process enables the target parameter to be a representation of the source parameter while minimizing the source empirical risk, thereby enhancing the transfer of knowledge that is beneficial to the target and preventing fine-tuning failures due to significantly deviated pre-trained parameters over sources. We develop convergence rates for the oracle CoRT to theoretically support its effectiveness and demonstrate that it accommodates a wider range of sources, achieving improved rates compared to existing two-step pre-training and fine-tuning frameworks.
- In the presence of outlier source domains and without prior knowledge about the sources, we propose a data-driven adaptive algorithm to identify these outlier sources, which prevents negative transfer in CoRT. Unlike existing source detection algorithms, whose performance heavily hinges on correctly selected hyperparameters, our approach eliminates such a need by leveraging a majority vote mechanism. We also theoretically show that this adaptive algorithm converges to the oracle.
- Beyond point estimation, we show that the proposed CoRT framework can also facilitate high-dimensional statistical inference by improving the coordinate-wise confidence interval (CI) for the target parameter. The CI construction is based on a desparsifying process of the CoRT estimator, and we theoretically show that the constructed CIs are narrower in length compared to those in single-task settings under certain conditions.

**Related Works.** Existing two-step HTL algorithms are based on the biased regularization (Schölkopf et al., 2001), which has been extensively applied and studied under supervised learning settings, e.g., Kuzborskij and Orabona (2013, 2017); Wang and Schneider (2015); Du et al. (2017); Lin and Reimherr (2024a,b).

Recently, this technique has been intensively adopted for various high-dimensional models in statistics with theoretical guarantees (Bastani, 2021; Li et al., 2022; Tian and Feng, 2023; Zhang and Zhu, 2022; Li et al., 2024; Liu, 2024; He et al., 2024). Specifically, Bastani (2021) studied the single-source TL in the high-dimensional linear regression and leveraged the biased regularization, which can be viewed as an extension from Kuzborskij and Orabona (2013) to the high-dimensional linear model. However, since Bastani (2021) considered an end-to-end training setting, the statistical convergence rate depends on both the source and target sample sizes, thereby demonstrating how a large sample size in the source domains can enhance transfer effectiveness. Li et al. (2023) extended the algorithm in Bastani (2021) to multiple sources by fusing all source datasets into the target dataset to build one pre-trained parameter. The convergence rates are shown to be faster than those using the target data under certain conditions. To exclude the impact from outlier sources, a data-driven algorithm using the Q-aggregation method (Rigollet and Tsybakov, 2011) is developed. Later, Tian and Feng (2023) extended the work of Li et al. (2022) to high-dimensional GLMs. Instead of aggregation, they proposed a sample splitting-based algorithm for outlier source detection. However, the threshold hyperparameter selection in the detection algorithm can significantly affect its performance and needs to be manually tuned. Li et al. (2024) studied the estimation and inference of the high-dimensional GLMs by constraining the empirical score functions. Both Tian and Feng (2023) and Li et al. (2024) provided coordinate-wise confidence intervals leveraged by desparsified Lasso and correction scores, respectively. It is worth mentioning that the aforementioned works on HTL in high-dimensional linear models primarily employ the two-step learning framework and the $\ell_1$-norm for regularization.

Broadly speaking, our work is related to multi-source domain adaptation (Sun et al., 2015). It has achieved extensive success in various applications, but the theoretical analysis remains somewhat lacking. Most theories are based on the assumption that the target distribution can be approximated by a mixture of source distributions, making the error bounds depend on divergence measures between domains (Blitzer et al., 2007; Ben-David et al., 2010; Mansour et al., 2008; Hoffman et al., 2018). These bounds are overly general and unable to provide fine-grained analysis about how larger source

sample size and minor domain discrepancies contribute to a faster learning rate. It is worth noting that multi-task learning is closely related to transfer learning, though their objectives differ. Multi-task learning aims to solve multiple learning tasks simultaneously by leveraging shared structure across tasks, like the Data Shared Lasso framework (Gross and Tibshirani, 2016; Ollier and Viallon, 2017) in high dimensions. In contrast, transfer learning focuses on improving performance on a specific target task by transferring knowledge from source data.

## 2 Transfer via Co-Regularization

**Problem Setup.** Consider $K + 1$ tasks, we observe the $k$-th task data $\mathcal{D}^{(k)} := \{(\mathbf{x}_i^{(k)}, y_i^{(k)})\}_{i=1}^{n_k}$ which are generated from the following canonical form of GLMs

$$\mathbb{P}\left(y_i^{(k)}|\mathbf{x}_i^{(k)}, \boldsymbol{\beta}^{(k)}\right) = \rho(y_i^{(k)}) \exp\left[y_i^{(k)}\mathbf{x}_i^{(k)\top}\boldsymbol{\beta}^{(k)} - \psi(\mathbf{x}_i^{(k)\top}\boldsymbol{\beta}^{(k)})\right]$$

for $k = 0, 1, \cdots, K$, where $k = 0$ denotes the target model and $k \neq 0$ denotes source models, $\boldsymbol{\beta}^{(k)}$ denotes the parameter, $\rho$ and $\psi$ are some known functions. For a given pair $(\mathbf{x}^{(k)}, y^{(k)})$ from the $k$-th model, the GLMs model the relationship between input and output via $\mathrm{E}(y^{(k)}|\mathbf{x}^{(k)}) = \psi'(\mathbf{x}^{(k)\top}\boldsymbol{\beta}^{(k)})$, where $\psi'$ is the first-order derivative of the cumulant function. The goal is to learn the target $\boldsymbol{\beta}^{(0)}$ using both the target and source data.

**Similar and Outlier Tasks.** In order to facilitate the algorithm design and develop corresponding statistical foundations, it necessitates some assumptions on the similarities between the source and target domains. We instantiate the statistical heterogeneity between the $k$-th source model and the target model as the $\ell_1$ norm between $\boldsymbol{\beta}^{(k)}$ and $\boldsymbol{\beta}^{(0)}$. Specifically, we define the $k$-th parameter contrast as $\boldsymbol{\delta}^{(k)} := \boldsymbol{\beta}^{(k)} - \boldsymbol{\beta}^{(0)}$ for $1 \leq k \leq K$, and the $k$-th source is considered as a "similar" source if $k \in \mathcal{S}_h = \{1 \leq k \leq K : \|\boldsymbol{\delta}^{(k)}\|_1 \leq h\}$ where $h$ is an *unknown* nonnegative parameter that controlling the similarity level. Intuitively, a larger $h$ indicates larger statistical heterogeneity and vice versa. For the sources in set $\mathcal{S}_h^c = \{1 \leq k \leq K : k \notin \mathcal{S}_h\}$, we can interpret them as "outlier" sources. Such notations have been widely adopted in the theoretical studies of transfer learning (Bastani, 2021; Li et al., 2022; Tian and Feng, 2023) or adversarial attacks/contaminations (Qiao, 2018; Konstantinov and Lampert, 2019; Konstantinov et al., 2020). In practice, such an oracle set $\mathcal{S}_h$ is typically unavailable, necessitating algorithms to identify those similar sources to better boost the learning of the parameters of interest. For simplicity, we denote $\mathcal{S}_h$ as $\mathcal{S}$ and $n_{\mathcal{S}} = \sum_{k \in \mathcal{S}} n_k$.

### 2.1 Co-Regularization Enhances Knowledge Transfer

To facilitate introducing the proposed Co-Regularization transfer scheme, we first focus on the case where there is no outlier source task, i.e., all sources are in $\mathcal{S}$, and anchor on this to develop an algorithm that can adapt to an unknown similarity level $h$.

Many existing transfer learning algorithms for high-dimensional models rely on a two-step paradigm to achieve knowledge transfer. Specifically, they leverage a data fusion technique in the first step by fusing all data from source models in $\mathcal{S}$ to obtain a pre-trained model and then fine-tune the model via the target data in the second step; see Appendix B for details. However, it has been empirically observed and theoretically revealed (Tian and Feng, 2023; Lin and Reimherr, 2024a,b; Gu et al., 2025) that if the fused source model and target model differ significantly in terms of signal strength and angle, fusing all source data will result a pre-trained model that suffers from significant bias, and thus it will be extremely difficult to have the subsequent fine-tuning steps to mitigate this bias, given the limited target data.

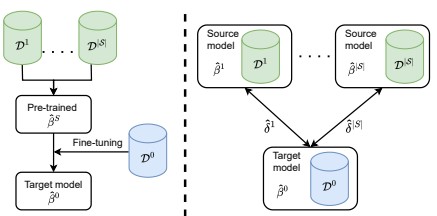

Figure 1: Left: Two-step pre-training and fine-tuning learning scheme (Li et al., 2022; Tian and Feng, 2023); Right: Co-Regularization transfer scheme (CoRT).

Consider the following scenario: while minimizing the empirical risk over the target domain, one can simultaneously train the source parameter $\boldsymbol{\beta}^{(k)}$ on the source domain by constraining the $\boldsymbol{\beta}^{(k)}$ to

be close to $\boldsymbol{\beta}^{(0)}$, which ensures only the beneficial knowledge is acquired by $\boldsymbol{\beta}^{(0)}$. Specifically, we can set $\boldsymbol{\beta}^{(k)}$ as $\boldsymbol{\beta}^{(0)} + \boldsymbol{\delta}^{(k)}$ during training to instantiate such a constraint. Based on this idea, we propose the *Co-Regularization Transfer* (CoRT), which is a co-learning process by minimizing the negative log-likelihood with Co-Regularization. Figure 1 provides an illustration of co-regularization and its comparison with classical two-step pre-training and fine-tuning transfer learning schemes.

**Definition 1** (Co-Regularization Transfer). *Define the negative log-likelihood over the $\mathcal{D}^{(k)}$ as*
$\ell(\boldsymbol{\beta}, \mathcal{D}^{(k)}) = \frac{1}{n_k} \sum_{i=1}^{n_k} \psi(\mathbf{x}_i^{(k)\top}\boldsymbol{\beta}) - y_i^{(k)}(\mathbf{x}_i^{(k)\top}\boldsymbol{\beta})$, *then solve the following objective*

$$\hat{\boldsymbol{\beta}}^{(0)}, \{\hat{\boldsymbol{\delta}}^{(k)}\}_{k\in\mathcal{S}} = \underset{\boldsymbol{\beta}^{(0)},\{\boldsymbol{\delta}^{(k)}\}_{k\in\mathcal{S}}}{\operatorname{argmin}} \mathcal{L}_\mathcal{D}\left(\boldsymbol{\beta}^{(0)}, \{\boldsymbol{\delta}^{(k)}\}_{k\in\mathcal{S}}\right), where \qquad (1)$$

$$\mathcal{L}_\mathcal{D}\left(\boldsymbol{\beta}^{(0)}, \{\boldsymbol{\delta}^{(k)}\}_{k\in\mathcal{S}}\right) := \sum_{k\in\mathcal{S}}\left\{\ell\left(\boldsymbol{\beta}^{(0)} + \boldsymbol{\delta}^{(k)}, \mathcal{D}^{(k)}\right) + P_{\lambda_k}(\boldsymbol{\delta}^{(k)})\right\} + \ell\left(\boldsymbol{\beta}^{(0)}, \mathcal{D}^{(0)}\right) + P_{\lambda_0}(\boldsymbol{\beta}^{(0)}).$$

*The regularization terms $P_{\lambda_0}$ and $P_{\lambda_k}$ satisfy some general conditions that are widely used for high-dimensional regression; see Appendix C for a more detailed discussion.*

**Remark 1.** *In general, the regularizers $P_{\lambda_0}$ and $P_{\lambda_k}$ can be customized and set as many famous regularizers, including the convex Lasso (Tibshirani, 1996) and nonconvex SCAD (Fan and Li, 2001). The estimation procedure follows by iteratively updating $\boldsymbol{\beta}^{(0)}$ and $\{\boldsymbol{\delta}^{(k)}\}_{k\in\mathcal{S}}$ using the coordinate descent algorithm (Breheny and Huang, 2011).*

Note that CoRT simultaneously estimates the target parameter $\boldsymbol{\beta}^{(0)}$ and the contrasts $\boldsymbol{\delta}^{(k)}$. One specific advantage of CoRT compared to the well-known Trans-GLM (Tian and Feng, 2023) with oracle $\mathcal{S}$ is that CoRT allows one to provide insights into which covariates are transferable in the $k$-th source domain. Specifically, one can obtain the set containing the transferable covariates in the $k$-th source model by $\{j : \hat{\delta}_j^{(k)} = 0, 1 \leq j \leq p\}$. In contrast, Trans-GLM fuses all the sources in $\mathcal{S}$ together to estimate the pre-trained parameter $\boldsymbol{\beta}_\mathcal{S}$ and the fused level contrast $\boldsymbol{\beta}^{(0)} - \boldsymbol{\beta}_\mathcal{S}$, which is unable to identify source-level transferable covariates. Similar limitation also exists in other two-step algorithms for multiple sources (Bastani, 2021; Li et al., 2022).

This paper focuses on GLMs primarily because this allows us to make head-to-head comparisons with the two-step seminal framework both theoretically and empirically; see Table 1 and Section 4.1. However, we would like to emphasize that the CoRT framework is model-agnostic, making it able to be generalized beyond GLMs naturally by simply replacing $\boldsymbol{\beta}^{(k)}$ and $\boldsymbol{\delta}^{(k)}$.

**Comparison to Multi-Task Learning.** Although the objective of CoRT (1) is similar to those objectives of multi-task learning (Zhang and Yang, 2021), their internal logic is fundamentally different. We compare CoRT to the following multi-task learning with biased-regularization (MTL-BR) framework, which, under the same problem setup, can be expressed as

$$\text{MTL-BR: } \hat{\boldsymbol{\beta}}^{(0)}, \{\hat{\boldsymbol{\delta}}^{(k)}\}_{k\in\mathcal{S}} = \underset{\boldsymbol{\beta}^{(0)},\{\boldsymbol{\delta}^{(k)}\}_{k\in0\cup\mathcal{S}}}{\operatorname{argmin}} \sum_{k\in0\cup\mathcal{S}}\left\{\ell\left(\boldsymbol{\beta} + \boldsymbol{\delta}^{(k)}, \mathcal{D}^{(k)}\right) + P_{\lambda_k}(\boldsymbol{\delta}^{(k)})\right\}. \qquad (2)$$

The MTL-BR is the backbone of various modern multi-task learning algorithms (Denevi et al., 2019; Duan and Wang, 2023; Tian et al., 2025). It decouples the learning of $\boldsymbol{\beta}^{(k)}$ as $\boldsymbol{\beta} + \boldsymbol{\delta}^{(k)}$, where $\boldsymbol{\beta}$ denotes the shared parameter and $\boldsymbol{\delta}^{(k)}$ denotes the task-specific parameter. By comparing with (2), it is clear that the CoRT objective (1) is asymmetric w.r.t. the source and target models and is target-centric, i.e., putting $\boldsymbol{\beta}^{(0)}$ as centre. Consequently, the tasks in MTL equally affect the shared parameter, whereas the source tasks in CoRT affect the target only as their parameters are regularized towards $\boldsymbol{\beta}^{(0)}$. Besides, the target-centric nature of CoRT allows $\boldsymbol{\beta}^{(0)}$ to remain in the source loss function and exploit the benefit of using source samples; see Theorem 1 for justification.

## 2.2 Adaptive Knowledge Transfer

In real-world applications, it is often challenging to determine which source belongs to the oracle set $\mathcal{S}$ for effective knowledge transfer. Therefore, the algorithm should exclude outlier sources during training, which is a practically challenging task (Pan and Yang, 2009).

To address this practical challenge, we propose a data-driven Algorithm 1 using the majority vote to adaptively detect "similar" sources without accessing $h$. The algorithm is based on the intuition that if incorporating the data from a similar source enhances learning over the target domain, then the error of $\hat{\boldsymbol{\beta}}^{(0k)}$, trained on $\mathcal{D}^{(0)} \cup \mathcal{D}^{(k)}$, should not exceed that of $\hat{\boldsymbol{\beta}}^{(0)}$, trained solely on $\mathcal{D}^{(0)}$.

---

**Algorithm 1:** Adaptive Co-Regularization Transfer

---

1: **Input:** Datasets $\mathcal{D}^{(k)}$ for $0 \le k \le K$.
2: **Data splitting:** randomly split the target data $\mathcal{D}^{(0)}$ into $T$ (an odd number) parts of equal size and denote them as $\mathcal{D}^{(0)}_{[1]}, \cdots, \mathcal{D}^{(0)}_{[T]}$.
3: **for** $k = 1$ **to** $K$ **do**
4:     **for** $t = 1$ **to** $T$ **do**
5:         Obtain $\hat{\boldsymbol{\beta}}^{(0)}_{[t]}$ by running Lasso on $\mathcal{D}^{(0)} \backslash \mathcal{D}^{(0)}_{[t]}$.
6:         Obtain $\hat{\boldsymbol{\beta}}^{(0k)}_{[t]}$ by running Lasso on $(\mathcal{D}^{(0)} \backslash \mathcal{D}^{(0)}_{[t]}) \cup \mathcal{D}^{(k)}$.
7:         Calculate errors over $\mathcal{D}^{(0)}_{[t]}$, $\ell(\hat{\boldsymbol{\beta}}^{(0)}_{[t]}, \mathcal{D}^{(0)}_{[t]})$ and $\ell(\hat{\boldsymbol{\beta}}^{(0k)}_{[t]}, \mathcal{D}^{(0)}_{[t]})$, respectively.
8:     **end for**
9:     Define $\mathcal{C}\left(\hat{\boldsymbol{\beta}}^{(0k)}\right)$ as $\sum_{t=1}^{T} \mathbb{1}\left\{\ell(\hat{\boldsymbol{\beta}}^{(0k)}_{[t]}, \mathcal{D}^{(0)}_{[t]}) = \min_{\boldsymbol{\beta} \in \{\hat{\boldsymbol{\beta}}^{(0)}_{[t]}, \hat{\boldsymbol{\beta}}^{(0k)}_{[t]}\}} \ell(\boldsymbol{\beta}, \mathcal{D}^{(0)}_{[t]})\right\}$.
10: **end for**
11: Obtain the estimator of $\mathcal{S}$ as $\hat{\mathcal{S}} = \left\{k : \mathcal{C}\left(\hat{\boldsymbol{\beta}}^{(0k)}\right) \ge (T+1)/2\right\}$.
12: Obtain $\hat{\boldsymbol{\beta}}^{(0)}$ by running CoRT with $\mathcal{D}^{(k)}, k \in \hat{\mathcal{S}} \cup \{0\}$.

---

One notable algorithm for identifying "similar" and "outlier" sources is from Tian and Feng (2023), which divides each source into three folds for validation and relies on two hyperparameters to determine whether a source should be included in $\hat{\mathcal{S}}$. These hyperparameters play a crucial role in the detection and can degrade performance if incorrectly selected, sometimes performing worse than single-task learning. We emphasize that while our adaptive algorithm presents a similar spirit to that in Tian and Feng (2023), it essentially differs in two key aspects: (1) we utilize a $T$-fold cross-validation to assess the impact of incorporating a specific source task; (2) we determine $\hat{\mathcal{S}}$ through a majority vote approach, eliminating the need for hyperparameter tuning.

## 2.3 Inference for Target Tasks

In this section, we aim to construct an asymptotic element-wise confidence interval (CI) for $\boldsymbol{\beta}^{(0)}$. Given the difficulty of building CI in a high-dimensional setting, we adopt the desparsified technique from Van de Geer et al. (2014) to our CoRT framework, which consists of two main steps. First, we construct the covariance matrix for target tasks and an asymptotic unbiased estimator through the desparsified technique. We then construct CI via the precision matrix. In the following, we briefly introduce the CI construction process and refer to Algorithm 4 in Appendix D for all details.

**Step 1: Target Covariance Matrix.** The population target covariate matrix is defined as $\boldsymbol{\Sigma}^{(0)}_{\boldsymbol{\beta}^{(0)}} = \mathrm{E}[\mathbf{x}^{(0)}(\mathbf{x}^{(0)})^T \psi''((\mathbf{x}^{(0)})^T \boldsymbol{\beta}^{(0)})]$, and the empirical version is defined as $\hat{\boldsymbol{\Sigma}}^{(0)}_{\hat{\boldsymbol{\beta}}^{(0)}} := \frac{1}{n_0} \mathbf{X}^{(0)\top}_{\hat{\boldsymbol{\beta}}^{(0)}} \mathbf{X}^{(0)}_{\hat{\boldsymbol{\beta}}^{(0)}}$,

$$\text{where } \mathbf{X}^{(0)}_{\hat{\boldsymbol{\beta}}^{(0)}} := \sqrt{\mathrm{diag}\left(\psi''(\mathbf{x}^{(0)\top}_i \hat{\boldsymbol{\beta}}^{(0)}), \cdots, \psi''(\mathbf{x}^{(0)\top}_{n_0} \hat{\boldsymbol{\beta}}^{(0)})\right)} \mathbf{X}^{(0)}.$$

**Step 2: Asymptotic Unbiased Estimator.** We now obtain the explicit form of the desparsified estimator for $\boldsymbol{\beta}^{(0)}$ under the CoRT framework, which is asymptotically unbiased.

**Proposition 1.** *Let $\hat{\boldsymbol{\beta}}^{(0)}$ denote the estimator of $\boldsymbol{\beta}^{(0)}$ obtained by minimizing the objective (1) and let $\hat{\boldsymbol{\Theta}}$ be a relaxed inverse of $\hat{\boldsymbol{\Sigma}}^{(0)}_{\hat{\boldsymbol{\beta}}^{(0)}}$, then the desparsified estimator of $\hat{\boldsymbol{\beta}}^{(0)}$ is*

$$\hat{\mathbf{b}}^{(0)} = \hat{\boldsymbol{\beta}}^{(0)} + \frac{1}{n_0} \hat{\boldsymbol{\Theta}} \mathbf{X}^{(0)\top} \left\{\mathbf{y}^{(0)} - \psi'(\mathbf{X}^{(0)} \hat{\boldsymbol{\beta}}^{(0)})\right\}. \tag{3}$$

**Remark 2.** *We highlight that the desparsified estimator (3) is not a simple extension of those in the literature (Van de Geer et al., 2014; Gueuning and Claeskens, 2016). Specifically, the desparsified estimator is constructed under the multi-source CoRT framework, while the classical desparsified estimators in the literature are constructed in a single-source setting.*

With the asymptotic normality of $\hat{\mathbf{b}}^{(0)}$ (proved in Theorem 3), one can construct the $(1-\alpha)$-CI for each coordinate of $\boldsymbol{\beta}^{(0)}$. The remaining task is to construct an estimator for the precision matrix $\hat{\Theta}$, which we finish by incorporating data from similar source tasks and applying the node-wise regression technique introduced by Meinshausen and Bühlmann (2006). This process also provides an "improved" estimator for the precision matrix compared to the single-task learning setting as indicated by Theorem 3, and thus can contribute to the precision matrix estimation problem.

## 3 Theoretical Analysis

In this section, we present the main theoretical analyses for CoRT, focusing on estimation and inference, along with their interpretations. Complete details, including the assumptions and proofs, can be found in the Appendix E and F, respectively. First, we state some standard assumptions under high-dimensional multi-source settings.

**Assumption 1.** *There exists some positive constant $\alpha_u$, such that $\psi''(t) \leq \alpha_u$, for all $t \in \mathbb{R}$.*

Assumption 1 restricts $\psi''$ in a bounded region, and we note that both linear and logistic regression satisfy it (Negahban et al., 2012; Loh and Wainwright, 2015). The purpose of this assumption is to ensure a finite variance.

**Assumption 2.** *For the $k$-th source, let $\mathcal{E}^{(k)}(\boldsymbol{\Delta}) = \langle \nabla\mathcal{L}_{\mathcal{D}^{(k)}}(\boldsymbol{\beta}+\boldsymbol{\Delta}) - \nabla\mathcal{L}_{\mathcal{D}^{(k)}}(\boldsymbol{\beta}), \boldsymbol{\Delta} \rangle$. We impose the following restricted strong convexity (RSC) condition*

*1. $\mathcal{E}^{(k)}(\boldsymbol{\Delta}) \geq \alpha_{1k}\|\boldsymbol{\Delta}\|_2^2 - \tau_{1k}\frac{\log p}{n_k}\|\boldsymbol{\Delta}\|_1^2, \forall\|\boldsymbol{\Delta}\|_2 \leq 1;$*

*2. $\mathcal{E}^{(k)}(\boldsymbol{\Delta}) \geq \alpha_{2k}\|\boldsymbol{\Delta}\|_2 - \tau_{2k}\sqrt{\frac{\log p}{n_k}}\|\boldsymbol{\Delta}\|_1, \forall\|\boldsymbol{\Delta}\|_2 \geq 1.$*

*where $\alpha_{jk} > 0$ and $\tau_{jk} \geq 0$ with $4\alpha_{jk} \geq 3\tau_{jk}$ for $j = \{1,2\}$ and $k \in \mathcal{S} \cup \{0\}$.*

Assumption 2 assumes the RSC condition for the loss function in each source. This is widely used to study the error bounds in high-dimensional statistics (Loh and Wainwright, 2015).

**Assumption 3.** *$\mathbf{x}_i^{(k)}$, $k \in \mathcal{S} \cup \{0\}$, are independent and identically distributed (i.i.d) sub-Gaussian random vector with mean zero and covariance matrix $\boldsymbol{\Sigma}^{(k)}$. The smallest and largest eigenvalues of $\boldsymbol{\Sigma}^{(k)}$ are bounded below and above by a constant, respectively.*

The sub-Gaussian assumption allows us to establish concentration inequalities for linear functions of those predictors. Particularly, it deals with sparse high-dimensional data where only a small number of predictors may be relevant due to the exponentially decaying tail bounds. The bounded eigenvalues of the covariance matrix ensure the well-posedness of the problem and the stability of the estimator.

### 3.1 Main Theoretical Results for Estimation.

For CoRT with $\mathcal{S}$, we have the following $\ell_1/\ell_2$-estimation and prediction error bounds.

**Theorem 1** (Errors for CoRT). *Denote $N := n_0 + n_{\mathcal{S}}$. Suppose Assumptions 1- 3 hold. For $k \in \mathcal{S}$, assume that $\|\boldsymbol{\beta}^{(0)}\|_0 \leq s$, $\|\boldsymbol{\delta}^{(k)}\|_0 \leq Cs$ for a constant $C$, and $(s\log p/N)^{\frac{1}{2}} + h^{\frac{1}{2}}(\log p/N)^{\frac{1}{4}} = o(1)$. Taking $\lambda_k \asymp \sqrt{\log p/N}$, then*

$$\ell_1 \text{ estimation error: } \|\hat{\boldsymbol{\beta}}^{(0)} - \boldsymbol{\beta}^{(0)}\|_1 \lesssim s\left(\frac{\log p}{N}\right)^{\frac{1}{2}} + \left(\frac{\log p}{N}\right)^{\frac{1}{4}}(sh)^{\frac{1}{2}},$$

$$\ell_2 \text{ estimation error: } \|\hat{\boldsymbol{\beta}}^{(0)} - \boldsymbol{\beta}^{(0)}\|_2 \lesssim \left(\frac{s\log p}{N}\right)^{\frac{1}{2}} + \left(\frac{\log p}{N}\right)^{\frac{1}{4}}h^{\frac{1}{2}},$$

$$\text{Prediction error: } D^2(\boldsymbol{\beta}^{(0)}, \hat{\boldsymbol{\beta}}^{(0)}) \lesssim \frac{s\log p}{N} + \left(\frac{\log p}{N}\right)^{\frac{3}{4}}(sh)^{\frac{1}{2}} + h\left(\frac{\log p}{N}\right)^{\frac{1}{2}},$$

hold with probability at least $1 - cp^{-1}$ for some positive c. Here $D^2(\boldsymbol{\beta}^{(0)}, \hat{\boldsymbol{\beta}}^{(0)})$ is the prediction error defined as $D^2(\boldsymbol{\beta}^{(0)}, \hat{\boldsymbol{\beta}}^{(0)}) = \langle \nabla \ell(\hat{\boldsymbol{\beta}}^{(0)}) - \nabla \ell(\boldsymbol{\beta}_0), \hat{\boldsymbol{\beta}}^{(0)} - \boldsymbol{\beta}^{(0)} \rangle$.

Table 1: Comparison of high-dimensional transfer learning models.

| Model | $\ell_2$ upper bound | Allow both convex and nonconvex $P_{\lambda_k}$? | Conditions for improvement over Lasso with $n_0 \ll n_{\mathcal{S}}$ |
|---|---|---|---|
| naive-Lasso (Tibshirani, 1996) | $\left( \frac{s \log p}{n_0} \right)^{1/2}$ | ✗ | - |
| Trans-Lasso (Li et al., 2022) | $\left( \frac{s \log p}{N} \right)^{1/2} + \left[ \left( \frac{\log p}{n_0} \right)^{\frac{1}{4}} h^{\frac{1}{2}} \right] \wedge h$ | ✗ | $h \ll s \left( \frac{\log p}{n_0} \right)^{1/2}$ |
| Trans-GLM (Tian and Feng, 2023) | $\left( \frac{s \log p}{N} \right)^{1/2} + \left[ \left( \frac{\log p}{n_0} \right)^{\frac{1}{4}} h^{\frac{1}{2}} \right] \wedge h$ | ✗ | $h \ll s \left( \frac{\log p}{n_0} \right)^{1/2}$ |
| CoRT (ours) | $\left( \frac{s \log p}{N} \right)^{\frac{1}{2}} + \left( \frac{\log p}{N} \right)^{\frac{1}{4}} h^{\frac{1}{2}}$ | ✓ | $h \ll s \left( \frac{(n_0 + n_{\mathcal{S}}) \log p}{n_0^2} \right)^{1/2}$ |

**Discussion.** If $h \lesssim s \sqrt{\log p / N}$, then the $\ell_2$ estimation and prediction error bounds become $\sqrt{s \log p / N}$, which is the convergence rate of fusing all source data and the target data. Besides, when $h \ll s \sqrt{N \log p} / n_0$ and $n_0 \ll n_{\mathcal{S}}$, the upper bound is sharper than implementing Lasso on the target data only (naive-Lasso), i.e., $\sqrt{s \log p / n_0}$. We present a comparison between CoRT and existing two-step algorithms for high-dimensional models, as reported in Table 1. Specifically, when presenting an improved error bound over the naive-Lasso, CoRT results show it allows for higher heterogeneous source tasks, i.e., a larger $h$, compared to the other approaches. This illustrates the capability of CoRT to more efficiently leverage the knowledge from less "similar" sources.

Furthermore, the two-step framework, like Trans-Lasso and Trans-GLM, typically requires the model/parameter shift assumption, i.e., only the parameters $\boldsymbol{\beta}^{(k)}$ differ across domains. To accommodate covariate shifts, they require additional similarity assumptions (Assumption 4 in Tian and Feng (2023)) between the covariance matrices. This amplifies the error bounds by a factor $C_\Sigma$ due to an asymptotic bias, which can diverge when the covariance matrices $\boldsymbol{\Sigma}^{(k)}$ are dissimilar. In contrast, CoRT avoids similarity assumptions on $\boldsymbol{\Sigma}^{(k)}$ and accommodates both shifts simultaneously, hence achieving robustness against covariate shift.

**Remark 3.** *Proving Theorem 1 presents new challenges. The existing two-step framework typically applies Lasso twice, making its error bounds decompose naturally by each step. In contrast, CoRT requires different technical tools since the objective* (1) *incorporates both target and source data, which we must express via a composite design matrix, thus introducing non-i.i.d. samples under co-regularization. Bounding the estimation error of GLMs typically needs an RSC condition. However, the existing RSC condition for high-dimensional GLMs fails in the non-i.i.d. case. We extend RSC to a non-i.i.d. setting and derive all subsequent theoretical results (see Appendix E).*

We next show that Algorithm 1 can adapt to unknown similarity, i.e., achieving the same performance as knowing $\mathcal{S}$ with high probability.

**Assumption 4** (An informal version of Assumptions 7)**.** *The heterogeneity between "similar" $\boldsymbol{\beta}^{(k)}$ ($k \in \mathcal{S}$) and "outlier" $\boldsymbol{\beta}^{(k)}$ ($k \in \mathcal{S}^c$) is sufficiently "large" so that they can be "separable".*

**Theorem 2** (Consistency of $\hat{\mathcal{S}}$)**.** *Let $\hat{\mathcal{S}}$ be the estimated set from Algorithm 1. Suppose Assumption 1-4 hold, then, for any $\delta > 0$, there exists $n(\delta)$ such that when $\min_{k \in \mathcal{S} \cup \{0\}} n_k \geq n(\delta)$, $P(\hat{\mathcal{S}} = \mathcal{S}) \geq 1 - \delta$ holds.*

### 3.2 Main Theoretical Results for Inference.

We present analyses for error bounds of the target precision matrix $\boldsymbol{\Theta}$ and CI via CoRT. To streamline the presentation, we provide informal versions of the assumptions used to derive theoretical results in the inference process, deferring formal details to Appendix F.

**Assumption 5** (An informal version of Assumption 8)**.** *The difference in the precision matrices of the similar source and target domains can be well-controlled by certain quantities $h_1$, $h_2$, and $h_3$.*

**Assumption 6** (An informal version of Assumption 9)**.** *Under some standard assumptions for high-dimensional GLMs, the CoRT desparsified estimator from Proposition 1 is consistent.*

**Theorem 3.** *Suppose Assumption 5-6 hold, and let* $R_1 = (s \log p/N)^{\frac{1}{2}} + (\log p/N)^{\frac{1}{4}} h^{\frac{1}{2}}$, *and* $\xi = R_1 + h_1^{\frac{1}{2}}(\log p/N)^{\frac{1}{4}} + h_1^{\frac{1}{2}} R_1^{\frac{1}{2}} + h_2 + h_3$. *Suppose* $R_1^2 = o(n_0^{-\frac{1}{2}})$, *then for* $j = 1, \cdots, p$, *the followings hold with probabilities at least* $1 - |\mathcal{S}| n_0^{-1}$,

$$\|\hat{\boldsymbol{\Theta}}_j - \boldsymbol{\Theta}_j\|_2 \lesssim \xi, \quad |(\hat{\boldsymbol{\Sigma}}_{\hat{\mathbf{b}}^{(0)}})_{jj} - \boldsymbol{\Theta}_{jj}| \lesssim h_1 + s^{\frac{1}{2}} \xi.$$

*Moreover, assume* $|(\hat{\boldsymbol{\Sigma}}_{\hat{\mathbf{b}}^{(0)}})_{jj} - \boldsymbol{\Theta}_{jj}| = o_P(1)$, *then*

$$\frac{\sqrt{n_0}(\hat{b}_j - \beta_j^{(0)})}{\sqrt{(\hat{\boldsymbol{\Sigma}}_{\hat{\mathbf{b}}^{(0)}})_{jj}}} \xrightarrow{d} \mathcal{N}(0,1), \quad with \quad \hat{\boldsymbol{\Sigma}}_{\hat{\mathbf{b}}^{(0)}} = \hat{\boldsymbol{\Theta}}^\top \left( \sum_{k \in \{0\} \cup \mathcal{S}} \frac{n_k}{N} \hat{\boldsymbol{\Sigma}}_{\hat{\boldsymbol{\beta}}^{(0)}}^{(k)} \right) \hat{\boldsymbol{\Theta}}. \quad (4)$$

Compared to the convergence rate of desparsified Lasso (Van de Geer et al., 2014), our rate improves when $h \ll s (\frac{N \log p}{n_0^2})^{\frac{1}{2}}$, $h_1 \ll (\frac{s \log p}{n_0})^{\frac{1}{2}} [s^{\frac{1}{2}} \wedge (\frac{N}{n_0})^{\frac{1}{2}}]$, $n_0 \ll N$, $h_1^{\frac{1}{2}} h^{\frac{1}{4}} \ll s^{\frac{1}{2}} (\log p)^{3/8} N^{1/8} n_0^{-\frac{1}{2}}$, $h_2 \ll (\frac{s \log p}{n_0})^{\frac{1}{2}}$, and $h_3 \ll (\frac{s \log p}{n_0})^{\frac{1}{2}}$. Therefore, the CI constructed by CoRT is narrower than the target-only ones. Again, compared to the two-step CI construction in Tian and Feng (2023), our approach accommodates "less similar" source tasks to achieve the same desired improved CI.

## 4 Empirical Results

In this section, we conduct empirical studies on synthetic data and real COVID-19 data to demonstrate the effectiveness of CoRT. [2]

### 4.1 Simulation

In this section, we compare CoRT to the existing methods, including (1) naive-Lasso: Lasso regression on the target data only, which serves as the baseline of our study; (2) Trans-GLM (Tian and Feng, 2023): the two-step transfer model for GLMs. CoRT (Lasso) and CoRT (SCAD) represent the training with Lasso and SCAD regularizers, respectively.

**Simulation with** $\mathcal{S}$**:** In this part, our goal is to show the performance of CoRT against naive-Lasso and Trans-GLM given known $\mathcal{S}$. For the target model, we set $\boldsymbol{\beta}^{(0)} = (0.5, -0.4, 0.7, -0.3, 0.8, \mathbf{0}_{495})$, where $\mathbf{0}_{495}$ means 495 repetitions of 0. The target covariates $\mathbf{x}_i^{(0)}$ are generated from $\mathcal{N}(\mathbf{0}_p, \boldsymbol{\Sigma})$ with element $\Sigma_{jj'} = 0.5^{|j-j'|}$ for $i = 1, \cdots, n_0$ and $1 \le j, j' \le p$. For the $k$-th source model, we set $\boldsymbol{\beta}^{(k)} = \boldsymbol{\beta}^{(0)} + (h/p)\mathcal{R}_p$, where $\mathcal{R}_p$ is a $p$-dimensional independent Rademacher variable. The source covarite is generated from $\mathcal{N}(\mathbf{0}_p, \boldsymbol{\Sigma} + \boldsymbol{\epsilon}\boldsymbol{\epsilon}^\top)$ with $\boldsymbol{\epsilon} \sim \mathcal{N}(\mathbf{0}_p, 0.3^2 \mathbf{I}_p)$ and the same $\boldsymbol{\Sigma}$ as used in the target data. We set $h \in \{5, 10, 20, 40\}$, $n_0 \in \{50, 75, 100\}$, and $n_k = 200$ for all $k = 1, \cdots, |\mathcal{S}|$.

Estimation results of logistic regression are presented in Figure 2a (Prediction results in Appendix G). The $\ell_2$ estimation errors for both transfer algorithms decrease as $|\mathcal{S}|$ increases, and both outperform the baseline naive-Lasso. In addition, while the simulation settings satisfy the condition $h \ll s\sqrt{\log p/n_0}$ in Theorem 1 of Tian and Feng (2023) in which regime the Trans-GLM is better than the naive-Lasso, CoRT also outperforms Trans-GLM, which supports our claim that CoRT has a wide positive transfer regime against Trans-GLM.

We also examine the performance of CI construction through the precision matrix estimation, the interval length, and the coverage probability, where these three aspects are evaluated by (1) the averaged absolute error $1/p \sum_{j=1}^p |\hat{\boldsymbol{\Theta}}_j^\top \hat{\boldsymbol{\Sigma}}_{\hat{\boldsymbol{\beta}}^{(0)}} \hat{\boldsymbol{\Theta}}_j - \boldsymbol{\Theta}_{jj}|$; (2) the averaged CI length of $\beta_j^{(0)}$; (3) the averaged coverage probability of $\beta_j^{(0)}$, for $j = 1, \cdots, p$, respectively. Figure 2b shows the results with different $|\mathcal{S}|$. For linear regression, CoRT has the lowest estimation error of the precision matrix and the best coverage probability, particularly when the number of transferable source data increases. However, the CI length tends to be wider, which also happens to Trans-GLM. For logistic regression, CoRT shows superior performance in all three metrics while desparsified Lasso and Trans-GLM are unable to maintain the nominal level $95\%$ and have large estimation errors of the precision matrix. We include more experiments to demonstrate the performance of logistic regression, including $n_0 = 40, h = 30, p = 800$ in Figure 2c to support our results.

---

[2]Code is available at https://github.com/shuoshuoliu/UTrans-CoRT.

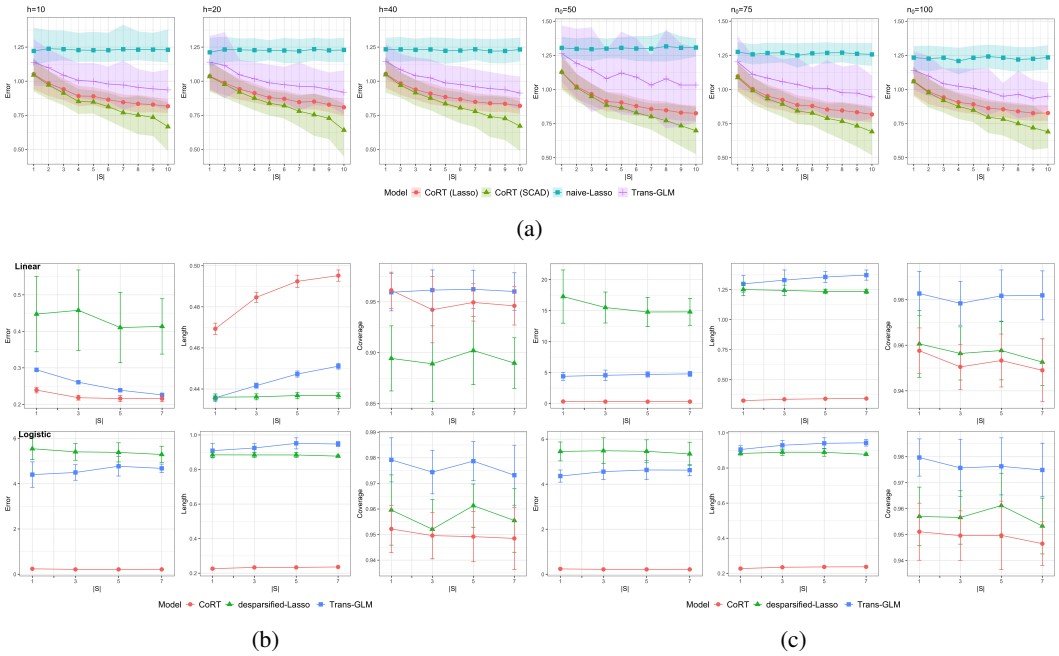

(a)

(b)                                                    (c)

Figure 2: Results of the CoRT and the existing TL methods: (a) Estimation errors of logistic regression with varying $h$ and $n_0$. Shade areas denote mean squared error (MSE) $\pm$ standard deviation (SD). (b) Inference results for linear and logistic regression. (c) More scenarios in logistic regression. Methods with lower error, shorter length, and near $95\%$ coverage have better performance.

**Adaptive Transfer in General Settings:**    In this section, we compare adaptive CoRT and source-detection Trans-GLM (Tian and Feng, 2023). We set $K = 10$ and simulate source data either in $\mathcal{S}$ or $\mathcal{S}^c$. Throughout this simulation, we fix $s = 10$, $p = 500$, and $n_k = 100, k = 0, \cdots, 10$. The target parameter is set to be $\boldsymbol{\beta}^{(0)} = (\mathbf{0.4}_3, \mathbf{0.5}_3, \mathbf{0.6}_4, \mathbf{0}_{490})$ and $h \in \{20, 40, 80\}$. For the source in $\mathcal{S}$, we let $\boldsymbol{\beta}^{(k)} = \boldsymbol{\beta}^{(0)} + h/p\mathcal{R}_p$. For sources in $\mathcal{S}^c$, we simulate $\boldsymbol{\beta}^{(k)}$ by $\beta_j^{(k)} = 0.5 + 2hr_j^{(k)}/p$ if $j \in \{s+1, \ldots, 2s\} \cup \mathrm{rand}(s)$ or $\beta_j^{(k)} = 2hr_j^{(k)}$ otherwise, where $\{r_j^{(k)}\}$ are Rademacher variables and $\mathrm{rand}(s)$ is a randomly generated index set of size $s$ from $\{2s+1, \cdots, p\}$. Figure 3 presents the $\ell_2$ estimation errors of all methods under different $h$ for both linear and logistic regression. Models start with "Pooled", which means that we directly pool all sources without adaptive procedures.

Two insights can be gained in the Figure 3. First, by comparing the pooled and adaptive versions, we can see that the adaptive versions always produce lower errors than the non-adaptive ones. This highlights the importance of excluding "outlier" sources before conducting knowledge transfer. Second, our CoRT outperforms Trans-GLM in most scenarios. Besides, adaptive CoRT is guaranteed not to be worse than the baseline naive-Lasso, but Trans-GLM fails to present such robustness.

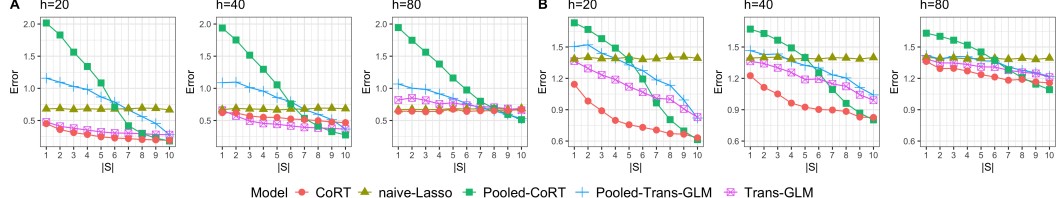

Figure 3: The $\ell_2$ estimation errors for linear regression (Panel A) and logistic regression (Panel B).

## 4.2   Real Data Example

In this experiment, we use county-level COVID-19 data to study (1) if we can improve predictions of a state's mortality rate using data from other states and (2) if we can identify states that are similar to

others. Details are available in Appendix G. Each time, we treat one state as the target and others as sources. We include two-way interactions of features, resulting in a total of $561$ features.

**Predictive analysis.** We compare our CoRT to Trans-GLM and other algorithms, including random forests (RF), XGBoost, and support vector machine (SVM), which only run on the target data. We randomly split the target data into $80\%$ for training and the remaining for testing. The results are presented in Table 2. We also employed feature extraction methods, including principal component analysis (PCA) for dimension reduction and neural networks. Both have explosive prediction errors and fail to work in our application since the sample size of training data is too small, around 50. Notably, CoRT performs the best in more than half of the target states. When CoRT is not the best, it still outperforms its interpretable competitor, Trans-GLM, in most states.

Table 2: The mean squared prediction errors for each target state. The mortality rate is multiplied by $10^5$. The bold numbers indicate the lowest prediction errors. Underlined numbers are not significantly different based on the t-test.

| Model | AL | CA | CO | FL | LA | MT | ND | NY | PA | SD | WI | WV |
|---|---|---|---|---|---|---|---|---|---|---|---|---|
| SVM | 0.28 | 4.43 | 9.26 | 0.21 | **0.15** | 6.91 | 4.59 | 0.68 | 0.24 | 4.53 | 0.29 | **4.26** |
| RF | 0.29 | **3.22** | **8.94** | 0.23 | 0.18 | 7.75 | **4.35** | 0.56 | 0.21 | **3.92** | 0.28 | 5.15 |
| XGBoost | 0.48 | 6.03 | 13.21 | 0.26 | 0.26 | 10.43 | 6.02 | 0.59 | 0.28 | 5.81 | 0.36 | 8.21 |
| Trans-GLM | 0.30 | 5.03 | 9.82 | 0.21 | 0.19 | 6.52 | 5.25 | 0.62 | 0.21 | 4.73 | 0.28 | 5.13 |
| CoRT | **0.26** | 4.35 | 9.02 | **0.18** | 0.19 | **6.14** | 5.22 | **0.52** | **0.19** | 4.66 | **0.27** | 4.69 |

**Network analysis.** We analyze the transferability of the source states to the target states. For each target state, we use Algorithm 1 to adaptively estimate $\mathcal{S}$ and CoRT to obtain $\hat{\boldsymbol{\delta}}^{(k)}$. Figure 4 shows the transfer network after 200 replications, where a presented directed edge indicates the source state gets selected at least 150 times, with thickness reflecting the average magnitude of $\|\hat{\boldsymbol{\delta}}_k\|^2$. We observe that (1) CA, MT, and WV have the most connections to source states due to their fewer counties, improving predictions with more source states. (2) Geographical proximity influences transfers, as neighboring states share similar county characteristics. For example, GA is close to FL and WV, leading to connections between them.

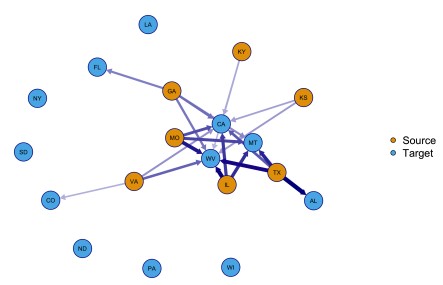

Figure 4: Transferabilities among the states. The thicker the edge is, the more transferable the source state is.

## 5 Discussion

In this paper, we present a Co-Regularization transfer framework, which enhances the multi-source knowledge transfer process in high-dimensions. We show that the CoRT framework outperforms the classical two-step paradigm both theoretically and empirically. Moreover, the model-agnostic nature of CoRT makes it highly adaptable and can be extended to other high-dimensional statistical models, further broadening its applicability across various fields, including social and biological sciences. We also develop mathematical tools for such a co-regularization framework, paving the way for theoretically driven transfer learning research.

There are also open problems left for this topic. For example, we notice that CoRT requires retraining when switching to new target tasks, resulting in an increased computational burden. Distributed techniques, such as divide-and-conquer, can be used to practically improve computational efficiency. The algorithm for divide-and-conquer in the high dimension regime has been well-developed with theory to guarantee its performance. These distributed algorithms can be directly plugged into the CoRT framework in practice when computation expense is a concern. Developing an alternative training scheme and leveraging the co-regularization spirit can also be of interest.

**Acknowledgment**: Li's research was partially supported by an NSF grant DMS 2514400 and an NIH grant R01GM163244.

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

*Supplement to*

# "Co-Regularization Enhances Knowledge Transfer in High Dimensions"

Appendix organization:

## A   Notations

In this section, we list the comprehensive notations that are used in the paper and the Appendix.

We denote scalars with unbolded letters (e.g., sample size $n$ and dimensionality $p$), (random) vectors with boldface lowercase letters (e.g., $\mathbf{y}$ and $\boldsymbol{\beta}$), and matrices with boldface capital letters (e.g., $\mathbf{X}$). Let $\{(\mathbf{x}^{(k)}, \mathbf{y}_k) : \mathbf{x}^{(k)} \in \mathbb{R}^{n_k \times p}, \mathbf{y}_k \in \mathbb{R}^{n_k}\}_{k=1}^{K'}$ denote the multiple source data and let $(\mathbf{X}^{(0)}, \mathbf{y}^{(0)})$ be the target data. We use $\top$ to represent the transpose of vectors or matrices, such as $\mathbf{x}^\top$ and $\mathbf{X}^\top$. For a $p$-dimensional vector $\mathbf{x} = (x_1, \cdots, x_p)$, the $\ell_q$ norm is $\|\mathbf{x}\|_q = (\sum_{i=1}^{p} |x_i|^q)^{1/q}$, the $\ell_0$ norm is the number of non-zero elements, and the infinity (maximum) norm is $\|\mathbf{x}\|_\infty = \max_i |x_i|$. Let $|\mathcal{M}|$ denote the cardinality of the set $\mathcal{M}$. and a set with superscript $c$ denotes its complement, for example, $\mathcal{M}^c$. Let $a \vee b$ and $a \wedge b$ denote the maximum and minimum between $a$ and $b$, respectively. We use letters $C$ and $c$ with different subscriptions (e.g., $C_1$ and $c_1$) to denote the positive and absolute constants. Let $a_n = O(b_n)$ denote $|a_n/b_n| \leq c$ for some constant $c$ when $n$ is large enough. Let $a_n = O_P(b_n)$ and $a_n \lesssim b_n$ denote $P(|a_n/b_n| \leq c) \to 1$ for $c < \infty$. Let $a_n = o_P(b_n)$ denote $P(|a_n/b_n| > c) \to 0$ for $c > 0$. Finally, $a_n \asymp b_n$ means that $a_n/b_n$ converges to some positive constant.

## B  Two-Step Framework for High-dimensional Models

In this section, we give a general description of a dominant two-step framework for transfer learning in high-dimensional models advocated by Li et al. (2022); Tian and Feng (2023). To facilitate comparison between this framework and our CoRT, we introduce this framework following the exact settings in Section 2.

Given the known $\mathcal{S}$, the Trans-GLM (Tian and Feng, 2023) estimates $\boldsymbol{\beta}^{(0)}$ through the following two steps by setting $\hat{\boldsymbol{\beta}}^{(0)} := \hat{\boldsymbol{\omega}}^{(\mathcal{S})} + \hat{\boldsymbol{\delta}}^{(0)}$, where

1. (Pre-training):

$$\hat{\boldsymbol{\omega}}^{(\mathcal{S})} = \arg\min_{\boldsymbol{\omega}} \left\{ \frac{1}{n_{\mathcal{S}} + n_0} \sum_{k \in \mathcal{S} \cup \{0\}} \ell(\boldsymbol{\beta}, \mathcal{D}^{(k)}) + \lambda_1 \|\boldsymbol{\omega}\|_1 \right\},$$

2. (Fine-tuning)

$$\hat{\boldsymbol{\delta}}^{(0)} = \arg\min_{\boldsymbol{\delta}} \left\{ \frac{1}{n_0} \ell(\hat{\boldsymbol{\omega}}^{(\mathcal{S})} + \boldsymbol{\delta}, \mathcal{D}^{(0)}) + \lambda_2 \|\boldsymbol{\delta}\|_1 \right\}.$$

Here, $\ell(\boldsymbol{\beta}, \mathcal{D}^{(k)})$ is the negative log-likelihood over $k$-th dataset with parameter $\boldsymbol{\beta}$, i.e., $= \frac{1}{n_k} \sum_{i=1}^{n_k} \psi(\mathbf{x}_i^{(k)\top} \boldsymbol{\beta}) - y_i^{(k)} (\mathbf{x}_i^{(k)\top} \boldsymbol{\beta})$.

The idea underpinning Trans-GLM is that for the source domains in set $\mathcal{S}$, the population version of the parameter $\boldsymbol{\beta}^{\mathcal{S}}$ is a mixture of $\boldsymbol{\beta}^{(k)}$, which is sparse and thus should be estimated with $\ell^1$-regularization. Besides, if $\boldsymbol{\beta}^{\mathcal{S}}$ is similar to $\boldsymbol{\beta}^0$, i.e., learning $\boldsymbol{\beta}^0$ can be accelerated by using the results of learning $\boldsymbol{\beta}^{\mathcal{S}}$, the contrast $\boldsymbol{\delta}^{(0)}$ should be sparse as well.

Under certain assumptions, the $\ell^1$ and $\ell^2$ estimation errors are

$$\ell_1 \text{ estimation error: } \|\hat{\boldsymbol{\beta}}^{(0)} - \boldsymbol{\beta}^{(0)}\|_1 \lesssim s \left( \frac{\log p}{N} \right)^{\frac{1}{2}} + s \left( \frac{\log p}{N} \right)^{\frac{1}{2}} \wedge h,$$

$$\ell_2 \text{ estimation error: } \|\hat{\boldsymbol{\beta}}^{(0)} - \boldsymbol{\beta}^{(0)}\|_2 \lesssim \left( \frac{s \log p}{N} \right)^{\frac{1}{2}} + \left( \frac{\log p}{N} \right)^{\frac{1}{2}} \wedge \left( \frac{\log p}{N} \right)^{\frac{1}{4}} h^{\frac{1}{2}} \wedge h.$$

## C  Regularizer Detail

In Section 2, we claim the regularization function $P_{\lambda_0}$ and $P_{\lambda_k}$ can be a general class that includes many famous regularizers, including Lasso, SCAD, MCP, etc. In the following, we provide specific conditions that the regularizers need to satisfy.

**Condition 1.** *We assume the regularizers $P_{\lambda_0}$ and $P_{\lambda_k}$ that used in CoRT satisfy the following conditions.*

(i) *$P_\lambda(0) = 0$ and $P_\lambda(t)$ is symmetric around 0.*

(ii) *$P_\lambda(t)$ is differentiable for $t \neq 0$ and $\lim_{t \to 0^+} P_\lambda'(t) = \lambda L$.*

(iii) *$P_\lambda(t)$ is a non-decreasing function on $t \in [0, \infty)$.*

(iv) *$P_\lambda(t)/t$ is a non-increasing function on $t \in (0, \infty)$.*

(v) *There exists $\gamma > 0$ such that $P_\lambda(t) + \frac{\gamma}{2} t^2$ is convex.*

**Remark 4.** *Conditions (i)–(iii) are relatively mild and used in Zhang and Zhang (2012). Condition (iv) makes sure that the bound of error $\|\hat{\boldsymbol{\beta}} - \boldsymbol{\beta}\|_2$ is vanishingly small. To incorporate the nonconvex regularizers, we include a convex side constraint $g$ that satisfies $R \geq g(\boldsymbol{\beta}) \geq \|\boldsymbol{\beta}\|_1$, where $\boldsymbol{\beta}$ lies in a convex set and $R$ is a tuning parameter that can be chosen to be proportional to $1/\lambda$ (Loh and Wainwright, 2015).*

# D  Construction Details for Confidence Interval

In this section, we provide the detailed process for constructing element-wise CI for $\boldsymbol{\beta}^{(0)}$. We begin by defining some notations. For any $\boldsymbol{\theta} \in \mathbb{R}^p$, we define the $k$-th diagonal weighted matrix as

$$\mathbf{W}_{\boldsymbol{\theta}}^{(k)} = \mathrm{diag}\left(\sqrt{\psi''(\mathbf{x}_i^{(k)\top}\boldsymbol{\theta})}, \cdots, \sqrt{\psi''(\mathbf{x}_{n_k}^{(k)\top}\boldsymbol{\theta})}\right),$$

the $k$-th weighted covariate as

$$\mathbf{X}_{\boldsymbol{\theta}}^{(k)} = \mathbf{W}_{\boldsymbol{\theta}}^{(k)}\mathbf{X}^{(k)},$$

and the empirical version as $\mathbf{X}_{\hat{\boldsymbol{\theta}}}^{(k)}$. Then, we define the $k$-th population weighted covariance matrix and its empirical version as

$$\boldsymbol{\Sigma}_{\boldsymbol{\omega}}^{(k)} = \mathrm{E}\left[\mathbf{x}^{(k)}\mathbf{x}^{(k)\top}\psi''(\mathbf{x}^{(k)\top}\boldsymbol{\omega})\right], \quad \hat{\boldsymbol{\Sigma}}_{\boldsymbol{\omega}}^{(k)} = \frac{1}{n_k}\mathbf{X}_{\hat{\boldsymbol{\theta}}}^{(k)\top}\mathbf{X}_{\hat{\boldsymbol{\theta}}}^{(k)}.$$

As stated in Section 2.3, the construction process is based on the following two steps:

1. Obtain an estimator for the precision matrix $\boldsymbol{\Theta}$.

2. Obtain the unbiased estimator of $\boldsymbol{\beta}^{(0)}$ by desparsified technique.

In the following, we present the details of each step.

---

**Algorithm 2:** Node-wise Regression for Constructing $\hat{\boldsymbol{\Theta}}$

1: **Input:** source data $\{\mathcal{D}^{(k)} = (\mathbf{X}^{(k)}, \mathbf{y}^{(k)}), k \in \mathcal{S}\}$, target data $\mathcal{D}^{(0)} = (\mathbf{X}^{(0)}, \mathbf{y}^{(0)})$.

2: **Output:** The relaxed inverse $\hat{\boldsymbol{\Theta}}$.

3: Obtain $\hat{\boldsymbol{\beta}}^{(0)}$ by CoRT.

4: For each $k \in \mathcal{S} \cup \{0\}$, we stack $\mathbf{X}_{\hat{\boldsymbol{\beta}}^{(0)}}^{(k)}$ horizontally and obtain a $N \times p$ covariate matrix $\mathbf{X}_{\hat{\boldsymbol{\beta}}^{(0)}}^{(\{0,1\cdots,|\mathcal{S}|\})}$

5: For each $j = 1, \cdots, p$, calculate the parameter $\boldsymbol{\gamma}_j$ of the node-wise regression of $\mathbf{X}_{\hat{\boldsymbol{\beta}}^{(0)},j}^{(\{0,1\cdots,|\mathcal{S}|\})}$ over $\mathbf{X}_{\hat{\boldsymbol{\beta}}^{(0)},-j}^{(\{0,1\cdots,|\mathcal{S}|\})}$ via Lasso, i.e.

$$\hat{\boldsymbol{\gamma}}_j = \arg\min_{\boldsymbol{\gamma}}\left\{-\frac{1}{2(n_\mathcal{S}+n_0)}\sum_{k\in\{0\}\cup\mathcal{S}}\left\|\mathbf{X}_{\hat{\boldsymbol{\beta}}^{(0)},j}^{(\{0,1\cdots,|\mathcal{S}|\})} - \mathbf{X}_{\hat{\boldsymbol{\beta}}^{(0)},-j}^{(\{0,1\cdots,|\mathcal{S}|\})}\boldsymbol{\gamma}\right\|_2^2 + \lambda_j\|\boldsymbol{\gamma}\|_1\right\}.$$

6: For each $j = 1, \cdots, p$, calculate

$$\hat{\tau}_{0j}^2 = \hat{\boldsymbol{\Sigma}}_{\boldsymbol{\theta},j,j}^{(0)} - \hat{\boldsymbol{\Sigma}}_{\boldsymbol{\theta},j,-j}^{(0)}\hat{\boldsymbol{\gamma}}_j.$$

7: Denote $\hat{\boldsymbol{\gamma}}_j = (\hat{\gamma}_{j,1}, \cdots, \hat{\gamma}_{j,j-1}, \hat{\gamma}_{j,j+1}, \cdots, \hat{\gamma}_{j,p})$ for each $j = 1, \cdots, p$, then construct the $\hat{\boldsymbol{\Theta}}$ as follows,

$$\hat{\boldsymbol{\Theta}} = \mathrm{diag}\left(\hat{\tau}_{01}^{-2}, \ldots, \hat{\tau}_{0p}^{-2}\right)\begin{pmatrix} 1 & -\hat{\gamma}_{1,2} & \cdots & -\hat{\gamma}_{1,p} \\ -\hat{\gamma}_{2,1} & 1 & \cdots & -\hat{\gamma}_{2,p} \\ \vdots & \vdots & \ddots & \vdots \\ -\hat{\gamma}_{p,1} & -\hat{\gamma}_{p,2} & \cdots & 1 \end{pmatrix}. \tag{5}$$

---

**Construction of $\hat{\boldsymbol{\Theta}}$ via Node-wise Regression.** In this part, we show the construction details of $\hat{\boldsymbol{\Theta}}$. This problem has been widely studied in the literature, and there are several techniques to construct a relaxed inverse of $\hat{\boldsymbol{\Sigma}}^{(0)}$, and our approach is based on node-wise regression.

For $k$-th weighted covariate, we further define $\mathbf{X}_{\boldsymbol{\theta},j}^{(k)}$ as the $j$-th column of $\mathbf{X}_{\boldsymbol{\theta}}^{(k)}$ and $\mathbf{X}_{\boldsymbol{\theta},-j}^{(k)}$ as the matrix without the $j$-th column. Further, define $\hat{\boldsymbol{\Sigma}}_{\boldsymbol{\theta},j,-j}^{(k)}$ the $j$-th row of $\hat{\boldsymbol{\Sigma}}_{\boldsymbol{\theta}}^{(k)}$ without the diagonal $(j,j)$ elements, and $\hat{\boldsymbol{\Sigma}}_{\boldsymbol{\theta},j,j}^{(k)}$ the diagonal $(j,j)$ elements of $\hat{\boldsymbol{\Sigma}}_{\boldsymbol{\theta}}^{(k)}$.

Given these notations, we now present the construction process in Algorithm 2.

**Construction of the Covariance of the Desparsified Estimator.** The following algorithm shows the procedure for constructing the covariance matrix of the desparsified estimator $\hat{\mathbf{b}}^0$.

---

**Algorithm 3:** Covariance of Desparsified Estimator

---

1: **Input:** source data $\{\mathcal{D}^{(k)} = (\mathbf{X}^{(k)}, \mathbf{y}^{(k)}), k \in \mathcal{S}\}$, target data $\mathcal{D}^{(0)} = (\mathbf{X}^{(0)}, \mathbf{y}^{(0)})$.

2: **Output:** The covariance matrix $\hat{\boldsymbol{\Sigma}}_{\hat{\mathbf{b}}^{(0)}}$.

3: Obtain $\hat{\boldsymbol{\beta}}^{(0)}$ by CoRT.

4: Obtain the relaxed inverse $\hat{\boldsymbol{\Theta}}$ via Algorithm 2.

5: Denote $\hat{\boldsymbol{\Sigma}}_{\hat{\boldsymbol{\beta}}^{(0)}} = \sum_{k \in \{0\} \cup \mathcal{S}} \frac{n_k}{N} \hat{\boldsymbol{\Sigma}}_{\hat{\boldsymbol{\beta}}^{(0)}}^{(k)}$ and the covariance matrix of $\hat{\mathbf{b}}^{(0)}$ is

$$\hat{\boldsymbol{\Sigma}}_{\hat{\mathbf{b}}^{(0)}} = \hat{\boldsymbol{\Theta}}^{\top} \hat{\boldsymbol{\Sigma}}_{\hat{\boldsymbol{\beta}}^{(0)}} \hat{\boldsymbol{\Theta}}$$

---

**Remark 5.** *The construction of $\hat{\boldsymbol{\Theta}}$ and $\hat{\boldsymbol{\Sigma}}_{\hat{\mathbf{b}}^{(0)}}$ originates from Van de Geer et al. (2014), and we notice that Tian and Feng (2023), who also considers the transfer learning under high-dimensional GLMs, presents a similar procedure. However, we would like to highlight the difference between our CI construction method and that of Tian and Feng (2023). In Tian and Feng (2023), the transfer learning approach is a two-step process, and accordingly, the node-wise regression to obtain $\hat{\gamma}$ is also a two-step. In contrast, our method only involves a one-shot loss minimization to obtain $\hat{\gamma}$, as we apply a co-learning procedure.*

With Algorithm 2 and Algorithm 3, the construction process of element-wise CI for $\boldsymbol{\beta}^{(0)}$ can be established in Algorithm 4. The general idea is to make sure the source tasks in $\mathcal{S}$ can be separated from source tasks in $\mathcal{S}^c$.

---

**Algorithm 4:** Confidence Interval for the Target parameter

---

1: **Input:** source data $\{\mathcal{D}^{(k)}\}_{k \in \mathcal{S}}$, target data $\mathcal{D}^{(0)}$, and confidence level $(1 - \alpha)$.

2: **Output:** $\mathrm{CI}(\boldsymbol{\beta}_j^{(0)}), j = 1, \cdots, p$.

3: Obatin the target task estimator $\hat{\boldsymbol{\beta}}^{(0)}$ by minimizing the co-learning loss (1).

4: Construct the related inverse $\hat{\boldsymbol{\Theta}}$, via Algorithm 2 in the Appendix, which leverages the node-wise Lasso technique over the target data $\mathcal{D}^{(0)}$.

5: Obtain the desparsified estimator $\hat{\mathbf{b}}^{(0)}$ via (3).

6: Construct the covariance $\hat{\boldsymbol{\Sigma}}_{\hat{\mathbf{b}}^{(0)}}$ of the desparsified estimator $\hat{\mathbf{b}}^{(0)}$ via Algorithm 3, which applies a covariate reweighting procedure over $\{\mathcal{D}^{(k)}\}$ for $k = \{0\} \cup \mathcal{S}$.

7: The confidence interval of $\boldsymbol{\beta}_j^{(0)}, j = 1, \cdots, p$ is

$$\left[ \hat{\mathbf{b}}_j^{(0)} - \frac{z_{1-\alpha/2}(\hat{\boldsymbol{\Sigma}}_{\hat{\mathbf{b}}^{(0)}})_{jj}}{\sqrt{n_0}}, \hat{\mathbf{b}}_j^{(0)} + \frac{z_{1-\alpha/2}(\hat{\boldsymbol{\Sigma}}_{\hat{\mathbf{b}}^{(0)}})_{jj}}{\sqrt{n_0}} \right],$$

where $z_{1-\alpha/2}$ is the $\alpha/2$ left tail quantile of standard normal distribution.

---

# E   Technical Details for Estimation

We begin our theoretical analysis with the estimation of CoRT and the detection algorithm based on CoRT.

## E.1   Assumptions

The following assumption is adopted from Assumption 5 in Tian and Feng (2023). We refer the readers there for details and the requirements for it to hold.

**Assumption 7.** *(Identifiability of $\mathcal{S}$) Let $\ell(\hat{\boldsymbol{\beta}}^{(0k)}, \mathcal{D}_{[t]}^{(0)})$ and $\ell(\hat{\boldsymbol{\beta}}^{(0)}, \mathcal{D}_{[t]}^{(0)})$ denote the $t$-th fold cross validation errors with $\hat{\boldsymbol{\beta}}^{(0k)}$ and $\hat{\boldsymbol{\beta}}^{(0)}$, respectively. Assume for some $h$ and for all $k$,*

$$P\left(\sup_s |\ell(\hat{\boldsymbol{\beta}}^{(0k)}, \mathcal{D}_{[t]}^{(0)}) - \ell(\hat{\boldsymbol{\beta}}^{(0)}, \mathcal{D}_{[t]}^{(0)})| > \Upsilon_1^{(k)} + \zeta\Gamma_1^{(k)}\right) \lesssim g_1^{(k)}(\zeta),$$

$$P\left(\sup_s |\ell(\hat{\boldsymbol{\beta}}^{(0k)}, \mathcal{D}_{[t]}^{(0)}) - \ell(\hat{\boldsymbol{\beta}}^{(0)}, \mathcal{D}_{[t]}^{(0)})| > \zeta\Gamma_2^{(k)}\right) \lesssim g_2^{(k)}(\zeta).$$

*For $k \in \mathcal{S}^c$, assume $\underline{\lambda} = \inf_k \Lambda_{min}(E[\int_0^1 \psi''((1-t)\mathbf{x}_0^\top \boldsymbol{\beta}^{(0)} + t\mathbf{x}_0^\top \boldsymbol{\beta}^{(k)})dt \cdot \mathbf{x}_0 \mathbf{x}_0^\top]) > 0$, $\Gamma_1^{(k)} = O(1)$, $\Gamma_2^{(k)} = O(1)$, and for sufficiently large $C_1$*

$$\|\boldsymbol{\beta}^{(k)} - \boldsymbol{\beta}^{(0)}\|_2 \geq \underline{\lambda}^{-\frac{1}{2}}\left[C_1\left(\sqrt{\Gamma_1^{(0)}} \vee \sqrt{\Gamma_2^{(0)}} \vee 1\right) + \sqrt{2\Upsilon_1^{(k)}}\right].$$

*For $k \in \mathcal{S}$, assume $\Upsilon_1^{(k)} + \Gamma_1^{(k)} + \Gamma_2^{(k)} + h^2 = O(1)$.*

We provide the explicit forms of $\Upsilon_1^{(k)}, \Gamma_1^{(k)}, \Gamma_2^{(k)}, g_1^{(k)}(\zeta)$, and $g_2^{(k)}(\zeta)$ in the proof of Theorem 2.

## E.2 Proofs for the Upper bound of CoRT

*Proof of Theorem 1.* We start by proving $\|\hat{\boldsymbol{\Delta}}\|_2 \leq 1$ with the appropriate choice of $\lambda$ and sufficient $n$. Note that the RSC condition has

$$\mathcal{E}_n(\hat{\boldsymbol{\Delta}}) = \langle \nabla\mathcal{L}_n(\hat{\boldsymbol{\beta}}) - \nabla\mathcal{L}_n(\boldsymbol{\beta}), \hat{\boldsymbol{\Delta}}\rangle \geq \alpha_2\|\hat{\boldsymbol{\Delta}}\|_2 - \tau_2\sqrt{\frac{\log p}{N}}\|\hat{\boldsymbol{\Delta}}\|_1. \tag{6}$$

The first-order condition returns

$$\langle \nabla\mathcal{L}_n(\hat{\boldsymbol{\beta}}) + \nabla P_\lambda(\hat{\boldsymbol{\beta}}), -\hat{\boldsymbol{\Delta}}\rangle \geq 0. \tag{7}$$

Thereby,

$$\langle -\nabla P_\lambda(\hat{\boldsymbol{\beta}}) - \nabla\mathcal{L}_n(\boldsymbol{\beta}), \hat{\boldsymbol{\Delta}}\rangle \geq \alpha_2\|\hat{\boldsymbol{\Delta}}\|_2 - \tau_2\sqrt{\frac{\log p}{N}}\|\hat{\boldsymbol{\Delta}}\|_1. \tag{8}$$

Holder inequality and the triangle inequality imply that

$$\langle -\nabla P_\lambda(\hat{\boldsymbol{\beta}}) - \nabla\mathcal{L}_n(\boldsymbol{\beta}), \hat{\boldsymbol{\Delta}}\rangle \leq \left\{\|\nabla P_\lambda(\hat{\boldsymbol{\beta}})\|_\infty + \|\nabla\mathcal{L}_n(\boldsymbol{\beta})\|_\infty\right\}\|\hat{\boldsymbol{\Delta}}\|_1.$$

Lemma 4 of Loh and Wainwright (2015) has $\|\nabla P_\lambda(\hat{\boldsymbol{\beta}})\|_\infty \le \lambda L$, so the next step is to bound

$$
\|\mathcal{L}_n(\boldsymbol{\beta})\|_\infty = \max \left| \frac{1}{n} \sum_{i=1}^{n} \left( \psi'(\mathbf{x}_i^\top \boldsymbol{\beta}) - y_i \right) x_{ij} \right|
$$

$$
= \max_{j=1,\cdots,p^*} \left| \frac{1}{n} \sum_{k \in \mathcal{S} \cup \{0\}} \sum_{i=1}^{n_k} \left( \psi'(\mathbf{x}_i^\top \boldsymbol{\beta}) - y_i^{(k)} \right) x_{ij} \right|, \quad x_{ij} = [x_{kij}, \cdots, 0, \cdots, x_{kij}]
$$

$$
= \max_{j=1,\cdots,p} \left| \frac{1}{n} \sum_{k \in \mathcal{S}} \sum_{i=1}^{n_k} \left[ \psi'(\mathbf{x}_i^{(k)\top}(\boldsymbol{\beta}^{(k)} - \boldsymbol{\beta}^{(0)}) + \mathbf{x}_i^{(k)\top} \boldsymbol{\beta}^{(0)}) - y_i^{(k)} \right] x_{kij} \right|
$$

$$
+ \max_{j=1,\cdots,p} \left| \frac{1}{n} \sum_{i=1}^{n_0} \left[ \psi'(\mathbf{x}_i^{(0)\top} \boldsymbol{\beta}^{(0)}) - y_i^{(0)} \right] x_{0ij} \right|
$$

$$
= \max_{j=1,\cdots,p} \left| \frac{1}{n} \sum_{k \in \mathcal{S}} \sum_{i=1}^{n_k} \left[ \psi'(\mathbf{x}_i^{(k)\top} \boldsymbol{\beta}^{(k)}) - y_i^{(k)} \right] x_{kij} \right| + \max_{j=1,\cdots,p} \left| \frac{1}{n} \sum_{i=1}^{n_0} \left[ \psi'(\mathbf{x}_i^{(0)\top} \boldsymbol{\beta}^{(0)}) - y_i^{(0)} \right] x_{0ij} \right|
$$

$$
\le \sum_{k \in \mathcal{S}} \max_{j=1,\cdots,p} \left| \frac{1}{n} \sum_{i=1}^{n_k} \left[ \psi'(\mathbf{x}_i^{(k)\top} \boldsymbol{\beta}^{(k)}) - y_i^{(k)} \right] x_{kij} \right| + \max_{j=1,\cdots,p} \left| \frac{1}{n} \sum_{i=1}^{n_0} \left[ \psi'(\mathbf{x}_i^{(0)\top} \boldsymbol{\beta}^{(0)}) - y_i^{(0)} \right] x_{0ij} \right|
$$

$$
= \sum_{k \in \mathcal{S}} \frac{n_k}{n} \max_{j=1,\cdots,p} \left| \frac{1}{n_k} \sum_{i=1}^{n_k} \left[ \psi'(\mathbf{x}_i^{(k)\top} \boldsymbol{\beta}^{(k)}) - y_i^{(k)} \right] x_{kij} \right| + \frac{n_0}{n} \max_{j=1,\cdots,p} \left| \frac{1}{n_0} \sum_{i=1}^{n_0} \left[ \psi'(\mathbf{x}_i^{(0)\top} \boldsymbol{\beta}^{(0)}) - y_i^{(0)} \right] x_{0ij} \right|
$$

$$
\le \sum_{k \in \mathcal{S} \cup \{0\}} c_k \sqrt{\frac{n_k \log p}{n^2}}
$$

$$
\asymp \sqrt{\frac{\log p}{N}}
$$

where the last inequality is due to the fact that each

$$
\max_{j=1,\cdots,p} \left| \frac{1}{n_k} \sum_{i=1}^{n_k} \left[ \psi'(\mathbf{x}_i^{(k)\top} \boldsymbol{\beta}^{(k)}) - y_i^{(k)} \right] x_{kij} \right| \le c_k \sqrt{\frac{\log p}{n_k}}
$$

with probability at least $1 - c_{1k} \exp(c_{2k} \log p)$. Therefore,

$$
\langle -\nabla P_\lambda(\hat{\boldsymbol{\beta}}) - \nabla \mathcal{L}_n(\boldsymbol{\beta}), \hat{\boldsymbol{\Delta}} \rangle \le (\lambda L + c_m \lambda) \|\hat{\boldsymbol{\Delta}}\|_1
$$

$$
\alpha_2 \|\hat{\boldsymbol{\Delta}}\|_2 - \tau_2 \sqrt{\frac{\log p}{N}} \|\hat{\boldsymbol{\Delta}}\|_1 \le (\lambda L + c_m \lambda) \|\hat{\boldsymbol{\Delta}}\|_1
$$

$$
\|\hat{\boldsymbol{\Delta}}\|_2 \le \frac{(\tau \lambda + L \lambda + c_m \lambda)}{\alpha_2} \|\hat{\boldsymbol{\Delta}}\|_1 \le \frac{2R}{\alpha_2} (\tau_2 + L + c_m) \lambda,
$$

where $c_m = \max_k c_k$, is at most 1 by carefully choosing $\tau_2$, $L$, and $c_m$. Finally, we would focus on the case

$$
\mathcal{E}_n(\hat{\boldsymbol{\Delta}}) \ge \alpha_1 \|\hat{\boldsymbol{\Delta}}\|_2^2 - \tau_1 \frac{\log p}{N} \|\hat{\boldsymbol{\Delta}}\|_1^2. \tag{9}
$$

Since $P_{\lambda,\mu}(\boldsymbol{\beta}) + \frac{\mu}{2} \|\boldsymbol{\beta}\|_2^2$ is convex,

$$
P_{\lambda,\mu}(\hat{\boldsymbol{\beta}}) - P_{\lambda,\mu}(\boldsymbol{\beta}) \le \langle \nabla P_{\lambda,\mu}(\hat{\boldsymbol{\beta}}), \hat{\boldsymbol{\Delta}} \rangle = \langle \nabla P_\lambda(\hat{\boldsymbol{\beta}}) + \mu \hat{\boldsymbol{\beta}}, \hat{\boldsymbol{\Delta}} \rangle
$$

implying

$$
\langle \nabla P_\lambda(\hat{\boldsymbol{\beta}}), \hat{\boldsymbol{\Delta}} \rangle \ge P_\lambda(\hat{\boldsymbol{\beta}}) - P_\lambda(\boldsymbol{\beta}) - \frac{\mu}{2} \|\hat{\boldsymbol{\Delta}}\|_2^2. \tag{10}
$$

Combining (7), (9), (10), we have

$$\alpha_1\|\hat{\boldsymbol{\Delta}}\|_2^2 - \tau_1\frac{\log p}{N}\|\hat{\boldsymbol{\Delta}}\|_1^2 \leq \langle\nabla\mathcal{L}_n(\hat{\boldsymbol{\beta}}) - \nabla\mathcal{L}_n(\boldsymbol{\beta}), \hat{\boldsymbol{\Delta}}\rangle$$

$$\leq -\langle\nabla P_\lambda(\hat{\boldsymbol{\beta}}), \hat{\boldsymbol{\Delta}}\rangle - \langle\nabla\mathcal{L}_n\boldsymbol{\beta}, \hat{\boldsymbol{\Delta}}\rangle$$

$$\leq P_\lambda(\boldsymbol{\beta}) - P_\lambda(\hat{\boldsymbol{\beta}}) + \frac{\mu}{2}\|\hat{\boldsymbol{\Delta}}\|_2^2 - \langle\nabla\mathcal{L}_n(\boldsymbol{\beta}), \hat{\boldsymbol{\Delta}}\rangle.$$

Thus,

$$\left(\alpha_1 - \frac{\mu}{2}\right)\|\hat{\boldsymbol{\Delta}}\|_2^2 \leq P_\lambda(\boldsymbol{\beta}) - P_\lambda(\hat{\boldsymbol{\beta}}) + \|\nabla L_n(\boldsymbol{\beta})\|_\infty\|\hat{\boldsymbol{\Delta}}\|_1 + \tau_1\frac{\log p}{N}\|\hat{\boldsymbol{\Delta}}\|_1^2$$

$$\leq P_\lambda(\boldsymbol{\beta}) - P_\lambda(\hat{\boldsymbol{\beta}}) + \left(\|\nabla\mathcal{L}_n(\boldsymbol{\beta})\|_\infty + 2R\tau_1\frac{\log p}{n}\right)\|\hat{\boldsymbol{\Delta}}\|_1$$

$$= P_\lambda(\boldsymbol{\beta}) - P_\lambda(\hat{\boldsymbol{\beta}}) + \left(\frac{L}{4}\lambda + \frac{L}{4}\lambda\right)\left(\frac{P_\lambda(\hat{\boldsymbol{\Delta}})}{\lambda L} + \frac{\mu\|\hat{\boldsymbol{\Delta}}\|_2^2}{2\lambda L}\right)$$

$$= P_\lambda(\boldsymbol{\beta}) - P_\lambda(\hat{\boldsymbol{\beta}}) + \frac{1}{2}P_\lambda(\hat{\boldsymbol{\Delta}}) + \frac{\mu}{4}\|\hat{\boldsymbol{\Delta}}\|_2^2$$

By taking $\mu = \tau_1$, we start from the bottom of Page 574 in Loh and Wainwright (2015). Let $\mathcal{M}$ be the support of $\boldsymbol{\beta}$, then

$$(\alpha_1 - 3\tau_1/4)\|\hat{\boldsymbol{\Delta}}\|_2^2 \leq P_\lambda(\boldsymbol{\beta}) - P_\lambda(\hat{\boldsymbol{\beta}}) + 1/2P_\lambda(\hat{\boldsymbol{\Delta}})$$

$$= P_\lambda(\boldsymbol{\beta}) - P_\lambda(\hat{\boldsymbol{\beta}}_\mathcal{M}) - P_\lambda(\hat{\boldsymbol{\beta}}_{\mathcal{M}^c}) + 1/2P_\lambda(\hat{\boldsymbol{\Delta}})$$

$$\leq P_\lambda(\hat{\boldsymbol{\Delta}}_\mathcal{M}) - P_\lambda(\hat{\boldsymbol{\beta}}_{\mathcal{M}^c}) + 1/2P_\lambda(\hat{\boldsymbol{\Delta}})$$

$$= 3/2P_\lambda(\hat{\boldsymbol{\Delta}}_\mathcal{M}) - P_\lambda(\hat{\boldsymbol{\Delta}}_{\mathcal{M}^c}) + 1/2P_\lambda(\hat{\boldsymbol{\Delta}}_{\mathcal{M}^c})$$

$$= 3/2P_\lambda(\hat{\boldsymbol{\Delta}}_\mathcal{M}) - 1/2P_\lambda(\hat{\boldsymbol{\Delta}}_{\mathcal{M}^c})$$

$$\leq 3/2P_\lambda(\hat{\boldsymbol{\Delta}}_{\mathbb{1}}) - 1/2P_\lambda(\hat{\boldsymbol{\Delta}}_{\mathbb{1}^c})$$

$$\leq \lambda L\left(3/2\|\hat{\boldsymbol{\Delta}}_{\mathbb{1}}\|_1 - 1/2\|\hat{\boldsymbol{\Delta}}_{\mathbb{1}^c}\|_1\right)$$

where the last inequality follows Lemma 2 (b). Therefore, we have

$$(2\alpha_1 - 3\tau_1/2)\|\hat{\boldsymbol{\Delta}}\|_2^2 \leq \lambda L\left(3\|\hat{\boldsymbol{\Delta}}_{\mathbb{1}}\|_1 - \|\hat{\boldsymbol{\Delta}}_{\mathbb{1}^c}\|_1\right)$$

$$\leq 3\lambda L\|\hat{\boldsymbol{\Delta}}_{\mathbb{1}}\|_1 + \lambda LKh$$

$$\lesssim 3\lambda\sqrt{s}\|\hat{\boldsymbol{\Delta}}\|_2 + \lambda h.$$

The proof is then concluded by solving a quadratic inequality.

**The $\ell_2$ prediction error:**

The prediction error is upper-bounded such that

$$\langle\nabla\mathcal{L}_n(\hat{\boldsymbol{\beta}}^{(0)}) - \nabla\mathcal{L}_n(\boldsymbol{\beta}^{(0)}), \hat{\boldsymbol{\Delta}}_0\rangle \leq \langle-\nabla\mathcal{L}_n(\hat{\boldsymbol{\beta}}^{(0)}) - \nabla P_\lambda(\hat{\boldsymbol{\beta}}^{(0)}), \hat{\boldsymbol{\Delta}}_0\rangle$$

$$\leq P_\lambda(\boldsymbol{\beta}^{(0)}) - P_\lambda(\hat{\boldsymbol{\beta}}^{(0)}) + \tau/2\|\hat{\boldsymbol{\Delta}}_0\|_2^2 + \|\nabla\mathcal{L}_n(\boldsymbol{\beta}^{(0)})\|_\infty\|\hat{\boldsymbol{\Delta}}_0\|_1.$$

Next, we have the following bounds:

$$P_\lambda(\boldsymbol{\beta}^{(0)}) - P_\lambda(\hat{\boldsymbol{\beta}}^{(0)}) \leq \lambda L\|\hat{\boldsymbol{\Delta}}_0\|_1 \quad \text{and} \quad \|\nabla\mathcal{L}_n(\boldsymbol{\beta}^{(0)})\|_\infty \lesssim \lambda.$$

Finally, the prediction error

$$\langle\nabla\mathcal{L}_n(\hat{\boldsymbol{\beta}}^{(0)}) - \mathcal{L}_n(\boldsymbol{\beta}^{(0)}), \hat{\boldsymbol{\Delta}}_0\rangle$$

$$\lesssim \lambda L\|\hat{\boldsymbol{\Delta}}_0\|_1 + \frac{\tau}{2}\|\hat{\boldsymbol{\Delta}}_0\|_2^2 + \lambda\|\hat{\boldsymbol{\Delta}}_0\|_1$$

$$\lesssim \lambda\sqrt{s}\|\hat{\boldsymbol{\Delta}}_0\|_2 + \|\hat{\boldsymbol{\Delta}}_0\|_2^2$$

The result follows by plugging in the $\ell_2$ error bound in Theorem 1 such that

$$D^2(\boldsymbol{\beta}^{(0)}, \hat{\boldsymbol{\beta}}^{(0)}) \lesssim \frac{s \log p}{n} + \left(\frac{\log p}{N}\right)^{\frac{3}{4}} \sqrt{sh} + h\sqrt{\frac{\log p}{N}}.$$

$\square$

## E.3 Proof for the Consistency in Detection

*Proof of Theorem 2.* We first obtain the Explicit forms of $\Upsilon_1^{(k)}, \Gamma_1^{(k)}, \Gamma_2^{(k)}, g_1^{(k)}(\zeta)$, and $g_2^{(k)}(\zeta)$.

Let $\boldsymbol{\beta}^{(k)}$ be the parameter for the $k$-th source model. Denote the $\boldsymbol{\beta}^{(0k)}$ as the probability limited of $\boldsymbol{\beta}^{(k)}$ for population version loss function, and its corresponding estimator $\hat{\boldsymbol{\beta}}^{(0k)}$. The following forms can be found in Proposition 1 of Tian and Feng (2023):

$$\Omega_k = \begin{cases} \sqrt{\frac{s \log p}{n_0}}, & k = 0 \\ \sqrt{\frac{s \log p}{n_k + n_0}} + \left(\frac{\log p}{n_k + n_0}\right)^{1/4} \sqrt{h} + \sqrt{sh}, & k \in \mathcal{S} \\ h'\sqrt{\frac{\log p}{n_k + n_0}} + \sqrt{\frac{s' \log p}{n_k + n_0}} W_k + \left(\frac{\log p}{n_k + n_0}\right)^{1/4} \sqrt{h' W_k}, & k \in \mathcal{S}^c, \end{cases}$$

where $W_k = 1 \vee \left\|\boldsymbol{\beta}^{(0k)} - \boldsymbol{\beta}^{(0)}\right\|_2 \vee \left\|\boldsymbol{\beta}^{(0k)} - \boldsymbol{\beta}^{(k)}\right\|_2$.

The followings list the forms of $\Upsilon_1^{(k)}, \Gamma_1^{(k)}, \Gamma_2^{(k)}, g_1^{(k)}(\zeta)$, and $g_2^{(k)}(\zeta)$.

(i) Linear model:

$$\Gamma_1^{(0)} = \sqrt{\frac{s \log p}{n_0}} \|\boldsymbol{\beta}^{(0)}\|_2, \quad \Gamma_2^{(0)} = \left(\|\boldsymbol{\beta}_0\|_2^2 \vee \|\boldsymbol{\beta}^{(0)}\|_2\right) / \sqrt{n_0},$$

$$\Upsilon_1^{(k)} = \Omega_k \cdot \left[\left\|\boldsymbol{\beta}^{(k)}\right\|_2 \mathbb{I}(k \in \mathcal{S}) + \|\boldsymbol{\beta}_{0k}\|_2 \mathbb{I}(k \in \mathcal{S}^c)\right],$$

$$\Gamma_1^{(k)} = \sqrt{\frac{1}{n_0}} \Omega_k \cdot \left[\left\|\boldsymbol{\beta}^{(k)}\right\|_2 \mathbb{I}(k \in \mathcal{S}) + \|\boldsymbol{\beta}_{0k}\|_2 \mathbb{I}(k \in \mathcal{S}^c)\right],$$

$$\Gamma_2^{(k)} = \sqrt{\frac{1}{n_0}} \left[\left(\left\|\boldsymbol{\beta}^{(k)}\right\|_2^2 \vee \|\boldsymbol{\beta}_k\|_2\right) \mathbb{I}(k \in \mathcal{S}) + \left(\left\|\boldsymbol{\beta}^{(0k)}\right\|_2^2 \vee \left\|\boldsymbol{\beta}^{(0k)}\right\|_2\right) \mathbb{I}(k \in \mathcal{S}^c)\right],$$

$$g_1^{(k)}(\zeta) = g_2^{(k)}(\zeta) = \exp(-\zeta^2), k \neq 0; g_1^{(0)}(\zeta) = \exp(-\zeta^2), g_2^{(0)}(\zeta) = \exp(-n_0) + \exp(-\zeta^2).$$

(ii) Logistic model:

$$\Gamma_1^{(0)} = \sqrt{\frac{s \log p}{n_0}}, \quad \Gamma_2^{(0)} = \|\boldsymbol{\beta}^{(0)}\|_2 / \sqrt{n_0},$$

$$\Upsilon_1^{(k)} = \Omega_k, \quad \Gamma_1^{(k)} = \sqrt{\frac{1}{n_0}} \Omega_k, \quad \Gamma_2^{(k)} = \sqrt{\frac{1}{n_0}} \left[\|\boldsymbol{\beta}_k\|_2 \mathbb{I}(k \in \mathcal{S}) + \left\|\boldsymbol{\beta}^{(0k)}\right\|_2 \mathbb{I}(k \in \mathcal{S}^c)\right],$$

$$g_1^{(k)}(\zeta) = g_2^{(k)}(\zeta) = \exp(-\zeta^2).$$

Let $\ell(\hat{\boldsymbol{\beta}}^{(0k)}, \mathcal{D}_{[t]}^{(0)})$ and $\ell(\hat{\boldsymbol{\beta}}^{(0)}, \mathcal{D}_{[t]}^{(0)})$ denote the $t$-th fold cross validation errors with $\hat{\boldsymbol{\beta}}^{(0k)}$ and $\hat{\boldsymbol{\beta}}^{(0)}$, respectively. Note that they are averaged by the testing data size. First, we have the following

$$
P(\hat{\mathcal{S}} \neq \mathcal{S}) \leq P\left(\bigcup_{k \in \mathcal{S}} \left\{\mathcal{C}\left(\hat{\boldsymbol{\beta}}^{(0k)}\right) < \mathcal{C}\left(\hat{\boldsymbol{\beta}}^{(0)}\right)\right\} \bigcup \bigcup_{k \in \mathcal{S}^c} \mathcal{C}\left(\hat{\boldsymbol{\beta}}^{(0k)}\right) \geq \mathcal{C}\left(\hat{\boldsymbol{\beta}}^{(0)}\right)\right)
$$

$$
\leq \sum_{k \in \mathcal{S}} P\left(\mathcal{C}\left(\hat{\boldsymbol{\beta}}^{(0k)}\right) < \frac{S+1}{2}\right) + \sum_{k \in \mathcal{S}^c} P\left(\mathcal{C}\left(\hat{\boldsymbol{\beta}}^{(0k)}\right) \geq \frac{S+1}{2}\right)
$$

$$
= \sum_{k \in \mathcal{S}} P\left(\sum_{s \in \mathcal{S}} I\left(\ell(\hat{\boldsymbol{\beta}}^{(0k)}, \mathcal{D}_{[t]}^{(0)}) \leq \ell(\hat{\boldsymbol{\beta}}^{(0)}, \mathcal{D}_{[t]}^{(0)})\right) < \frac{S+1}{2}\right) +
$$

$$
\sum_{k \in \mathcal{S}^c} P\left(\sum_{s \in \mathcal{S}} I\left(\ell(\hat{\boldsymbol{\beta}}^{(0k)}, \mathcal{D}_{[t]}^{(0)}) \leq \ell(\hat{\boldsymbol{\beta}}^{(0)}, \mathcal{D}_{[t]}^{(0)})\right) \geq \frac{S+1}{2}\right)
$$

$$
\leq \sum_{k \in \mathcal{S}} \left(\frac{S+1}{2} p_\epsilon\right) + \sum_{k \in \mathcal{S}^c} \frac{2}{S+1} E\left(\sum_{s \in \mathcal{S}} I\left(\ell(\hat{\boldsymbol{\beta}}^{(0k)}, \mathcal{D}_{[t]}^{(0)}) \leq \ell(\hat{\boldsymbol{\beta}}^{(0)}, \mathcal{D}_{[t]}^{(0)})\right)\right)
$$

$$
= \frac{1}{S} \sum_{k \in \mathcal{S}} \sum_{s \in \mathcal{S}} \left\{1 - P\left(\ell(\hat{\boldsymbol{\beta}}^{(0k)}, \mathcal{D}_{[t]}^{(0)}) - \ell(\hat{\boldsymbol{\beta}}^{(0)}, \mathcal{D}_{[t]}^{(0)}) \leq 0\right)\right\} +
$$

$$
\frac{2}{S+1} \sum_{k \in \mathcal{S}^c} \sum_{s \in \mathcal{S}} \left\{1 - P\left(\ell(\hat{\boldsymbol{\beta}}^{(0k)}, \mathcal{D}_{[t]}^{(0)}) - \ell(\hat{\boldsymbol{\beta}}^{(0)}, \mathcal{D}_{[t]}^{(0)}) > 0\right)\right\},
$$

where the last inequality is due to Lemma 3 and Markov's inequality.

Next, we need to prove, for any $s$,

(i) $p_\epsilon$ holds with a low probability. In other words, $P\left(\ell(\hat{\boldsymbol{\beta}}^{(0k)}, \mathcal{D}_{[t]}^{(0)}) - \ell(\hat{\boldsymbol{\beta}}^{(0)}, \mathcal{D}_{[t]}^{(0)}) \leq 0\right), k \in \mathcal{S}$, holds with a high probability.

(ii) $P\left(\ell(\hat{\boldsymbol{\beta}}^{(0k)}, \mathcal{D}_{[t]}^{(0)}) - \ell(\hat{\boldsymbol{\beta}}^{(0)}, \mathcal{D}_{[t]}^{(0)}) > 0\right), k \in \mathcal{S}^c$, holds with a high probability.

Intuitively, for $k \in \mathcal{S}$, $\hat{\boldsymbol{\beta}}^{(k)}$ is obtained with a larger sample size than $\hat{\boldsymbol{\beta}}^{(0)}$, so we expect a lower cross-validation error, i.e., $\ell(\hat{\boldsymbol{\beta}}^{(0k)}, \mathcal{D}_{[t]}^{(0)}) - \ell(\hat{\boldsymbol{\beta}}^{(0)}, \mathcal{D}_{[t]}^{(0)}) \leq 0$. For $k \in \mathcal{S}^c$, $\hat{\boldsymbol{\beta}}^{(k)}$ is obtained with data deviating from the target data, so we expect a higher cross-validation error, i.e., $\ell(\hat{\boldsymbol{\beta}}^{(0k)}, \mathcal{D}_{[t]}^{(0)}) - \ell(\hat{\boldsymbol{\beta}}^{(0)}, \mathcal{D}_{[t]}^{(0)}) > 0$.

For (i), using claims in the proof of Theorem 4 in Tian and Feng (2023), we know

$$
\sup_s |\ell(\hat{\boldsymbol{\beta}}^{(0k)}, \mathcal{D}_{[t]}^{(0)}) - \ell(\hat{\boldsymbol{\beta}}^{(0)}, \mathcal{D}_{[t]}^{(0)})| \lesssim \zeta \left\{\Gamma_1^{(k)} + \Gamma_2^{(k)} + h^2\right\} \xrightarrow{n_k \to \infty} 0,
$$

for $k \in \mathcal{S} \cup \{0\}$, holds with probability at least $1 - 2\exp(-\zeta^2)$.

For (ii), from the proof of Theorem 4 in Tian and Feng (2023), we see

$$
\inf_s \ell(\hat{\boldsymbol{\beta}}^{(0k)}, \mathcal{D}_{[t]}^{(0)}) - \ell(\hat{\boldsymbol{\beta}}^{(0)}, \mathcal{D}_{[t]}^{(0)}) > C_0(\hat{\sigma} \vee 1) > 0
$$

holds with probability at least $1 - 2\left[\exp(-C_0^{-2}) + \exp(-\zeta^2)\right]$ for $C_0 > 0$ and $\zeta > 0$.

Combining (i) and (ii), we have

$$
P(\hat{\mathcal{S}} \neq \mathcal{S}) \leq \frac{1}{S} \sum_{k \in \mathcal{S}} \sum_{s=1}^S 2\exp(-\zeta^2) + \frac{2}{S+1} \sum_{k \in \mathcal{S}^c} \sum_{s=1}^S \left[2\exp(-C_0^{-2}) + 2\exp(-\zeta^2)\right]
$$

$$
= 2|\mathcal{S}| \exp(-\zeta^2) + \frac{2S}{S+1} |\mathcal{S}^c| \left[2\exp(-C_0^{-2}) + 2\exp(-\zeta^2)\right].
$$

For any $\delta > 0$, there exist $C_0 > 0$ and $\zeta > 0$ such that

$$2|\mathcal{S}|\exp\left(-\zeta^2\right) \leq \frac{\delta}{2}, \quad \frac{2S}{S+1}|\mathcal{S}^c|\left[2\exp\left(-C_0^{-2}\right) + 2\exp\left(-\zeta^2\right)\right] \leq \frac{\delta}{2}.$$

In summary, for any $\delta > 0$, there exist $C_0(\delta) > 0$ and $\zeta(\delta) > 0$ such that

$$P(\hat{\mathcal{S}} = \mathcal{S}) \geq 1 - \delta, \quad \text{as} \quad n_k \to \infty, k \in \mathcal{S} \cup \{0\}.$$

$\square$

## E.4 Lemma

**Lemma 1.** *Suppose Assumption 2 holds. Let $n > 4R^2\tau_1^2 \log p$ for some positive constants $\alpha_1, \tau_1$ and $R$ proportional to $1/\lambda$ and $\hat{\boldsymbol{\Delta}} = \hat{\boldsymbol{\beta}} - \boldsymbol{\beta}$, then*

$$\mathcal{E}_n(\hat{\boldsymbol{\Delta}}) \geq \begin{cases} \alpha_1\|\hat{\boldsymbol{\Delta}}\|_2^2 - \tau_1\dfrac{\log p}{N}\|\hat{\boldsymbol{\Delta}}\|_1^2, & \|\hat{\boldsymbol{\Delta}}\|_2 \leq 1 & \text{(11a)} \\[2ex] \alpha_2\|\hat{\boldsymbol{\Delta}}\|_2 - \tau_2\sqrt{\dfrac{\log p}{N}}\|\hat{\boldsymbol{\Delta}}\|_1, & \|\hat{\boldsymbol{\Delta}}\|_2 \geq 1. & \text{(11b)} \end{cases}$$

*Proof.* For any $\boldsymbol{\Delta} \in \mathbb{R}^p$, we assume $\|\boldsymbol{\Delta}\|_1 \leq 2R$ since $\boldsymbol{\beta}$ and $\boldsymbol{\beta} + \boldsymbol{\Delta}$ lie in the feasible set. We first note that

$$\begin{aligned}
\mathcal{E}^{(k)}(\boldsymbol{\Delta}) &= \left\langle \nabla\mathcal{L}_{\mathcal{D}^{(k)}}\left(\boldsymbol{\beta}^{(k)} + \boldsymbol{\Delta}\right) - \nabla\mathcal{L}_{\mathcal{D}^{(k)}}\left(\boldsymbol{\beta}^{(k)}\right), \boldsymbol{\Delta} \right\rangle \\
&= \frac{1}{n_k}\sum_{i=1}^{n_k}\left[\psi'(\langle \mathbf{x}_i^{(k)}, \boldsymbol{\beta}^{(k)} + \boldsymbol{\Delta}\rangle) - \psi'(\langle \mathbf{x}_i^{(k)}, \boldsymbol{\beta}^{(k)}\rangle)\right]\mathbf{x}_i^{(k)\top}\boldsymbol{\Delta} \\
&= \frac{1}{n_k}\sum_{i=1}^{n_k}\psi''\left(\langle \mathbf{x}_i^{(k)}, \boldsymbol{\beta}^{(k)}\rangle + t_{ki}\langle \mathbf{x}_i^{(k)}, \boldsymbol{\Delta}\rangle\right) \cdot \left(\langle \mathbf{x}_i^{(k)}, \boldsymbol{\Delta}\rangle\right)^2
\end{aligned}$$

by the mean value theorem, for some $t_{ki} \in [0, 1]$. For our model, let $\boldsymbol{\Delta} = (\boldsymbol{\Delta}_1^\top, \cdots, \boldsymbol{\Delta}_K^\top, \boldsymbol{\Delta}_0^\top)^\top \in \mathbb{R}^{p^*}$ and $\mathbf{x}_i^{(k)*} \in \mathbb{R}^{p^*}$ denote the $i$-th observation of the $k$-th data in $\mathbf{X}$. Note that $\mathbf{x}_i^{(k)*}$ contains many 0s as constructed in $\mathbf{X}$. Then we have

$$\begin{aligned}
\mathcal{E}_n(\boldsymbol{\Delta}) &= \frac{1}{n}\left[\sum_{i=1}^{n_1}\psi''\left(\langle \mathbf{x}_i^{(1)*}, \boldsymbol{\beta}\rangle + t_{1i}\langle \mathbf{x}_i^{(1)*}, \boldsymbol{\Delta}\rangle\right) \cdot \left(\langle \mathbf{x}_i^{(1)*}, \boldsymbol{\Delta}\rangle\right)^2 + \cdots \right. \\
&\quad + \sum_{i=1}^{n_K}\psi''\left(\langle \mathbf{x}_i^{(k)*}, \boldsymbol{\beta}\rangle + t_{Ki}\langle \mathbf{x}_i^{(k)*}, \boldsymbol{\Delta}\rangle\right) \cdot \left(\langle \mathbf{x}_i^{(k)*}, \boldsymbol{\Delta}\rangle\right)^2 \\
&\quad \left. + \sum_{i=1}^{n_0}\psi''\left(\langle \mathbf{x}_i^{(0)*}, \boldsymbol{\beta}\rangle + t_{0i}\langle \mathbf{x}_i^{(0)*}, \boldsymbol{\Delta}\rangle\right) \cdot \left(\langle \mathbf{x}_i^{(0)*}, \boldsymbol{\Delta}\rangle\right)^2\right] \\
&= \frac{1}{n}\left[\sum_{i=1}^{n_1}\psi''\left(\mathbf{x}_i^{(1)\top}\boldsymbol{\beta}^{(1)} + t_{1i}\mathbf{x}_i^{(1)\top}\left(\boldsymbol{\Delta}_1 + \boldsymbol{\Delta}_0\right)\right) \cdot \left\{\mathbf{x}_i^{(1)\top}\left(\boldsymbol{\Delta}_1 + \boldsymbol{\Delta}_0\right)\right\}^2 + \cdots \right. \\
&\quad \left. + \sum_{i=1}^{n_0}\psi''\left(\mathbf{x}_i^{(0)\top}\boldsymbol{\beta}^{(0)} + t_{0i}\mathbf{x}_i^{(0)\top}\boldsymbol{\Delta}_0\right) \cdot \left(\mathbf{x}_i^{(0)\top}\boldsymbol{\Delta}_0\right)^2\right] \\
&= \frac{1}{n}\left[n_1\mathcal{E}_{n_1}(\boldsymbol{\Delta}_1 + \boldsymbol{\Delta}_0) + \cdots + n_K\mathcal{E}^{(k)}(\boldsymbol{\delta}^{(k)} + \boldsymbol{\Delta}_0) + n_0\mathcal{E}_{n_0}(\boldsymbol{\Delta}_0)\right] \\
&\geq \sum_{k \in \mathcal{S}}\frac{n_k}{n}\left(\alpha_k\|\boldsymbol{\delta}^{(k)} + \boldsymbol{\Delta}_0\|_2^2 - \tau_k\sqrt{\frac{\log p}{n_k}}\|\boldsymbol{\delta}^{(k)} + \boldsymbol{\Delta}_0\|_1\|\boldsymbol{\delta}^{(k)} + \boldsymbol{\Delta}_0\|_2\right) \\
&\quad + \frac{n_0}{n}\left(\alpha_0\|\boldsymbol{\Delta}_0\|_2^2 - \tau_0\sqrt{\frac{\log p}{n_0}}\|\boldsymbol{\Delta}_0\|_1\|\boldsymbol{\Delta}_0\|_2\right),
\end{aligned}$$

where the last inequality follows the result (Eq. 65) of Loh and Wainwright (2015) that

$$\mathcal{E}^{(k)}(\boldsymbol{\delta}^{(k)}+\boldsymbol{\Delta}_0) \geq \alpha_k\|\boldsymbol{\delta}^{(k)}+\boldsymbol{\Delta}_0\|_2^2 - \tau_k\sqrt{\frac{\log p}{n_k}}\|\boldsymbol{\delta}^{(k)}+\boldsymbol{\Delta}_0\|_1\|\boldsymbol{\delta}^{(k)}+\boldsymbol{\Delta}_0\|_2, \quad \forall\|\boldsymbol{\delta}^{(k)}+\boldsymbol{\Delta}_0\|_2 \leq 1$$

$$\mathcal{E}_{n_0}(\boldsymbol{\Delta}_0) \geq \alpha_0\|\boldsymbol{\Delta}_0\|_2^2 - \tau_0\sqrt{\frac{\log p}{n_0}}\|\boldsymbol{\Delta}_0\|_1\|\boldsymbol{\Delta}_0\|_2, \quad \forall\|\boldsymbol{\Delta}_0\|_2 \leq 1$$

with probability at least $1 - c_1\exp(-c_2 n_k), k \in \mathcal{S} \cup \{0\}$. For simplicity, we use $\alpha_k$ and $\tau_k$ for $\alpha_{1k}$ and $\tau_{1k}$, respectively.

By the mean-geometric mean inequality, we further have

$$\tau_k\sqrt{\frac{\log p}{n_k}}\|\boldsymbol{\delta}^{(k)}+\boldsymbol{\Delta}_0\|_1\|\boldsymbol{\delta}^{(k)}+\boldsymbol{\Delta}_0\|_2 \leq \frac{\alpha_k}{2}\|\boldsymbol{\delta}^{(k)}+\boldsymbol{\Delta}_0\|_2^2 + \frac{\tau_k^2}{2\alpha_k}\frac{\log p}{n_k}\|\boldsymbol{\delta}^{(k)}+\boldsymbol{\Delta}_0\|_1^2$$

$$\tau_0\sqrt{\frac{\log p}{n_0}}\|\boldsymbol{\Delta}_0\|_1\|\boldsymbol{\Delta}_0\|_2 \leq \frac{\alpha_0}{2}\|\boldsymbol{\Delta}_0\|_2^2 + \frac{\tau_0^2}{2\alpha_0}\frac{\log p}{n_0}\|\boldsymbol{\Delta}_0\|_1^2.$$

Therefore,

$$\mathcal{E}_n(\boldsymbol{\Delta}) \geq \sum_{k\in\mathcal{S}}\frac{n_k}{n}\left(\frac{\alpha_k}{2}\|\boldsymbol{\delta}^{(k)}+\boldsymbol{\Delta}_0\|_2^2 - \frac{\tau_k^2}{2\alpha_k}\frac{\log p}{n_k}\|\boldsymbol{\delta}^{(k)}+\boldsymbol{\Delta}_0\|_1^2\right) + \frac{n_0}{n}\left(\frac{\alpha_0}{2}\|\boldsymbol{\Delta}_0\|_2^2 - \frac{\tau_0^2}{2\alpha_0}\frac{\log p}{n_0}\|\boldsymbol{\Delta}_0\|_1^2\right)$$

$$= \sum_{k\in\mathcal{S}}\left(\frac{\alpha'}{2}\|\boldsymbol{\delta}^{(k)}+\boldsymbol{\Delta}_0\|_2^2 - \frac{\tau_k^2}{2\alpha_k}\frac{\log p}{N}\|\boldsymbol{\delta}^{(k)}+\boldsymbol{\Delta}_0\|_1^2\right) + \left(\frac{\alpha_0'}{2}\|\boldsymbol{\Delta}_0\|_2^2 - \frac{\tau_0^2}{2\alpha_0}\frac{\log p}{N}\|\boldsymbol{\Delta}_0\|_1^2\right),$$

where $\alpha' = n_k\alpha_k/n$ and $\alpha_0' = n_0\alpha_0/n$.

Replacing $\boldsymbol{\Delta}$ by $\hat{\boldsymbol{\Delta}} = \hat{\boldsymbol{\beta}} - \boldsymbol{\beta}$, we have the following results

$$\frac{\alpha'}{2}\|\hat{\boldsymbol{\beta}}\|_2^2 \leq \frac{\alpha'}{2}\sum_{k\in\mathcal{S}}\|\hat{\boldsymbol{\beta}}^{(k)}-\boldsymbol{\beta}^{(k)}\|_2^2 + \frac{\alpha'}{2}(|\mathcal{S}|+1)\|\hat{\boldsymbol{\beta}}^{(0)}-\boldsymbol{\beta}^{(0)}\|_2^2$$

$$= \frac{\alpha'}{2}\sum_{k\in\mathcal{S}}\|\hat{\boldsymbol{\Delta}}_k+\hat{\boldsymbol{\Delta}}_0\|_2^2 + \frac{\alpha'}{2}(|\mathcal{S}|+1)\|\hat{\boldsymbol{\Delta}}_0\|_2^2 \tag{12}$$

by taking $\alpha_0' = \alpha'(|\mathcal{S}|+1)$. With the definitions of $\alpha'$ and $\alpha_0'$, we choose $\alpha_0 = n_k\alpha_k(|\mathcal{S}|+1)/n_0$.

Denote $\tau_k = \tau$ and $\alpha_k = \alpha$ for all $k \in \mathcal{S}$,

$$\sum_{k\in\mathcal{S}}\frac{\tau_k^2}{2\alpha_k}\frac{\log p}{N}\|\hat{\boldsymbol{\Delta}}_k+\hat{\boldsymbol{\Delta}}_0\|_1^2 \leq \frac{\tau^2}{2\alpha}\frac{\log p}{N}\sum_{k\in\mathcal{S}}\left(\|\hat{\boldsymbol{\Delta}}_k\|_1^2 + \|\hat{\boldsymbol{\Delta}}_0\|\right)$$

$$= \frac{\tau^2}{2\alpha}\frac{\log p}{N}\|\hat{\boldsymbol{\Delta}}\|_1^2 + \frac{\tau^2}{2\alpha}\frac{\log p}{N}(|\mathcal{S}|-1)\|\hat{\boldsymbol{\Delta}}_0\|_1^2$$

$$\leq \frac{\tau^2}{2\alpha}\frac{\log p}{N}\|\hat{\boldsymbol{\Delta}}\|_1^2 + \frac{\tau^2}{2\alpha}\frac{\log p}{N}|\mathcal{S}|\|\hat{\boldsymbol{\Delta}}\|_1^2 \tag{13}$$

$$= \frac{\tau_0^2}{2\alpha_0}\frac{\log p}{N}\|\hat{\boldsymbol{\Delta}}\|_1^2$$

by taking $\tau_0 = \sqrt{|\mathcal{S}|+1}\tau$. Finally,

$$\frac{\tau_0^2}{2\alpha_0}\frac{\log p}{N}\|\hat{\boldsymbol{\Delta}}_0\|_1^2 \leq \frac{\tau_0^2}{2\alpha_0}\frac{\log p}{N}\|\hat{\boldsymbol{\Delta}}\|_1^2. \tag{14}$$

Adding Equations (12), (13), and (14), we have

$$\mathcal{E}_n(\hat{\boldsymbol{\Delta}}) \geq \frac{\alpha'}{2}\|\hat{\boldsymbol{\Delta}}\|_2^2 - \frac{\tau_0^2}{\alpha_0}\frac{\log p}{N}\|\hat{\boldsymbol{\Delta}}\|_1^2$$

$$= \alpha_1\|\hat{\boldsymbol{\Delta}}\|_2^2 - \tau_1\frac{\log p}{N}\|\hat{\boldsymbol{\Delta}}\|_1^2,$$

where we take $\alpha_1 = \alpha'/2$ and $\tau_1 = \tau_0^2/\alpha_0$. This establishes (11a). For (11b), by assuming $n \geq 4R^2\tau_1^2\log p$, then it holds according to Lemma 8 of Loh and Wainwright (2015). $\qquad\square$

**Lemma 2** (Lemmas 4(b) and 5 of Loh and Wainwright (2015))**.** *With the regularization function $P_\lambda$ satisfying the conditions (i)–(v),*

(a) *For any $\mathbf{w}$, we have $\lambda L \|\mathbf{w}\|_1 \leq P_\lambda(\mathbf{w}) + \tau/2 \|\mathbf{w}\|_2^2$*

(b) *Let $\mathcal{I}$ be the index set of the $s^*$ largest elements of $\mathbf{w}$ in magnitude. Suppose $c > 0$ is such that $cP_\lambda(\mathbf{w}_\mathbb{1}) - P_\lambda(\mathbf{w}_{\mathbb{1}^c}) \geq 0$, then*

$$cP_\lambda(\mathbf{w}_\mathbb{1}) - P_\lambda(\mathbf{w}_{\mathbb{1}^c}) \leq \lambda L \left( c\|\mathbf{w}_\mathbb{1}\|_1 - \|\mathbf{w}_{\mathbb{1}^c}\|_1 \right).$$

**Lemma 3.** *Let $I_s = I(\ell(\hat{\boldsymbol{\beta}}^{(0k)}, \mathcal{D}_{[t]}^{(0)}) \leq \ell(\hat{\boldsymbol{\beta}}^{(0)}, \mathcal{D}_{[t]}^{(0)}))$, then*

$$P\left(\sum_{s=1}^{S} I_s < \frac{S+1}{2}\right) \leq \frac{S+1}{2} p_\epsilon,$$

*where $p_\epsilon$ is the upper bound of $P(I_s = 0)$.*

*Proof.*

$$P\left(\sum_{s=1}^{S} I_s < \frac{S+1}{2}\right)$$

$$= P\left(\sum_{s=1}^{S} I_s < \frac{S+1}{2}, I_S = 1\right) + P\left(\sum_{s=1}^{S} I_s < \frac{S+1}{2}, I_S = 0\right)$$

$$= P\left(\sum_{s=1}^{S-1} I_s < \frac{S-1}{2}\right) + P\left(I_S = 0\right)$$

$$= P\left(\sum_{s=1}^{S+1} I_s < 1\right) + \sum_{s=\frac{S+3}{2}}^{S} P\left(I_s = 0\right)$$

$$= P\left(\bigcap_{s=1}^{\frac{S+1}{2}} I_s = 0\right) + \sum_{s=\frac{S+3}{2}}^{S} P\left(I_s = 0\right)$$

$$\leq P\left(I_{\frac{S+1}{2}} = 0\right) + \sum_{s=\frac{S+3}{2}}^{S} P\left(I_s = 0\right)$$

$$\leq \frac{S+1}{2} p_\epsilon.$$

$\square$

# F  Technical Details for Inference

## F.1  Assumptions

To develop the theoretical results for Algorithm 4, we further suppose some additional assumptions. Throughout the paper, the matrix with subscript $\boldsymbol{\beta}$ means the weighted matrix by $\boldsymbol{\beta}$. Matrix with superscript $(k)$ is for the $k$-th data, $k \in \mathcal{S} \cup \{0\}$. Matrix with subscript $j$ denotes the $j$-th column. For example, $\boldsymbol{\Sigma}_{\boldsymbol{\beta}_0, j, j}^{(k)}$ denotes the $(j, j)$-th element of the weighted covariance matrix for the $k$-th data, with weight $\boldsymbol{\beta}_0$. We also define

$$\boldsymbol{\gamma}_j^{(k)} = \arg\min_{\boldsymbol{\gamma}} \mathrm{E}\left\{\psi''(\boldsymbol{\beta}^T \mathbf{x}^{(k)}) \cdot [\mathbf{x}_j^{(k)} - (x_{-j}^{(k)})^T \boldsymbol{\gamma}]^2\right\} = (\boldsymbol{\Sigma}_{\boldsymbol{\beta}, -j, -j}^{(k)})^{-1} \boldsymbol{\Sigma}_{\boldsymbol{\beta}, -j, j}^{(k)}. \tag{15}$$

**Assumption 8.** *We assume the following conditions hold*

a) $\sup_{k \in \mathcal{S}, j=1:p} \left| (\boldsymbol{\Sigma}_{\boldsymbol{\beta}^{(0)}, -j, -j}^{(0)})^{-1} \boldsymbol{\Sigma}_{\boldsymbol{\beta}^{(0)}, -j, j}^{(0)} - (\boldsymbol{\Sigma}_{\boldsymbol{\beta}^{(0)}, -j, -j}^{(k)})^{-1} \boldsymbol{\Sigma}_{\boldsymbol{\beta}^{(0)}, -j, j}^{(k)} \right|_1 \leq h_1.$

b) $\sup_{k\in\mathcal{S},j=1:p}\left\{\left|\boldsymbol{\Sigma}_{\boldsymbol{\beta}^{(0)},j,j}^{(k)} - \boldsymbol{\Sigma}_{\boldsymbol{\beta}^{(0)},j,j}^{(0)}\right| \vee \left|(\boldsymbol{\Sigma}_{\boldsymbol{\beta}^{(0)},j,-j}^{(k)} - \boldsymbol{\Sigma}_{\boldsymbol{\beta}^{(0)},j,-j}^{(0)})\boldsymbol{\gamma}_{0j}\right|\right\} \leq h_2.$

c) $\sup_{j=1:p}\left|\Sigma_{j,j}^{(0)} - \boldsymbol{\Sigma}_{-j,j}^{(0)}\boldsymbol{\gamma}_j - \left(\Sigma_{\boldsymbol{\beta}^{(0)},j,j}^{(0)} - \boldsymbol{\Sigma}_{\boldsymbol{\beta}^{(0)},-j,j}^{(k)}\boldsymbol{\gamma}_j\right)\right| \leq h_3.$

**Remark 6.** *Condition a) implies $\sup_{k\in\mathcal{S},j=1:p}\|\boldsymbol{\gamma}_j^{(k)} - \boldsymbol{\gamma}_j^{(0)}\|_1 \lesssim h_1$, i.e., the maximum distance between parameters of the node-wise regression from the $k$-th source data and the target data is bounded by $h_1$ (Tian and Feng, 2023). Condition b) bounds the variance difference between the $k$-th source data and the target data by $h_2$. Condition c) constrains $|\tau_{0j}^2 - \tau_j^2| \leq h_3$, which is for the logistic regression only. These three conditions quantify the impacts of source data in node-wise regression for Theorem 3.*

**Assumption 9.** *We assume the following conditions hold*

(i) $\sup_{k\in\mathcal{S}\cup\{0\}} \|\mathbf{x}^{(k)}\|_\infty \leq U < \infty$, $\sup_{k\in\mathcal{S}\cup\{0\}} |\mathbf{x}^{(k)\top}\boldsymbol{\beta}^{(k)}| \leq U' < \infty$ *a.s.*

(ii) $\inf_{k\in\mathcal{S}\cup\{0\}} \lambda_{\min}(\boldsymbol{\Sigma}_{\boldsymbol{\beta}^{(k)},k}) \geq \bar{U} > 0$.

(iii) $\sup_{k\in\mathcal{S}\cup\{0\}} \sup_{|z|\leq\bar{U}} \psi'''(\mathbf{x}^{(k)\top}\boldsymbol{\beta}^{(k)} + z) \leq M_\psi < \infty$ *a.s.*

(iv) *There exist $C > 0$ and $C' > 0$ such that $\|\boldsymbol{\gamma}_j\|_2 \leq C$ and $\|\boldsymbol{\gamma}_{0j}\|_2 \leq C'$, for $j = 1, \cdots, p$.*

(v) *$\sup_j s_j \lesssim s$ where $s_j = \|\boldsymbol{\gamma}_j\|_0 \vee \|\boldsymbol{\gamma}_{0j}\|_0$ and $1/\tau_j^2 = O(1)$.*

(vi)

$$\left[R_1 + h_1^{\frac{1}{2}}\left(\frac{\log p}{N}\right)^{\frac{1}{4}} + R_1^{\frac{1}{2}}h_1^{\frac{1}{2}} + h_2 + h_3 + n_0^{-\frac{1}{2}}\right]\left[s\left(\frac{\log p}{N}\right)^{\frac{1}{2}} + \left(\frac{\log p}{N}\right)^{\frac{1}{4}}(sh)^{\frac{1}{2}}\right] = o(n_0^{-\frac{1}{2}}),$$

$$\left[1 + s^{\frac{1}{2}}R_1 + (sh_1R_1)^{\frac{1}{2}} + h_1 + s^{\frac{1}{2}}(h_2 + h_3)\right] = O(1),$$

$$\|\boldsymbol{\Sigma}_{\boldsymbol{\beta}^{(0)}}^{(0)}\boldsymbol{\Theta}_j\|_1 \leq C$$

*for a positive constant $C$.*

**Remark 7.** *Items (i)(ii) are the usual assumptions needed for the high-dimensional GLMs. (ii) makes sure that $\boldsymbol{\Theta}_{jj}^{-1}$ stays away from zero. (iv) avoids the diverging $\boldsymbol{\gamma}_j$ for linear model and diverging $\boldsymbol{\gamma}_{0j}$ for logistic model. $\sup_j s_j \lesssim s$ in (v) is for technical simplicity, the same assumption in Tian and Feng (2023). Item (vi) guarantees the consistency of $\hat{\mathbf{b}}$, which is part of the proof of the asymptotic normality in Theorem 3.*

### F.2 Proof of Proposition 1

In this section, we prove Proposition 1, which states that even $\hat{\boldsymbol{\beta}}^{(0)}$ is obtained from a co-learning scheme, its desparsified form takes the same form as the single-task training scheme, i.e., the form in Van de Geer et al. (2014).

*Proof.* Before we start the proof, we reformulate all the datasets in the following way. Without loss of generality, we assume $\mathcal{S} = \{1, \cdots, K\}$ in the proof of Proposition 1.

$$\mathbf{X}^* = \begin{bmatrix} \mathbf{X}^{(1)} & \mathbf{0} & \cdots & \cdots & \mathbf{0} & \mathbf{X}^{(1)} \\ \mathbf{0} & \mathbf{X}^{(2)} & \mathbf{0} & \cdots & \mathbf{0} & \mathbf{X}^{(2)} \\ \vdots & \vdots & \vdots & \vdots & \vdots & \vdots \\ \mathbf{0} & \cdots & \cdots & \cdots & \mathbf{X}^{(K)} & \mathbf{X}^{(K)} \\ \mathbf{0} & \cdots & \cdots & \cdots & \mathbf{0} & \mathbf{X}^{(0)} \end{bmatrix} \in \mathbb{R}^{N\times p^*},$$

where $p^* = p(1 + |\mathcal{S}|)$. Besides, recall that for $k \in \{0\} \cup \mathcal{S}$ and $\hat{\boldsymbol{\beta}}^{(k)}$, we define the weighted version of $\mathbf{X}^{(k)}$ as

$$\mathbf{X}_{\hat{\boldsymbol{\beta}}^{(k)}}^{(k)} = \sqrt{\text{diag}\left(\psi''(\mathbf{x}_i^{(k)\top}\hat{\boldsymbol{\beta}}^{(k)}), \cdots, \psi''(\mathbf{x}_{n_k}^{(k)\top}\hat{\boldsymbol{\beta}}^{(k)})\right)}\mathbf{X}^{(k)},$$

and we can define the weighted version of $\mathbf{X}^*$ as

$$\mathbf{X}_{\hat{\boldsymbol{\beta}}}^* = \begin{bmatrix} \mathbf{X}_{\hat{\boldsymbol{\beta}}^{(1)}}^{(1)} & \mathbf{0} & \cdots & \cdots & \mathbf{0} & \mathbf{X}_{\hat{\boldsymbol{\beta}}^{(1)}}^{(1)} \\ \mathbf{0} & \mathbf{X}_{\hat{\boldsymbol{\beta}}^{(2)}}^{(2)} & \mathbf{0} & \cdots & \mathbf{0} & \mathbf{X}_{\hat{\boldsymbol{\beta}}^{(2)}}^{(2)} \\ \vdots & \vdots & \vdots & \vdots & \vdots & \vdots \\ \mathbf{0} & \cdots & \cdots & \cdots & \mathbf{X}_{\hat{\boldsymbol{\beta}}^{(K)}}^{(K)} & \mathbf{X}_{\hat{\boldsymbol{\beta}}^{(K)}}^{(K)} \\ \mathbf{0} & \cdots & \cdots & \cdots & \mathbf{0} & \mathbf{X}_{\hat{\boldsymbol{\beta}}^{(0)}}^{(0)} \end{bmatrix}.$$

Therefore,

$$\hat{\boldsymbol{\Sigma}}_{\hat{\boldsymbol{\beta}}}^* = \frac{1}{n}\mathbf{X}_{\hat{\boldsymbol{\beta}}}^{*\top}\mathbf{X}_{\hat{\boldsymbol{\beta}}}^* = \frac{1}{N}\begin{bmatrix} n_1\hat{\boldsymbol{\Sigma}}_{\hat{\boldsymbol{\beta}}^{(1)}}^{(1)} & \mathbf{0} & \cdots & \cdots & \mathbf{0} & n_1\hat{\boldsymbol{\Sigma}}_{\hat{\boldsymbol{\beta}}^{(1)}}^{(1)} \\ \mathbf{0} & n_2\hat{\boldsymbol{\Sigma}}_{\hat{\boldsymbol{\beta}}^{(2)}}^{(2)} & \mathbf{0} & \cdots & \mathbf{0} & n_2\hat{\boldsymbol{\Sigma}}_{\hat{\boldsymbol{\beta}}^{(2)}}^{(2)} \\ \vdots & \vdots & \vdots & \vdots & \vdots & \vdots \\ \mathbf{0} & \cdots & \cdots & \cdots & n_K\hat{\boldsymbol{\Sigma}}_{\hat{\boldsymbol{\beta}}^{(K)}}^{(K)} & n_K\hat{\boldsymbol{\Sigma}}_{\hat{\boldsymbol{\beta}}^{(K)}}^{(K)} \\ n_1\hat{\boldsymbol{\Sigma}}_{\hat{\boldsymbol{\beta}}^{(1)}}^{(1)} & n_2\hat{\boldsymbol{\Sigma}}_{\hat{\boldsymbol{\beta}}^{(2)}}^{(2)} & \cdots & \cdots & n_K\hat{\boldsymbol{\Sigma}}_{\hat{\boldsymbol{\beta}}^{(K)}}^{(K)} & \sum_{k\in\mathcal{S}\cup\{0\}} n_k\hat{\boldsymbol{\Sigma}}_{\hat{\boldsymbol{\beta}}^{(0)}}^{(0)} \end{bmatrix} := \begin{bmatrix} A & B \\ C & D \end{bmatrix},$$

where

$$A = \begin{bmatrix} n_1\hat{\boldsymbol{\Sigma}}_{\hat{\boldsymbol{\beta}}^{(1)}}^{(1)} & \mathbf{0} & \cdots & \cdots & \mathbf{0} \\ \mathbf{0} & n_2\hat{\boldsymbol{\Sigma}}_{\hat{\boldsymbol{\beta}}^{(2)}}^{(2)} & \mathbf{0} & \cdots & \mathbf{0} \\ \vdots & \vdots & \vdots & \vdots & \vdots \\ \mathbf{0} & \cdots & \cdots & \cdots & n_K\hat{\boldsymbol{\Sigma}}_{\hat{\boldsymbol{\beta}}^{(K)}}^{(K)} \end{bmatrix}, \quad B = \begin{bmatrix} n_1\hat{\boldsymbol{\Sigma}}_{\hat{\boldsymbol{\beta}}^{(1)}}^{(1)} \\ \vdots \\ n_1\hat{\boldsymbol{\Sigma}}_{\hat{\boldsymbol{\beta}}^{(K)}}^{(K)} \end{bmatrix}.$$

Besides, $C = B^\top$ and $D = \left[\sum_{k\in\mathcal{S}\cup\{0\}} n_k\hat{\boldsymbol{\Sigma}}_{\hat{\boldsymbol{\beta}}^{(0)}}^{(0)}\right]$.

We also introduce the coordinate-wise representation of a function, i.e., for a vector $\mathbf{x} \in \mathbb{R}^p$ and a function $\phi : \mathbb{R} \to \mathbb{R}$, $\phi \circ (\mathbf{x}) = (\phi(x_1), \cdots, \phi(x_p))^\top$.

With these notations, we can now rewrite that loss function $\mathcal{L}_\mathcal{D}\left(\boldsymbol{\beta}^{(0)}, \{\boldsymbol{\delta}^{(k)}\}_{k\in\mathcal{S}}\right) := \mathcal{L}_\mathcal{D}(\boldsymbol{\beta})$ without regularization terms as

$$-(\mathbf{Y}^{(1)\top}, \cdots, \mathbf{Y}^{(K)\top}, \mathbf{Y}^{(0)\top})X^* \begin{bmatrix} \boldsymbol{\beta}^{(1)} - \boldsymbol{\beta}^{(0)} \\ \vdots \\ \boldsymbol{\beta}^{(K)} - \boldsymbol{\beta}^{(0)} \\ \boldsymbol{\beta}^{(0)} \end{bmatrix} + (\mathbf{1}^\top, \cdots, \mathbf{1}^\top)\phi \circ \left(X^* \begin{bmatrix} \boldsymbol{\beta}^{(1)} - \boldsymbol{\beta}^{(0)} \\ \vdots \\ \boldsymbol{\beta}^{(K)} - \boldsymbol{\beta}^{(0)} \\ \boldsymbol{\beta}^{(0)} \end{bmatrix}\right),$$

which is in a single-task loss setting.

With this reformulation, we can apply the desparsified form of GLMs (Equation (18) in Van de Geer et al. (2014)) directly, which is

$$\hat{\mathbf{b}} = \hat{\boldsymbol{\beta}} - \left(\hat{\boldsymbol{\Sigma}}_{\hat{\boldsymbol{\beta}}}^*\right)^{-1} \frac{\partial}{\partial\boldsymbol{\beta}}\mathcal{L}_\mathcal{D}\left(\boldsymbol{\beta}^{(0)}, \{\boldsymbol{\delta}^{(k)}\}_{k\in\mathcal{S}}\right),$$

and the desparsified form of $\hat{\boldsymbol{\beta}}^{(0)}$ is by taking the last $p$ elements of $\hat{\mathbf{b}}$.

The proof of Proposition 1 is then completed by using the following equation

$$\begin{bmatrix} A & B \\ C & D \end{bmatrix}^{-1} = \begin{bmatrix} A^{-1} + A^{-1}B(D - CA^{-1}B)^{-1}CA^{-1} & -A^{-1}B(D - CA^{-1}B)^{-1} \\ -(D - CA^{-1}B)^{-1}CA^{-1} & (D - CA^{-1}B)^{-1} \end{bmatrix},$$

and

$$\frac{\partial}{\partial\boldsymbol{\beta}}\mathcal{L}_\mathcal{D}\left(\boldsymbol{\beta}^{(0)}, \{\boldsymbol{\delta}^{(k)}\}_{k\in\mathcal{S}}\right) = X^{*\top}\left\{\begin{bmatrix} \mathbf{Y}^{(1)} \\ \vdots \\ \mathbf{Y}^{(K)} \\ \mathbf{Y}^{(0)} \end{bmatrix} - \phi' \circ \left(X^* \begin{bmatrix} \boldsymbol{\beta}^{(1)} - \boldsymbol{\beta}^{(0)} \\ \vdots \\ \boldsymbol{\beta}^{(K)} - \boldsymbol{\beta}^{(0)} \\ \boldsymbol{\beta}^{(0)} \end{bmatrix}\right)\right\},$$

and then taking the last $p$ elements. $\qquad\square$

## F.3 Proof of Theorem 3

The formal version of Theorem 3 is stated as follows.

**Theorem 4.** *Denote $R_1 = (s \log p/N)^{\frac{1}{2}} + (\log p/N)^{\frac{1}{4}} h^{\frac{1}{2}}$. Assume Assumption 1-3, 7 and 9 hold and $R_1^2 = o(n_0^{-\frac{1}{2}})$, then for $j = 1, \cdots, p$, the following hold with probabilities at least $1 - K n_0^{-1}$,*

$$\|\hat{\boldsymbol{\Theta}}_j - \boldsymbol{\Theta}_j\|_2 \lesssim R_1 + h_1^{\frac{1}{2}} \left(\frac{\log p}{N}\right)^{\frac{1}{4}} + h_1^{\frac{1}{2}} R_1^{\frac{1}{2}} + h_2 + h_3,$$

$$|(\hat{\boldsymbol{\Sigma}}_{\hat{\mathbf{b}}^{(0)}})_{jj} - \boldsymbol{\Theta}_{jj}| \lesssim h_1 + s^{\frac{1}{2}} \left[R_1 + h_1^{\frac{1}{2}} \left(\frac{\log p}{N}\right)^{\frac{1}{4}} + R_1^{\frac{1}{2}} h_1^{\frac{1}{2}} + h_2 + h_3\right].$$

*Moreover, assume $|(\hat{\boldsymbol{\Sigma}}_{\hat{\mathbf{b}}^{(0)}})_{jj} - \boldsymbol{\Theta}_{jj}| = o_P(1)$, then*

$$\frac{\sqrt{n_0}(\hat{b}_j - \beta_j^{(0)})}{\sqrt{(\hat{\boldsymbol{\Sigma}}_{\hat{\mathbf{b}}^{(0)}})_{jj}}} \xrightarrow{d} \mathcal{N}(0, 1).$$

*Proof.* First, note that $\hat{\tau}_{0j}^2 \geq 4\hat{\tau}_j^2$ for linear and logistic regressions due to the property of the weight matrix. We bound

$$\begin{aligned}
\|\hat{\boldsymbol{\Theta}}_j - \boldsymbol{\Theta}_j\|_1 &= \|\hat{C}_j/\hat{\tau}_{0j}^2 - C_j/\tau_j^2\|_1 \\
&\leq \|\hat{\boldsymbol{\gamma}}_j - \boldsymbol{\gamma}_j\|_1/\hat{\tau}_{0j}^2 + \|\boldsymbol{\gamma}_j\|_1(\hat{\tau}_{0j}^{-2} - \tau_j^{-2}) \\
&\leq \|\hat{\boldsymbol{\gamma}}_j - \boldsymbol{\gamma}_j\|_1/(4\hat{\tau}_j^2) + \|\boldsymbol{\gamma}_j\|_1(\hat{\tau}_{0j}^{-2} - \tau_j^{-2}) \\
&\lesssim s^{\frac{1}{2}} R_1 + (s h_1 R_1)^{\frac{1}{2}} + h_1 + s^{\frac{1}{2}}(h_2 + h_3)
\end{aligned}$$

since $\hat{\tau}_j^2$ is a consistent estimator of $\tau_j^2$, $1/\tau_j^2 = O(1)$, and $\|\boldsymbol{\gamma}_j\|_1 = O(\sqrt{s})$. Next, we bound

$$\begin{aligned}
\|\hat{\boldsymbol{\Theta}}_j - \boldsymbol{\Theta}_j\|_2 &\leq \|\hat{\boldsymbol{\gamma}}_j - \boldsymbol{\gamma}_j\|_2/\hat{\tau}_{0j}^2 + \|\boldsymbol{\gamma}_j\|_2(\hat{\tau}_{0j}^{-2} - \tau_j^{-2}) \\
&\lesssim R_1 + h_1^{\frac{1}{2}} \left(\frac{\log p}{N}\right)^{\frac{1}{4}} + h_1^{\frac{1}{2}} R_1^{\frac{1}{2}} + h_2 + h_3.
\end{aligned}$$

Next, we bound the estimation error of the precision matrix

$$|\hat{\boldsymbol{\Theta}}_j^{\top} \hat{\boldsymbol{\Sigma}}_{\hat{\boldsymbol{\beta}}^{(0)}} \hat{\boldsymbol{\Theta}}_j - \boldsymbol{\Theta}_{jj}| \leq \left|\hat{\boldsymbol{\Theta}}_j^{\top} \left(\hat{\boldsymbol{\Sigma}}_{\hat{\boldsymbol{\beta}}^{(0)}} \hat{\boldsymbol{\Theta}}_j - \boldsymbol{\Sigma}_{\boldsymbol{\beta}^{(0)}}^{(0)} \boldsymbol{\Theta}_j\right)\right| + \left|\left(\hat{\boldsymbol{\Theta}}_j - \boldsymbol{\Theta}_j\right) \boldsymbol{\Sigma}_{\boldsymbol{\beta}^{(0)}}^{(0)} \boldsymbol{\Theta}_j\right|.$$

First, notice that

$$\left| \hat{\boldsymbol{\Sigma}}_{\hat{\boldsymbol{\beta}}^{(0)}} \hat{\boldsymbol{\Theta}}_j - \boldsymbol{\Sigma}^{(0)}_{\boldsymbol{\beta}^{(0)}} \boldsymbol{\Theta}_j \right|_\infty$$

$$\lesssim \underbrace{\left\| \frac{1}{n} \sum_{i,k} \mathbf{x}_{ki}^\top \left[ \psi'' \left( \mathbf{x}_i^{(k)\top} \hat{\boldsymbol{\beta}}_0 \right) - \psi'' \left( \mathbf{x}_i^{(k)\top} \boldsymbol{\beta}^{(0)} \right) \right] \mathbf{x}_{ki}^\top \frac{\hat{\boldsymbol{\gamma}}_j}{\hat{\tau}_{0j}^2} \right\|_\infty}_{(4)}$$

$$+ \underbrace{\left\| \frac{1}{n} \sum_{i,k} \mathbf{x}_i^{(k)} \psi'' \left( \mathbf{x}_i^{(k)\top} \boldsymbol{\beta}^{(0)} \right) \mathbf{x}_i^{(k)\top} \left[ \hat{\boldsymbol{\gamma}}_j - \boldsymbol{\gamma}_j \right] \hat{\tau}_{0j}^{-2} \right\|_\infty}_{(5)} + \underbrace{\left\| \frac{1}{n} \sum_{i,k} \mathbf{x}_i^{(k)} \psi'' \left( \mathbf{x}_i^{(k)\top} \boldsymbol{\beta}^{(0)} \right) \cdot \mathbf{x}_i^{(k)\top} \hat{\boldsymbol{\gamma}}_j \left( \hat{\tau}_{0j}^2 - \tau_j^2 \right) \right\|_\infty}_{(6)}$$

$$+ \underbrace{\frac{1}{n} \left\| \sum_{i,k} \left[ \mathbf{x}_i^{(k)} \psi'' \left( \mathbf{x}_i^{(k)\top} \boldsymbol{\beta}^{(0)} \right) \mathbf{x}_i^{(k)\top} \frac{\boldsymbol{\gamma}_j}{\tau_j^2} \right] - \sum_{k \in \{0\} \cup \mathcal{S}_h} n_k E \left[ \mathbf{x}_i^{(k)\top} \psi'' \left( \mathbf{x}_i^{(k)\top} \boldsymbol{\beta}^{(0)} \right) \mathbf{x}_i^{(k)\top} \frac{\boldsymbol{\gamma}_j}{\tau_j^2} \right] \right\|_\infty}_{(7)}$$

$$+ \underbrace{\left\| \sum_{k \in \{0\} \cup \mathcal{S}_h} \frac{n_k}{n} E \left[ \mathbf{x}_i^{(k)\top} \psi'' \left( \mathbf{x}_i^{(k)\top} \boldsymbol{\beta}^{(0)} \right) \mathbf{x}_i^{(k)\top} \frac{\boldsymbol{\gamma}_j}{\tau_j^2} - \mathbf{x}_i^{(0)\top} \psi'' \left( \mathbf{x}_i^{(0)\top} \boldsymbol{\beta}^{(0)} \right) \mathbf{x}_i^{(0)\top} \frac{\boldsymbol{\gamma}_j}{\tau_j^2} \right] \right\|_\infty}_{(8)}.$$

$$(16)$$

To bound $|\hat{\boldsymbol{\Theta}}_j^\top \hat{\boldsymbol{\Sigma}}_{\hat{\boldsymbol{\beta}}^{(0)}} \hat{\boldsymbol{\Theta}}_j - \boldsymbol{\Theta}_{jj}|$, we notice

$$(4) \lesssim \| \hat{\boldsymbol{\beta}}^{(0)} - \boldsymbol{\beta}^{(0)} \|_2 \lesssim R_1.$$

$$(5) \lesssim \| \hat{\boldsymbol{\gamma}}_j - \boldsymbol{\gamma}_j \|_2 + | \hat{\tau}_{0j}^{-2} - \tau_j^{-2} | \lesssim R_1 + h_1^{\frac{1}{2}} \left( \frac{\log p}{N} \right)^{\frac{1}{4}} + R_1^{\frac{1}{2}} h_1^{\frac{1}{2}} + h_2 + h_3.$$

$$(6) \lesssim | \hat{\tau}_{0j}^{-2} - \tau_j^{-2} | \lesssim R_1 + h_1^{\frac{1}{2}} \left( \frac{\log p}{N} \right)^{\frac{1}{4}} + h_1^{\frac{1}{2}} R_1^{\frac{1}{2}} + h_2 + h_3.$$

$$(8) \lesssim \sup_{k \in \mathcal{S}} \| \boldsymbol{\Sigma}^{(k)}_{\boldsymbol{\beta}^{(0)}} - \boldsymbol{\Sigma}^{(0)}_{\boldsymbol{\beta}^{(0)}} \|_\infty \left| \frac{\boldsymbol{\gamma}_j}{\tau_j^2} \right|_1 \lesssim h_2.$$

Therefore,

$$\left| \hat{\boldsymbol{\Theta}}_j^\top \left( \hat{\boldsymbol{\Sigma}}_{\hat{\boldsymbol{\beta}}^{(0)}} \hat{\boldsymbol{\Theta}}_j - \boldsymbol{\Sigma}^{(0)}_{\boldsymbol{\beta}^{(0)}} \boldsymbol{\Theta}_j \right) \right| \lesssim s^{\frac{1}{2}} \left[ R_1 + h_1^{\frac{1}{2}} \left( \frac{\log p}{N} \right)^{\frac{1}{4}} + R_1^{\frac{1}{2}} h_1^{\frac{1}{2}} + h_2 + h_3 \right]$$

since $\| \hat{\boldsymbol{\Theta}}_j \|_1 \lesssim s^{\frac{1}{2}}$. We bound the second term

$$\left| \left( \hat{\boldsymbol{\Theta}}_j - \boldsymbol{\Theta}_j \right) \boldsymbol{\Sigma}^{(0)}_{\boldsymbol{\beta}^{(0)}} \boldsymbol{\Theta}_j \right| \lesssim \| \hat{\boldsymbol{\Theta}}_j - \boldsymbol{\Theta}_j \|_1$$

$$\lesssim s^{\frac{1}{2}} R_1 + (s h_1 R_1)^{\frac{1}{2}} + h_1 + s^{\frac{1}{2}} (h_2 + h_3).$$

Finally, we have

$$| \hat{\boldsymbol{\Theta}}_j^\top \hat{\boldsymbol{\Sigma}}_{\hat{\boldsymbol{\beta}}^{(0)}} \hat{\boldsymbol{\Theta}}_j - \boldsymbol{\Theta}_{jj} | \lesssim h_1 + s^{\frac{1}{2}} \left[ R_1 + h_1^{\frac{1}{2}} \left( \frac{\log p}{N} \right)^{\frac{1}{4}} + R_1^{\frac{1}{2}} h_1^{\frac{1}{2}} + h_2 + h_3 \right].$$

To show the asymptotic normality, we can decompose $\hat{b}_j - \beta_j^{(0)}$ by

$$\hat{b}_j - \beta_j^{(0)} = \underbrace{\left[ \boldsymbol{e}_j - \hat{\boldsymbol{\Theta}}_j^\top \hat{\boldsymbol{\Sigma}}^{(0)}_{\boldsymbol{\beta}} \right]^\top (\boldsymbol{\beta}_0 - \hat{\boldsymbol{\beta}}_0)}_{(1)} + \underbrace{\frac{1}{n_0} \hat{\boldsymbol{\Theta}}_j^\top \mathbf{X}^{(0)\top} \left[ \mathbf{y}^{(0)} - \boldsymbol{\psi}' \left( \mathbf{X}^{(0)} \boldsymbol{\beta}^{(0)} \right) \right]}_{(2)}$$

$$+ \underbrace{\frac{1}{n_0} \hat{\boldsymbol{\Theta}}_j^\top \mathbf{X}^{(0)\top} \operatorname{diag} \left( \left\{ \psi'' \left( \tilde{u}_i^{(0)} \right) - \psi'' \left( \mathbf{x}_i^{(0)\top} \boldsymbol{\beta}^{(0)} \right) \right\}_{i=1}^{n_0} \right) \mathbf{X}^{(0)} (\boldsymbol{\beta}_0 - \hat{\boldsymbol{\beta}}_0)}_{(3)},$$

where $\tilde{u}_i^{(0)}$ falls on the line between $\mathbf{x}_i^{(0)\top}\boldsymbol{\beta}^{(0)}$ and $\mathbf{x}_i^{(0)\top}\hat{\boldsymbol{\beta}}^{(0)}$.

$$\begin{aligned}
(1) &\lesssim \left( \|\hat{\boldsymbol{\gamma}}_j - \boldsymbol{\gamma}_j\|_2 + |\hat{\tau}_{0j}^{-2} - \tau_j^{-2}| + n_0^{-\frac{1}{2}} \right) \|\hat{\boldsymbol{\beta}}^{(0)} - \boldsymbol{\beta}^{(0)}\|_1 \\
&\lesssim \left[ R_1 + h_1^{\frac{1}{2}} \left( \frac{\log p}{N} \right)^{\frac{1}{4}} + R_1^{\frac{1}{2}} h_1^{\frac{1}{2}} + h_2 + h_3 + n_0^{-\frac{1}{2}} \right] \\
&\quad \times \left[ s \left( \frac{\log p}{N} \right)^{\frac{1}{2}} + \left( \frac{\log p}{N} \right)^{\frac{1}{4}} (sh)^{\frac{1}{2}} \right] \\
&= o(n_0^{-\frac{1}{2}})
\end{aligned} \tag{17}$$

under Condition 6 (vi).

$$\sqrt{n_0} \cdot (2) \to \mathcal{N}(0, \boldsymbol{\Theta}_{jj}). \tag{18}$$

$$\begin{aligned}
(3) &\lesssim \left( 1 + \|\hat{\boldsymbol{\Theta}}_j - \boldsymbol{\Theta}_j\|_1 \right) \|\hat{\boldsymbol{\beta}}^{(0)} - \boldsymbol{\beta}^{(0)}\|_2^2 \\
&\lesssim \left[ 1 + s^{\frac{1}{2}} R_1 + (sh_1 R_1)^{\frac{1}{2}} + h_1 + s^{\frac{1}{2}}(h_2 + h_3) \right] R_1^2 \\
&= o(n_0^{-\frac{1}{2}})
\end{aligned} \tag{19}$$

under Condition 6 (vi). Combined with (17), (18), and (19), the result follows.

**Final result:** Denote $m_{23} = \left[ \left( \frac{s \log p}{n_0} \right)^{\frac{1}{2}} + h_3 \right] \vee h_2$. Combining the results of linear, Poisson, and logistic models, we have

$$\|\hat{\boldsymbol{\Theta}}_j - \boldsymbol{\Theta}_j\|_2 \lesssim R_1 + h_1^{\frac{1}{2}} \left( \frac{\log p}{N} \right)^{\frac{1}{4}} + h_1^{\frac{1}{2}} R_1^{\frac{1}{2}} + m_{23}.$$

$$|\hat{\boldsymbol{\Theta}}_j^\top \hat{\boldsymbol{\Sigma}}_{\hat{\boldsymbol{\beta}}^{(0)}} \hat{\boldsymbol{\Theta}}_j - \boldsymbol{\Theta}_{jj}| \lesssim h_1 + s^{\frac{1}{2}} \left[ R_1 + h_1^{\frac{1}{2}} \left( \frac{\log p}{N} \right)^{\frac{1}{4}} + R_1^{\frac{1}{2}} h_1^{\frac{1}{2}} + m_{23} \right].$$

$$\frac{\sqrt{n_0}(\hat{b}_j - \beta_j^{(0)})}{\sqrt{\hat{\boldsymbol{\Theta}}_j^\top \hat{\boldsymbol{\Sigma}}_{\hat{\boldsymbol{\beta}}^{(0)}} \hat{\boldsymbol{\Theta}}_j}} \xrightarrow{d} \mathcal{N}(0, 1).$$

$\square$

### F.4 Lemma

**Lemma 4.** *Assume Conditions 1-4 and C6 (i–iv). Let* $\lambda_j \asymp \left( \frac{\log p}{N} \right)^{\frac{1}{2}} + \left( \frac{\log p}{N} \right)^{\frac{1}{4}} \left( \frac{h}{s} \right)^{\frac{1}{2}}$ *for any* $j = 1, \cdots, p$. *Suppose* $h_1 \lesssim s^{-\frac{1}{2}}$, *then with probability at least* $1 - K n_0^{-1}$,

$$\|\hat{\boldsymbol{\gamma}}_j - \boldsymbol{\gamma}_j\|_2^2 \lesssim h_1 \left( \frac{\log p}{N} \right)^{\frac{1}{2}} + R_1^2 + R_1 h_1$$

$$\|\hat{\boldsymbol{\gamma}}_j - \boldsymbol{\gamma}_j\|_1 \lesssim s^{\frac{1}{2}} R_1 + s^{\frac{1}{4}} h_1^{\frac{1}{2}} R_1^{\frac{1}{2}} + h_1.$$

The proof is very similar to Lemma 4 of Tian and Feng (2023). Note that our $\hat{\boldsymbol{\gamma}}_j$ is denoted as $\hat{\boldsymbol{\gamma}}_j^{\mathcal{S}_h}$ in Trans-GLM and the details to bound it are the same. The result is determined by the bound of $\|\hat{\boldsymbol{\beta}}^{(0)} - \boldsymbol{\beta}^{(0)}\|_2$, i.e., $R_1$. We replace our bound of $\|\hat{\boldsymbol{\beta}}^{(0)} - \boldsymbol{\beta}^{(0)}\|_2$ to get the desired result.

**Lemma 5.** *Assume Assumption 9 (i–iv) and the conditions in Lemma 4. For the linear regression and logistic regression, if conditions a) – c) hold, then*

$$|\hat{\tau}_{0j}^2 - \tau_j^2| \vee |\hat{\tau}_{0j}^{-2} - \tau_j^{-2}| \lesssim R_1 + h_1^{\frac{1}{2}} \left( \frac{\log p}{N} \right)^{\frac{1}{4}} + h_1^{\frac{1}{2}} R_1^{\frac{1}{2}} + h_2 + h_3.$$

**Proof:**

Recall that $\tau_{0j}^2 = (\mathbf{X}_j^{(0)} - \mathbf{X}_{-j}^{(0)}\boldsymbol{\gamma}_j)^\top \mathbf{X}_j^{(0)}/n_0$ and $\hat{\tau}_{0j}^2 = (\mathbf{X}_j^{(0)} - \mathbf{X}_{-j}^{(0)}\hat{\boldsymbol{\gamma}}_j)^\top \mathbf{X}_j^{(0)}/n_0$. We decompose

$$|\hat{\tau}_{0j}^2 - \tau_j^2| \leq \underbrace{|\hat{\tau}_{0j}^2 - \tau_{0j}^2|}_{(1)} + \underbrace{|\tau_{0j}^2 - \tau_j^2|}_{(2)}.$$

For (1), we can use similar techniques of S.2.4.2 in Tian and Feng (2023). Note that (1) does not have the weight and uses the target data only. Therefore, we can show that

$$(1) \lesssim R_1 + h_1^{\frac{1}{2}}\left(\frac{\log p}{N}\right)^{\frac{1}{4}} + h_1^{\frac{1}{2}}R_1^{\frac{1}{2}} + h_2.$$

For (2), by the assumption,

$$
\begin{aligned}
|\tau_{0j}^2 - \tau_j^2| &= |(\mathbf{X}_j^{(0)} - \mathbf{X}_{-j}^{(0)}\boldsymbol{\gamma}_j)^\top \mathbf{X}_j^{(0)}/n_0 - (\mathbf{X}_{\boldsymbol{\beta}^{(0)},j}^{(0)} - \mathbf{X}_{\boldsymbol{\beta}^{(0)},-j}^{(0)}\boldsymbol{\gamma}_j)^\top \mathbf{X}_{\boldsymbol{\beta}^{(0)},j}^{(0)}/n_0| \\
&= \left|\Sigma_{j,j}^{(0)} - \boldsymbol{\Sigma}_{j,-j}^{(0)}\boldsymbol{\gamma}_j - \left(\Sigma_{\boldsymbol{\beta}^{(0)},j,j}^{(0)} - \boldsymbol{\Sigma}_{\boldsymbol{\beta}^{(0)},j,-j}^{(0)}\boldsymbol{\gamma}_j\right)\right| \\
&\leq h_3
\end{aligned}
$$

as assumed in Condition (c) in Assumption 8. Therefore, the result follows.

## G    Additional Experiments and Details

To implement the methods mentioned in Section 4.1, we use R package *glmnet* for naive-Lasso, R package *glmtrans* for Trans-GLM, and R package *ncvreg* for our proposed CoRT (Breheny and Huang, 2011; R Core Team, 2025). Loh and Wainwright (2015) shows that the coordinate descent algorithms in Breheny and Huang (2011) are guaranteed to converge to a stationary point and matches our theoretical development. All the simulations are on a desktop computer running Windows 11 with an Intel Core i9 CPU at 5 GHz with 32 GB of RAM.

### G.1    Additional Simulation for Section 4.1

In the known $\mathcal{S}$ case setting, we only present the $\ell_2$ estimation error for logistic regression. Here, we include the AUC score.

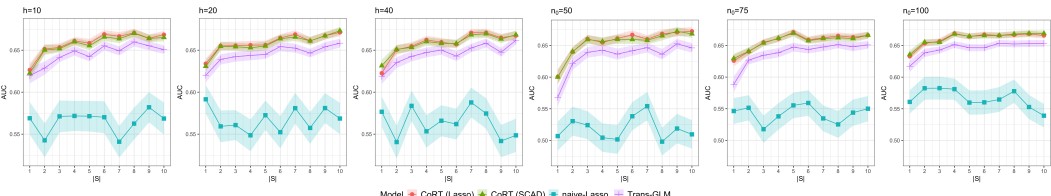

Figure 5: Averaged AUC scores for logistic regression.

In the unknown $\mathcal{S}$ case setting, we only present the $\ell_2$ estimation error. Here, we include the prediction error for linear and logistic regression in Figure 6 for completeness.

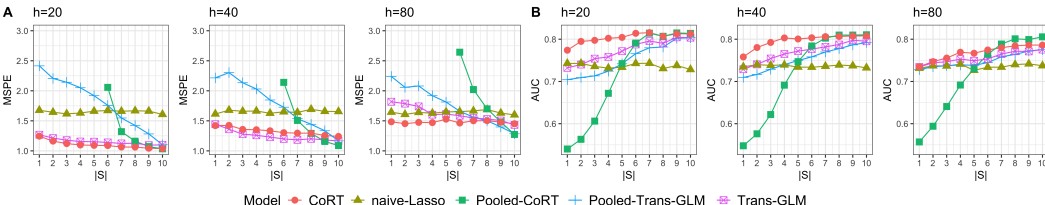

Figure 6: The MSPE for linear regression (Panel A) and averaged AUC scores for logistic regression (Panel B).

## G.2 Computational Complexity Comparison

We conduct the following two additional experiments to investigate the computational cost w.r.t. $K$ and $p$, with the same settings in Section 4. For our proposal, we consider two regularizers, Lasso and SCAD. In scenario 1, we fix $p = 300$ and let the number of sources range from 10 to 30. In scenario 2, we fix $n_0 = 50$ and let $p$ range from 300 to 700. Results are in seconds and averaged from 100 replicates.

Table 3: Results for varying source number $K$.

| $K$ | Trans-GLM | Naive-Lasso | CoRT (Lasso) | CoRT (SCAD) |
|-----|-----------|-------------|--------------|-------------|
| 10  | 0.41      | 0.11        | 0.24         | 0.54        |
| 15  | 0.53      | 0.11        | 0.40         | 0.80        |
| 20  | 0.66      | 0.12        | 0.67         | 1.37        |
| 25  | 0.91      | 0.11        | 1.01         | 2.03        |
| 30  | 1.24      | 0.11        | 1.51         | 2.99        |

Table 4: Results for varying dimension $p$.

| $p$ | Trans-GLM | Naive-Lasso | CoRT (Lasso) | CoRT (SCAD) |
|-----|-----------|-------------|--------------|-------------|
| 300 | 0.41      | 0.11        | 0.23         | 0.54        |
| 400 | 0.63      | 0.10        | 0.49         | 0.88        |
| 500 | 0.74      | 0.08        | 0.92         | 1.49        |
| 600 | 0.92      | 0.08        | 1.54         | 2.28        |
| 700 | 1.18      | 0.09        | 2.17         | 3.56        |

The computation time of our proposed CoRT (Lasso) and CoRT (SCAD) increases as K or p increases. This is because we use the coordinate descent algorithm to implement our methods. In either case, the increasing $\mathcal{S}$ or $p$ results in more parameters to be estimated, as our methods need to find the contrast $\boldsymbol{\delta}^{(k)}$. However, we note that for high-dimensional data with $p \gg n$, the CoRT still achieves great scalability, despite its running time.

## G.3 COVID Data Details

The data that we study is synthesized from various government and nonprofit institutions for all 3142 counties in the US. The data are stored from 1/22/20-12/21/20 and publicly available [3]. We refer interested readers to Li et al. (2021) for more details about the data collection. County-level characteristics include demographic, race, socioeconomic, and medical comorbidities variables.

The data preprocessing leads to 12 target states, including Alabama (AL-67), California (CA-58), Colorado (CO-64), Florida (FL-67), Louisiana (LA-64), Montana (MT-56), North Dakota (ND-53), New York (NY-62), Pennsylvania (PA-67), South Dakota (SD-65), Wisconsin (WI-72), and West Virginia (WV-55). This leads to 7 source states, including Georgia (GA-159), Illinois (IL-102), Kansas (KS-105), Kentucky (KY-120), Missouri (MO-115), Texas (TX-253), and Virginia (VA-133). The numbers in the brackets denote the number of counties. We pick states with 50 to 75 counties as the target states, while source states have more than 100 counties. These numbers reflect the fact that source sample sizes are typically larger than target sample sizes.

We use the R packages *randomForest*, *xgboost*, and *e1071* (R Core Team, 2025) for implementations. We use max_depth=15 and nrounds=50 for XGBoost and default parameters in the other packages.

---

[3]`https://github.com/lin-lab/COVID-Health-Disparities`

