# OpenReview forum: "Co-Regularization Enhances Knowledge Transfer in High Dimensions"
_NeurIPS.cc/2025/Conference — NeurIPS 2025 poster_

### Official Review · Reviewer_MAPn · 2025-06-26

**Clarity:** 3
**Significance:** 2
**Originality:** 3
**Rating:** 4
**Confidence:** 5

**Summary:**

This paper discusses the multi-source transfer learning for high-dimensional generalized linear models (GLMs). The Author proposed a co-regularization process that simultaneously learns source and target hypothesis with target domain and source domains that are similar to the target. Theoretical results shows that the proposed CoRT has a tighter error bound and a narrower confidence interval than target-only Lasso. Empirical studies validated its effectiveness.

**Questions:**

1. Discussion for more general model is suggested. Will the CoRT still work for more complicated non-linear model? What about the convergence?

2. The idea of Co-Regularization seems to be similar to the multi-task learning. The similarities and differences between them should be discussed.

3. Further, I am curious about whether the training of target model will be disturbed by the gradient of the source models, making it harder to reach the optimal solution.

4. Since all models are known in the simulation experiment, numerical validation of the error bounds and the convergence behavior will more strongly support the main theorems of the paper.

If the authors can address most of the questions, I am willing to raise the score.

**Ethical Concerns:**

["NO or VERY MINOR ethics concerns only"]

**Final Justification:**

My main methodological concerns have been largely addressed.

**Limitations:**

Discussion for more general model is lacked.

**Paper Formatting Concerns:**

None.

**Quality:**

2

**Strengths And Weaknesses:**

Strengths:

1. The method has clear modeling logic and is easy to follow.

2. The theoretical analysis provide guarantee to the convergence of the optimization process.

Weaknesses

1. The paper only discussed method and theory for the GLMs, while even the small datasets may have more complicated hypothesis space. Thus, further discussion for more general model is necessary.

2. The objective of the CoRT, i.e. Eq. (1), seeks for optimal solution of both source and target model, which is harder than only training the optimal target model. The gradients may cause negative impact to each other.

3. The evaluation experiments are insufficient and further numerical analysis of the method is lacked.

---

> ### Author Rebuttal · Authors · 2025-07-31
>
> Thank you for your comments! Please see our responses in the following.
>
> **1. The paper only discussed method and theory for the GLMs, while even the small datasets may have more complicated hypothesis space. Thus, further discussion for more general model is necessary. Discussion for more general model is suggested. Will the CoRT still work for more complicated non-linear model? What about the convergence?**
>
> We appreciate the insightful comment.
>
> The key idea of CoRT can actually generalize beyond GLMs naturally. This is because the core of CoRT is to simultaneously regularize the complexity of (1) the bias between source and target models and (2) the target model itself; see Eq. (1). Therefore, for more generic cases, we can modify the objective to
> $$
> \sum\_{k \in \mathcal{S}} \left( \text{Loss}( f^{(0)} + \delta^{(k)}, \mathcal{D}^{(k)} )  + \lambda\_{k} \\|\delta^{(k)} \\|\_{\mathcal{H}\_{1}}  \right) + \text{Loss}( f^{(0)} , \mathcal{D}^{(0)} ) + \lambda\_{0} \\| f^{(0)} \\|\_{\mathcal{H}\_{2}}
> $$
>
> where $\mathcal{H_{1}}$, $\mathcal{H}_{2}$ represent generic hypothesis spaces and $f$ denote the parameters/models of interests. Therefore, the non-linear models can be incorporated into the CoRT framework naturally by setting the hypothesis spaces as specified function spaces, like RKHS, class of DNN, etc. Besides, the adaptive techniques (Algorithm 2) for excluding outlier sources are also model-agnostic, which thus can be generalized well to other models and used broadly by the community.
>
> We focused on high-dimensional GLMs for two primary reasons.
> - First, since our motivation is to provide a better knowledge transfer training scheme than the two-step seminal frameworks [1] and [2] paradigm, we follow their setting which allows us to make head-to-head comparison both theoretically and empirically. Besides, the generalization of CoRT is analogous to the generalization of the two-step frameworks, where the techniques of regularizing the bias between source and target parameters in GLMs have already been naturally extended to different parametric/non-parametric models, from graphic models, to autoregressive models, and even neural networks.
>
> - Second, the high-dimensional GLMs has already been a relatively broad model family and has shown its excellent versatility in various scientify fields such as biological and social applications. Therefore, we believe the study of transfer learning in GLMs provides researchers from different fields with powerful tools, while the extension of CoRT to other models can be natural.
>
> For the convergence results, we believe the form of the error bounds for different models should stay in a similar fashion as long as it applies to the CoRT framework, but their exact form should be case by case since the model assumptions and analysis tools are different. This is consistent with prior literature: when the two-step method was extended from GLMs to nonparametric regression [3], autoregressive models [4], or Fréchet regression [5], the form of the convergence rate was preserved, with only the model-specific rates being updated, e.g., replacing the sparse learning parametric rate by the nonparametric rate in [3]. Therefore, we expect the form of convergence rates for using CoRT on different models will stay similar asymptotically, but just having the components being more model-specified. We believe this can be an interesting future direction.
>
>
> **2. The objective of the CoRT, i.e. Eq. (1), seeks for optimal solution of both source and target model, which is harder than only training the optimal target model. The gradients may cause negative impact to each other. Further, I am curious about whether the training of target model will be disturbed by the gradient of the source models, making it harder to reach the optimal solution.**
>
> We thank the reviewer for this insightful question.
>
> As the objective indicates, the training is not to seek optimal solutions for target and source models, i.e., $\beta^{(0)}$ and $\beta^{(k)}$. Instead, we reparameterize $\beta^{(k)}$ as $\beta^{(0)} +\delta^{(k)}$, seeking for optimal solution for the target coefficient and the bias terms, i.e., $\delta^{(k)}$. While simultaneously optimizing them can be challenging, our optimization procedure, based on coordinate descent algorithms (for nonconvex penalized regression [6] via package ncvreg), iteratively updates the target model $\beta^{(0)}$ and each bias $\delta^{(k)}$ term. It effectively decouples the optimization steps and ensures a more stable convergence path by isolating the high-dimensional gradient updates. This procedure guarantees convergence to a stationary point [7].
>
> We also mentioned this in the appendix, see line 659.
>
>
> **3. The idea of Co-Regularization seems to be similar to the multi-task learning. The similarities and differences between them should be discussed.**
>
> It is worth noting that multi-task learning is closely related to transfer learning, though their objectives differ. Multi-task learning aims to solve multiple learning tasks simultaneously by leveraging shared structure across tasks, like the Data Shared Lasso framework [8, 9] in high dimensions. In contrast, transfer learning focuses on improving performance on a specific target task by transferring knowledge from source data.
>
> We believe that ``CoRT is similar to multi-task learning'' is due to the fact that the objective of CoRT, i.e., Eq. (1), looks like the objective in multi-task learning. However, we would like to note that the CoRT objective is actually asymmetric w.r.t. the source and target, as the last two terms put specific emphasis on the target model (thus focus on optimizing the learning over the target domain). This is the core of CoRT, as it adopts the spirit of multi-task learning via a shared objective function across domains while having its own special focus on the target. We will add related discussion in the revised version once more space is allowed.
>
>
> **4. Since all models are known in the simulation experiment, numerical validation of the error bounds and the convergence behavior will more strongly support the main theorems of the paper.**
>
> Yes, all the simulation studies, as well as the real data study, were designed to support the theoretical results. The trends in the experiments all illustrate the convergence behavior of the main theorems, and are discussed in the result description in Sections 4 and 5.
>
> **Reference**
>
> [1] Li, S., Cai, T. T., and Li, H. (2022). Transfer learning for high-dimensional linear regression: Prediction, estimation and minimax optimality. Journal of the Royal Statistical Society Series B: Statistical Methodology.
>
> [2] Tian, Y. and Feng, Y. (2023). Transfer learning under high-dimensional generalized linear models. Journal of the American Statistical Association.
>
> [3] Lin, H., & Reimherr, M. (2024). Smoothness adaptive hypothesis transfer learning. ICML.
>
> [4] Ma, M., & Safikhani, A. (2025). Transfer Learning for High-dimensional Reduced Rank Time Series Models. AISTATS
>
> [5] Zhang, K., Zhang, S., Zhou, D., & Zhou, Y. (2025). Wasserstein Transfer Learning. arXiv.
>
> [6] Breheny, P., & Huang, J. (2011). Coordinate descent algorithms for nonconvex penalized regression, with applications to biological feature selection. The annals of applied statistics.
>
> [7] Loh, P. L., & Wainwright, M. J. (2015). Regularized M-estimators with nonconvexity: Statistical and algorithmic theory for local optima. The Journal of Machine Learning Research.
>
> [8] Samuel M Gross and Robert Tibshirani. (2016). Data shared lasso: A novel tool to discover uplift. Computational Statistics & Data Analysis.
>
> [9] Edouard Ollier and Vivian Viallon. (2017). Regression modelling on stratified data with the
> lasso. Biometrika.

---

> > ### Comment · Reviewer_MAPn · 2025-08-01
> >
> > Thank you for the rebuttal. However, I still have concerns regarding the **co-optimization procedure.**
> >
> > From the formulation in Eq. (1), the target model $\beta^0$ and each source model $\delta^k$ appear to be treated symmetrically. In fact, the last two terms in the domain loss $\mathcal{L}_D$ can essentially be folded into the summation term, making it unclear whether the objective function is specifically tailored to emphasize the optimization of the target model.
> >
> > Moreover, while the authors note that the coordinate descent algorithm guarantees convergence to a stationary point, it only guarantees the **local optima.** One concrete concern is whether it is possible for the optimization to **reach global optima for some source models $\delta^k$ while only attaining a suboptimal local solution for the target model $\beta^0$.** Given that the number of source models $k$ is typically larger than one, such an imbalance could make the optimization **more prone to favoring source models.**

---

> > > ### Author Response · Authors · 2025-08-03
> > >
> > > Thanks for the insightful questions.
> > >
> > > First, the loss function was designed to learn both the target parameter and the contrast parameter $\delta^{(k)}$, not the source parameter $\beta^{(k)}$. Using the coordinate descent algorithm, we are not learning the source parameter from the loss function, but to improve the learning of the target parameter both from the estimation of the contrast parameter and from the larger sample size. Therefore, the loss objective is asymmetric in terms of $\beta^{(0)}$ and $\beta^{(k)}$.
> > >
> > > For the convergence issue, we believe the reviewer has concerns about the non-convexity of the loss objective (due to using non-convex penalty function, such as SCAD), in comparison to convex loss objective (using Lasso penalty). We’d like to discuss the theoretical and empirical results. From the theoretical side, with certain conditions, Corollary 2 of [7] shows that the coordinate descent algorithm is guaranteed to converge to a stationary point within close proximity of the true parameter. In our manuscript, we mentioned those conditions both in Remark 4 (Line 469) and Lemma 1 (Line 552). Therefore, both the target parameter and contrast parameter converge to a stationary point within close proximity of the true parameter theoretically, with certain conditions. From the empirical side, CoRT with nonconvex SCAD performs better than CoRT with Lasso. For example, see Figure 1(a) that CoRT (SCAD) has lower estimation errors than CoRT (Lasso) in all scenarios. We further comment that $\delta^{(k)}$ measures the difference between $\beta^{(k)}$ and $\beta^{(0)}$, and learning $\delta^{(k)}$ is to help the estimation of $\beta^{(0)}$. It will NOT be prone to source model since our model does not estimate source parameter $\beta^{(k)}$ as an individual parameter and we require the l1 norm $||\delta^{(k)}||_1<h$. When there are outlier sources, we use Algorithm 1 to first select the transferable sources.
> > >
> > > We hope this helps address the reviewer’s concerns.

---

> > > > ### Comment · Reviewer_MAPn · 2025-08-06
> > > >
> > > > Thank you for the authors’ response, which addresses part of my concerns. However, I still have reservations regarding the design and underlying principle of the objective. In particular, it remains unclear to me—at an intuitive level—how incorporating $\delta^k$ into the optimization contributes to learning a better $\delta^0$.

---

> > > > > ### Author Response · Authors · 2025-08-07
> > > > >
> > > > > We would like to clarify that there is no quantity $\delta^{(0)}$. Given that the mathematical definition of $\delta^{(k)}$ represents the difference between $\beta^{(0)}$ and $\beta^{(k)}$, the $\delta^{(0)}$ will be 0. We believe what the reviewer refers to is ``how incorporating $\delta^{(k)}$ into the optimization contributes to learning a better $\beta^{(0)}’’.
> > > > >
> > > > > Regarding this, we would like to go through the evolution of our idea, which we believe will better illustrate the idea.
> > > > >
> > > > > When the learner has access to both source and target data, one of the most naive estimators for transfer learning can be obtained by finding the minimizer of the pooled objective over all datasets. However, this indeed comes with the problem that we treat all sources and targets equally (think of multi-task learning where target tasks are not focused).
> > > > > To emphasize on the target task, i.e., a better estimation for $\beta^{(0)}$, we introduce $\delta^{(k)}=\beta^{(k)} - \beta^{(0)}$ to incorporate the source data in a more flexible compared to pooling approach. These $\delta^{(k)}$ can be viewed as adapters but they are intentionally estimated (sparsely) by the regularization terms in the objective. A "larger" $\delta^{(k)}$ is indeed less similar between the $k$-th source and the target, making terms involving $\delta^{(k)}$ more isolated in the objective. In contrast, a "smaller" $\delta^{(k)}$ indicates higher similarity and tries to make the objective act like a pooled loss over $k$-th source and target.
> > > > >
> > > > > We also believe a comparison to multi-task learning can help better to intuitively illustrate our design (point 3 we wrote in our initial response). In multi-task learning, each task parameter can be decomposed as $\beta^{(k)} = \beta + \delta^{(k)}$ where $\beta$ is the shared model across tasks and $\delta^{(k)}$ is the task specific component. In our design, we don’t have such $\beta$ and make $\beta^{(0)}$ in the central place, while $\delta^{k}$ serves as a bridge to acquire knowledge from source.

---

> > > > > > ### Comment · Reviewer_MAPn · 2025-08-07
> > > > > >
> > > > > > Thank you for your response. Could you further elaborate on why $\delta^k$ is inherently more beneficial than $\beta^k$ for learning $\beta^0$? I believe this clarification is crucial for distinguishing your method from multi-task learning and for justifying its suitability for transfer learning. In particular, I am curious about your statement: "In our design, we don’t have such $\beta$ and make $\beta^0$ in the central place, while $\delta^k$ serves as a bridge to acquire knowledge from source." I would appreciate a deeper explanation.

---

> > > > > > > ### Author Response · Authors · 2025-08-07
> > > > > > >
> > > > > > > We would like to clarify that if one does NOT parametrize $\beta^k$ in some specific way to associate it with $\beta^{0}$, one is indeed learning them separately, i.e. return back to single task learning. All the learning paradigms that leverage the similarity between tasks to enhance learning efficiency, no matter multi-task or transfer learning, requires certain types of similarity structure and thus structure constraints. That is why we need to re-parametrize $\beta^k $ and connect it to $\beta^0$.  The reparametrization we consider is particularly beneficial in learning $\beta^0$ as all source and target data contribute to learning $\beta^0$, which is more efficient than using target data only (when there is no connection between $\beta^0$ and $\beta^k$).
> > > > > > >
> > > > > > > Regarding our statement, we don’t want to add additional confusion, thus we would like to explain it via the following comparison.
> > > > > > >
> > > > > > > - **Multi-task learning**: for $k \in 0 \cup S$, denote $\beta^{k} = \beta + \delta^{k}$ and the objective becomes
> > > > > > > $$
> > > > > > > \hat{\beta}, \\{ \hat{\delta^{k}} \\}\_\{k \in 0 \cup S \} = \underset{  \beta,\{\delta^{k}\}\_\{k \in 0 \cup S\} }{\operatorname{argmin}}   \sum\_{k \in 0 \cup S} \mathcal{L}( \beta + \delta^{k}, \mathcal{D}^{k} ) + P\_{\lambda_{k}}(\delta^{k})
> > > > > > > $$
> > > > > > >
> > > > > > > - **CoRT (Transfer Learning)**: for $k \in S$, denote $\beta^{k} = \beta^{0} + \delta^{k}$ and the objective becomes
> > > > > > > $$
> > > > > > > \hat{\beta^0}, \\{ \hat{\delta^{k}} \\}\_\{k \in  S \} = \underset{  \beta^0, \{\delta^{k}\}\_\{k \in  S\} }{\operatorname{argmin}}   \\{ \sum\_{k \in  S} \mathcal{L}( \beta^{0} + \delta^{k}, \mathcal{D}^{k} ) + P\_{\lambda_{k}}(\delta^{k}) \\} + \mathcal{L}( \beta^{0}, \mathcal{D}^{k} ) + P\_{\lambda_0}(\beta^{0})
> > > > > > > $$
> > > > > > >
> > > > > > > Through such parameterization, we allow $\beta^{0}$ to keep in the source loss function and there exploit the benefit of using source samples (while in multi-task learning, only the $\beta$ enjoys such benefit).

---

> > > > > > > > ### Comment · Reviewer_MAPn · 2025-08-08
> > > > > > > >
> > > > > > > > Thank you for your response. After considering the series of clarifications you have provided, my main methodological concerns have been largely addressed. I recommend that the authors further emphasize the underlying principles that make CoRT effective for transfer learning, so as to alleviate potential doubts from researchers in this field. In addition, given that the primary area of this work is *Theory*, it would be valuable to incorporate a discussion linking the method to theoretical analyses of target-domain generalization error in transfer learning. In light of these considerations, I will raise my final rating.

---

> > > > > > > > > ### Author Response · Authors · 2025-08-08
> > > > > > > > >
> > > > > > > > > Thank you!
> > > > > > > > >
> > > > > > > > > We are happy that these clarifications help to address your concerns. If we finally get the chance to include an extra page in the camera-ready version, we will turn some of the clarifications in this rebuttal into extra paragraphs that emphasize the underlying principles of CoRT. We also plan to add the comparison to multi-task learning in the appendices.
> > > > > > > > >
> > > > > > > > > Thank you for the suggestion of linking our theoretical results to the domain adaptation/transfer learning generalization error. We will expand our response to point 4 of Reviewer CHM6 to link our theoretical analyses to these classical results.

---

### Official Review · Reviewer_1KFr · 2025-06-30

**Clarity:** 3
**Significance:** 2
**Originality:** 2
**Rating:** 3
**Confidence:** 3

**Summary:**

This paper investigates high dimensional (dimension larger than the number of data in target domain) transfer learning with General Linear Models. Unlike standard two steps approach, pretraining on the source data and fine tuning on the target data, this paper proposed a co-regularization learning scheme where source data act as a regularization for learning in the target domain. Specifically, the paper proposes co-regularization transfer (CoRT), which estimates the target domain parameter $\beta$ and parameter contrast $\delta$ simultaneously. Moreover, the authors propose an adaptive algorithm that identifies potential outlier sources to avoid transferring "bad" sources. Simulations show the favorable performance of CoRT compared to two-step training schemes.

**Questions:**

See weakness.

**Ethical Concerns:**

["NO or VERY MINOR ethics concerns only"]

**Quality:**

3

**Strengths And Weaknesses:**

Strength:
1. The paper is overall well written and easy to follow.
2. Strong statistical analysis is provided that shows the clear advantage of the proposed CoRT framework vs existing baselines such as Trans-GML.

Weakness:

1. Even though the specific combinations of components are unique, the idea of one step learning for both source and target domain are not exactly brand new in the literature of domain adaptation.
2. The practicality of the paper is also of concern, as even though simultaneously training on source and target domain would intuitively leads to better estimation error, having access to both source data at the time of learning in target domain is a rather restrictive assumption. As the establishment of two-step training paradigm is partially conditioned on the difficulty of training on both source and target data. For example, the proposed approach would make switching to new target domain extremely expensive, as the source data need to be involved all the time.
3. There are some minor presentation issues, including really small font size in Figure 1, making them hard to read.

---

> ### Author Rebuttal · Authors · 2025-07-31
>
> Thank you for your comments! Please see our responses in the following.
>
> **1. The practicality of the paper is of concern, as even though simultaneously training on source and target domain would intuitively leads to better estimation error, having access to both source data at the time of learning in target domain is a rather restrictive assumption. As the establishment of two-step training paradigm is partially conditioned on the difficulty of training on both source and target data. For example, the proposed approach would make switching to new target domain extremely expensive, as the source data need to be involved all the time.**
>
> We thank the reviewer for the insightful comment. We agree that the proposal CoRT may appear restrictive in some scenarios, but we believe the approach remains both meaningful and practical for the following reasons:
>
> (1) First, we’d like to point out that those seminal works for the two-step paradigm for high-dimensional data, such as Trans-GLM [1], also necessitated access to source datasets during their training phase. In these works, having access to source data is not only common but often necessary to ensure more robust knowledge transfer across domains as they need to control the training on source domains. Therefore, our work actually considers the same setting as those seminal transfer learning works in high-dimensional GLMs. The proposed joint training scheme (co-regularization in our model context) advances the two-step paradigm under the same setting by being more tolerant to less similar sources (better utilities knowledge from these sources.)
>
> (2) Second, having access to both source and target data is common in the field of transfer learning. Methodologically, researchers from core-ML, statistics and deep learning community have make such assumptions to achieve design algorithm that can transfer knowledge better; see [2,3,4,5,6] and more reference therein. Besides, this assumptions is also practically satisfied in some specific studies, such as biological applications. For example, [7] investigates genetic effects on obesity in a multicenter study where data from both Black/African American (target) and White (source) populations are available. In [8], methods for multi-omics data treat one omic as the target while the others serve as sources.
>
> (3) From a theoretical aspect, access the source data ensures the error bounds, such as convergence rates, to incorporate both source and target size. This allows us to provide insight into when transfer learning outperforms the non-transfer case without directly making assumptions on pre-trained models (like those domain adaptation works ranging from 2006 to 2010s).
>
> (4) Finally, although CoRT requires retraining when switching to new target tasks, distributed techniques, like divide-and-conquer, can be used to practically improved the computational efficiency. The algorithm and theory for divide-and-conquer high-dimensional statistical models have been well-developed with theory to guarantee their performance. These distributed algorithms can be directly plug-in into the CoRT framework in practice when computation expense is a concern. However, this is out of the scope of current manuscript.
>
> At a high level, our vision of the proposed CoRT is that it is a complement to the modern transfer learning community, rather than a replacement of the successful two-step paradigm. We believe our work does provide a new perspective to better leverage the source knowledge. To provide a balanced perspective, we will also dedicate a discussion to the potential limitations and challenges of this method, together with potential distributed style solutions in practice, in the revised version.
>
>
>
> **2. Even though the specific combinations of components are unique, the idea of one step learning for both source and target domain are not exactly brand new in the literature of domain adaptation.**
>
> We agree with the reviewer that the one step learning for both source and target is not exactly brand new in the community, especially in computer vision field. These one step approaches are also typically require access to both source and target data and retraining when switching to new target task (this actually also reasonalizes our setting better in question 1).
>
> Our contribution, however, is specifically situated within the high-dimensional $(p\gg n)$ setting, where the research landscape is markedly different.
>
> In this high-dimensional regime, the community has almost exclusively followed the two-step paradigm since the seminal works of [1,4,9]. This two-step approach has been the default for adapting transfer learning to various high-dimensional models. Our work, CoRT, challenges this status quo by introducing a fundamentally different, one-step paradigm tailored for this setting. The novelty of our work is therefore twofold:
>
> - Methodological contribution: The design of CoRT departs from the dominant paradigm, leverages the source knowledge better, and accommodates less similar source, which are all illustrated by the theories and experiments.
>
> - Technical Contributions: Analyzing a joint-training scheme requires a completely different theoretical approach than analyzing a two-step procedure. We develope new mathematical tools to derive the results for joint training (CoRT), which, we believe, can benefit the community in the future.
>
>
>
>
> **3. There are some minor presentation issues, including really small font size in Figure 1, making them hard to read.**
>
> Thank you for pointing this out. We admit that it is hard to trade off between informativeness and readability with multiple figures are presented. We will revise the figures to improve readability once more spaces are allowed.
>
>
> **Reference**
>
> [1] Tian, Y. and Feng, Y. (2023). Transfer learning under high-dimensional generalized linear models. Journal of the American Statistical Association, 118(544):2684–2697.
>
> [2] Dai, Wenyuan, et al. (2006) Eigentransfer: a unified framework for transfer learning. ICML.
>
> [3] Du, Simon S., et al. (2017) Hypothesis transfer learning via transformation functions. NeurIPS.
>
> [4] Li, S., Cai, T. T., & Li, H. (2022). Transfer learning for high-dimensional linear regression: Prediction, estimation and minimax optimality. Journal of the Royal Statistical Society Series B: Statistical Methodology.
>
> [5] Sun, B., & Saenko, K. (2016, October). Deep coral: Correlation alignment for deep domain adaptation. In European conference on computer vision.
>
> [6] Ganin, Yaroslav, et al. (2016) Domain-adversarial training of neural networks. Journal of machine learning research.
>
> [7] Li, S., Cai, T., & Duan, R. (2023). Targeting underrepresented populations in precision medicine: A federated transfer learning approach. The annals of applied statistics.
>
> [8] Lac, L., Leung, C. K., & Hu, P. (2024). Computational frameworks integrating deep learning and statistical models in mining multimodal omics data. Journal of Biomedical Informatics.
>
> [9] Bastani, H. (2021). Predicting with proxies: Transfer learning in high dimension. Management Science.

---

> > ### Comment · Reviewer_1KFr · 2025-08-05
> >
> > I would like to thank the authors for their response.
> >
> > However, my concern over the practicality of the one step approach remains, as the provided references are mostly "old" works with relatively simple contexts, which does not really translate to the focus of current age knowledge transfer problems where data are not only high in dimension, but also extremely high in volume. This is also part of the reason why the community mostly have been focusing on the two-step approach (as pointed out the authors).
> >
> > Therefore, I would like to keep my original score.

---

> > > ### Author Response · Authors · 2025-08-06
> > >
> > > We respectively clarify that our work evolves the knowledge transfer process in high dimensions [1,4], which are published within 3 years. Besides, tons of papers have also been focused on such topics and researchers in the community are still actively looking for approaches to make the process more effective and efficient. It is unfair to consider they are ``old’’ works, but very fundamental frameworks for high-dimensional statistical models. These models have been and still serve as the backbone in a lot of fields, especially biology, medical and related ones.
> > >
> > > Specifically, data in a high volume scenario is common in some specific field, like CV or NLP, but uncommon in subfields of biology where collecting high volume data is expensive. The high expense in collecting data, less samples in certain species that biologists can collect, and genetics data for rare disease patients, all justify the importance of considering the high-dimensional  regime $p \gg n$.
> > >
> > > Besides, as we wrote in our responses, the two-step approach also necessitates access to source datasets during training and we also commented that accessing source data is not only common in landmark statistical and ML works but to ensure more robust knowledge transfer in theory. Therefore, from these aspects, two-step approaches does not provide additional benefits.
> > >
> > > Finally, we did NOT think the transfer learning research should all follow the two-step paradigm simply because they are mostly focused. While we admit some potential limitations of our proposal, this framework supplements the transfer learning community from a different perspective. It presents a more effective knowledge transfer process, illustrated both theoretically and empirically compared to existing works, providing new possibilities in the community.

---

### Official Review · Reviewer_CHM6 · 2025-07-03

**Clarity:** 2
**Significance:** 3
**Originality:** 4
**Rating:** 5
**Confidence:** 3

**Summary:**

This paper introduces the Co-Regularization Transfer (CoRT) framework for multi-source transfer learning in generalized linear models (GLMs). Instead of the traditional two-step approach of pre-training and fine-tuning, CoRT jointly estimates the target parameter and source deviations via co-regularized optimization. The paper contributions also include i) an adaptive knowledge transfer algorithm using the majority vote to detect outlier source domains; ii) an asymptotic element-wise confidence interval for the estimate of the target domain parameter; and iii) an extensive theoretical analysis proving robustness to heterogeneous source domains.

**Questions:**

- Please elaborate on runtime complexity and scalability to large K (number of sources) or p (dimensionality).
- Regarding the derived theoretical guarantees, how do these relate to or differ from discrepancy-based domain adaptation methods in terms of assumptions and guarantees?
- Are you willing to release the source code implementing your methods and experiments?

**Ethical Concerns:**

["NO or VERY MINOR ethics concerns only"]

**Final Justification:**

The authors agreed on including an experimental analysis of the computational complexity of their method, and I also found their remaining answers to my questions sufficiently convincing. Thus, and also taking into account the other reviews and the authors' responses to them, I keep my recommendation for acceptance.

**Limitations:**

The limitations are only marginally addressed in the Discussion section. The paper would benefit from having a dedicated Limitations section.

**Quality:**

3

**Strengths And Weaknesses:**

Strengths:
- The proposed CoRT framework is novel and avoids the bias introduced by the usual pre-training + fine-tuning paradigm.
- The paper presents solid theoretical foundations and the provided convergence bounds compare favorably with the existing methods.
- The proposed adaptive algorithm for identifying outlier sources eliminates the need for hyperparameter tuning, which is a significant practical advantage over existing methods.
- The experimental results both in synthetic and real world data confirm the effectiveness of the method

Weaknesses:
- Although the authors claim that the methodology can be extended to other types of models, the paper is focused on GLMs, so it becomes unclear how broadly CoRT can generalize beyond this family.
- There’s no discussion on computational complexity of the derived algorithms.
- The real world dataset (COVID-19 mortality) is insightful but relatively small.
- The authors claim that CoRT can identify which specific covariates in the source domain are transferable, enhancing interpretability. While I agree with this observation, it would be nice to see this analysis in the experiments.

---

> ### Author Rebuttal · Authors · 2025-07-31
>
> Thank you for your comments! Please see our responses in the following.
>
> **1. Although the authors claim that the methodology can be extended to other types of models, the paper is focused on GLMs, so it becomes unclear how broadly CoRT can generalize beyond this family.**
>
> We believe the key idea of co-regularization can generalize beyond GLMs naturally, as the key idea is to simultaneously regularize (1) the distance between source and target parameters and (2) the target parameter itself; see Eq. (1).
>
> Therefore, for more generic cases, we can modify the objective to
> $$
> \sum\_{k \in \mathcal{S}} \left( \text{Loss}( f^{(0)} + \delta^{(k)}, \mathcal{D}^{(k)} )  + \lambda\_{k} \\|\delta^{(k)} \\|\_{\mathcal{H}\_{1}}  \right) + \text{Loss}( f^{(0)} , \mathcal{D}^{(0)} ) + \lambda\_{0} \\| f^{(0)} \\|\_{\mathcal{H}\_{2}}
> $$
> where $\mathcal{H_{1}}$, $\mathcal{H}_{2}$ represent generic hypothesis spaces and $f$ denote the parameters/models of interests. This allows us to incorporate most linear/non-linear models that learned via empirical risk minimization. Besides, the adaptive techniques (Algorithm 2) for excluding outlier sources are also model-agnostics, which thus can be generalized well to other models and used broadly by the community.
>
> We focused on high-dimensional GLMs for two primary reasons.
> - First, since our motivation is to provide a better knowledge transfer training scheme than the two-step seminal frameworks [1] and [2] paradigm, we follow their setting which allows us to make head-to-head comparison both theoretically and empirically. Besides, the generalization of CoRT is analogous to the generalization of the two-step frameworks, where the techniques of regularizing the bias between source and target parameters in GLMs has already been naturally extended to different parametric/non-parametric models, from graphic models, to autoregressive models, and even neural networks.
>
> - Second, the high-dimensional GLMs has already been a relatively broad model family and has shown its excellent versatility in various scientify fields such as biological and social applications. Therefore, we believe the study of transfer learning in GLMs provides researchers from different fields with powerful tools, while the extension of CoRT to other models can be natural.
>
>
>
>
> **2. There’s no discussion on computational complexity of the derived algorithms. Please elaborate on runtime complexity and scalability to large K (number of sources) or p (dimensionality).**
>
> To optimize objective Eq. (1), we leverage the package ncvreg [3], which contains a very efficient and robust optimization coordinate-descent algorithm.
>
> We conduct the following two additional experiments to investigate the computational cost w.r.t. $K$ and $p$, with the same settings in Section 4. For our proposal, we consider two regularizers, Lasso and SCAD.
> In scenario 1 (Table 1), we fix $p=300$ and let the number of sources range from 10 to 30. In scenario 2 (Table 2), we fix $n_{0} = 50$ and let $p$ range from $300$ to $700$. Results are in seconds and averaged from 100 replicates.
>
>
> | K | Trans-GLM      |  naive-Lasso      | CoRT (Lasso)       | CoRT (SCAD) |
> |----------------|----------------|----------------|----------------|----------------|
> | 10 | 0.41  | 0.11  | 0.24  | 0.54 |
> | 15 | 0.53  | 0.11  | 0.40  | 0.80 |
> | 20 | 0.66  | 0.12  | 0.67  | 1.37 |
> | 25 | 0.91  | 0.11  | 1.01  | 2.03 |
> | 30 | 1.24  | 0.11  | 1.51  | 2.99 |
>
>
> | p | Trans-GLM      |  naive-Lasso      | CoRT (Lasso)       | CoRT (SCAD) |
> |----------------|----------------|----------------|----------------|----------------|
> | 300 | 0.41  | 0.11  | 0.23  | 0.54 |
> | 400 | 0.63  | 0.10  | 0.49  | 0.88 |
> | 500 | 0.74  | 0.08  | 0.92  | 1.49 |
> | 600 | 0.92  | 0.08  | 1.54  | 2.28 |
> | 700 | 1.18  | 0.09  | 2.17  | 3.56 |
>
> The computation time of our proposed CoRT (Lasso) and CoRT (SCAD) increases as K or p increases. This is because we use the coordinate descent algorithm [3] to implement our methods. In either case, the increasing |S| or p leads to more parameters to be estimated since our methods need to find the contrast $\delta^{(k)} = \beta^{k} - \beta^{0}$. However, we note that for those high-dimensional data where $p \gg n$, the CoRT still achieve great scalability given its running time.
>
> We will include the results in the revised version.
>
>
>
> **3. The real-world dataset (COVID-19 mortality) is insightful but relatively small.**
>
> Thank you for the acknowledgement. Datasets in biological or social sciences typically come with small sizes (at each site/domain). While DNN-based TL methods typically work well in large datasets, they tend to perform poorly on small datasets (small size but large dimensionality). Our experiments actually illustrate its effectiveness to unique challenges in high dimensions.
>
>
>
> **4. Regarding the derived theoretical guarantees, how do these relate to or differ from discrepancy-based domain adaptation methods in terms of assumptions and guarantees?**
>
> This is a great question. It can be long to have a detailed discussion, so we try to make it concise but clear. At a high level, the theoretical study of domain adaptation (transfer learning) consists of two lines.
>
> The first is, as you mentioned, ``discrepancy-based’’ results; see [4,5] and etc. They obtain model-agnostic and divergence-centric bounds which characterize worst-case performance, making bounds tend to be general and overly pessimistic, particularly when data from the target domain is available, making them less instructive.
>
> The second one (where our work falls in) is based on more refined assumptions than the first line. For example, works in this line typically assume distribution shifts (our work is a combination of model and covariate shifts) and access to unlabeled/labeled data on the target domain. These settings bring more refined control over the discrepancy between domains, allowing researchers to design specific transfer learning algorithms. The statistical properties and guarantees are also more optimistic, as we can conduct inference on the parameters (Section 2.3) and also investigate the minimax optimality.
>
>
> **5. Are you willing to release the source code implementing your methods and experiments?**
>
>  Yes, we will release the corresponding codes after acceptance. Thank you very much.
>
> **Reference**
>
> [1] Li, S., Cai, T. T., and Li, H. (2022). Transfer learning for high-dimensional linear regression: Prediction, estimation and minimax optimality. Journal of the Royal Statistical Society Series B: Statistical Methodology.
>
> [2] Tian, Y. and Feng, Y. (2023). Transfer learning under high-dimensional generalized linear models. Journal of the American Statistical Association.
>
> [3] Breheny, P., & Huang, J. (2011). Coordinate descent algorithms for nonconvex penalized regression, with applications to biological feature selection. The annals of applied statistics.
>
> [4] Shai Ben-David, John Blitzer, Koby Crammer, and Fernando Pereira. (2006)  Analysis of representations for domain adaptation. NeurIPS.
>
> [5] Yishay Mansour, Mehryar Mohri, and Afshin Rostamizadeh. Domain adaptation: Learning bounds and algorithms.

---

> > ### Comment · Reviewer_CHM6 · 2025-08-06
> > **Response to Authors's rebuttal**
> >
> > Dear Authors,
> >
> > Thank you for your point by point response to my questions and observations. I am glad that you agreed on including an experimental analysis of the computational complexity of your method, and I also found your remaining answer sufficiently convincing. Thus, and also taking into account the other reviews and your responses to them, I will keep my recommendation for acceptance.

---

> > > ### Author Response · Authors · 2025-08-08
> > >
> > > Thank you for your support!

---

### Official Review · Reviewer_8wQE · 2025-07-03

**Clarity:** 3
**Significance:** 2
**Originality:** 4
**Rating:** 4
**Confidence:** 3

**Summary:**

This paper proposes a two-step framework for multi-source transfer learning. The proposed method combines Lasso and Co-Regularized Transfer (CoRT) to prevent overfitting and selects the most suitable source dataset ($\hat{S}$) among multiple sources, providing predictions optimized for the target data.

**Questions:**

1. An explanation is needed for what "E" and the "acute" symbol in line 94 mean.
2. The effectiveness of the proposed method could be further emphasized by presenting results for each source without the adaptive technique.
3. In what scenarios is the proposed method more effective (excluding the size of the target data)?
4. The authors mention that their method would be effective when applied to neural networks. What considerations should be made when applying it to DNNs?
5. Instead of dropping specific source domains, is there a way to maximize the knowledge that can be obtained from each source domain (e.g., sample-wise)?

**Ethical Concerns:**

["NO or VERY MINOR ethics concerns only"]

**Final Justification:**

Thank you for your thorough and considerate responses to the reviewer’s comments. I believe this manuscript holds sufficient scholarly merit to be recognized for its research value.

**Limitations:**

1. Some parameters still require tuning.
2. The observed improvement in target task performance is relatively modest. This may indicate that the framework struggles in cases where source-target alignment is weak.

**Quality:**

2

**Strengths And Weaknesses:**

$\textbf{Strengths}$

1. The authors' efforts to provide mathematical interpretations as evidence and theoretical support for the proposed method are noteworthy.
2. By splitting the target data to calculate errors, the method minimizes overfitting to specific sources in transfer learning.
3. Compared to conventional ML techniques, it offers automation advantages, such as reducing the need for parameter tuning.

$\textbf{Weaknesses}$

1. The explanation of the proposed method is somewhat difficult to understand. For example, in line 49 of the introduction, it should be clarified how adaptive identification of outlier sources is achieved and how this information is utilized during training.
2. The improvement in task performance by the proposed method is somewhat limited.
3. The analysis of experimental results is not described in depth.
4. Minor: The text and captions in the figures are too small. Careful review by the authors is needed.

---

> ### Author Rebuttal · Authors · 2025-07-31
>
> Thank you for your comments! Please see our responses in the following.
>
> **1. The explanation of the proposed method is somewhat difficult to understand. For example, in line 49 of the introduction, it should be clarified how adaptive identification of outlier sources is achieved and how this information is utilized during training.**
>
> Thanks for the comment. Due to the space limit, we can only briefly mention that it is achieved via the majority vote mechanism in line 51. This information is then utilized by excluded identified outlier sources in the Co-regularization process. We will add additional details once there is more space.
>
> **2. An explanation is needed for what "E" and the "acute" symbol in line 94 mean.**
>
> E stands for expectation, and $\psi’$ is the first-order derivative of the cumulant function used in GLMs, see [1].
>
> **3. The effectiveness of the proposed method could be further emphasized by presenting results for each source without the adaptive technique.**
>
> We believe the reviewer is referring to Figure 2. As we mentioned on line 298: The Model starts with ``pooled'', which means that we directly pool all sources without an adaptive procedure. Therefore, Pooled-CoRT and Pooled-Trans-GLM are the results without the adaptive techniques, i.e., all sources are directly used in the proposed co-regularization scheme. Note that the goal of transfer learning is to estimate the estimation error of the target data. Current results on Figure 2 show the averaged error of all targets, which should have similar patterns to the results of each target.
>
> **4. In what scenarios is the proposed method more effective (excluding the size of the target data)?**
>
> As Table 1 indicates, to outperform single-task learning (naive-Lasso), either a similar source (small $h$) or a larger transferable source sample size (larger $N$) can lead to smaller errors. Compared to the existing two-step TL, we allow less similar sources to achieve the same improved error bounds.
>
> **5. The authors mention that their method would be effective when applied to neural networks. What considerations should be made when applying it to DNNs?**
>
> In the discussion, we only mentioned the limitation of DNN when applied to a very small dataset. Small-sized datasets with extremely large-sized covariates are common in certain fields like social and biological sciences, where high-dimensional statistical models (and the TL techniques in high-dimensions) usually outperform DNN-based methods.
>
> However, our framework can generalize to other models naturally as the key idea is to simultaneously regularize (1) the distance between source and target parameters and (2) the target parameter itself; see Eq. (1).
>
> Therefore, for more generic cases, we can modify the objective to
> $$
> \sum\_{k \in \mathcal{S}} \left( \text{Loss}( f^{(0)} + \delta^{(k)}, \mathcal{D}^{(k)} )  + \lambda\_{k} \\|\delta^{(k)} \\|\_{\mathcal{H}\_{1}}  \right) + \text{Loss}( f^{(0)} , \mathcal{D}^{(0)} ) + \lambda\_{0} \\| f^{(0)} \\|\_{\mathcal{H}\_{2}}
> $$
> where $\mathcal{H_{1}}$, $\mathcal{H}_{2}$ represent generic model spaces and $f$ denote the parameters/models of interests. When extended to DNN, we just need to set these $f$ as the neural network. One consideration would be that these DNN needs to be parameterized similarly.
>
>
> **6. Instead of dropping specific source domains, is there a way to maximize the knowledge that can be obtained from each source domain (e.g., sample-wise)?**
>
> One way to maximize the knowledge from each source domain is to aggregate models from each source domain by different aggregation techniques, like [2] and [3]. Sample-wise approach could be possible, but it is more related to the topic in active learning.
>
> That being said, either aggregation or sample-wise knowledge transfer typically comes with extensive computational cost. For example, aggregation-based needs to calculate source models multiple times, and sample-wise typically requires one to score each sample within the source domain. On the other hand, source-wise selection achieves a better trade-off between performance and efficiency.
>
>
> **Reference**
>
> [1] Loh, P. L., & Wainwright, M. J. (2015). Regularized M-estimators with nonconvexity: Statistical and algorithmic theory for local optima. The Journal of Machine Learning Research.
>
> [2] Li, S., Cai, T. T., and Li, H. (2022). Transfer learning for high-dimensional linear regression: Prediction, estimation and minimax optimality. Journal of the Royal Statistical Society Series B: Statistical Methodology.
>
> [3] Lin, Haotian, and Matthew Reimherr. (2024) On Hypothesis Transfer Learning of Functional Linear Models. ICML.

---

> > ### Comment · Reviewer_8wQE · 2025-08-09
> >
> > I appreciate your thorough responses.
> > The authors have addressed all my comments with clarity.
> > Upon reviewing all responses, I can confirm that the overall clarity of the method has been significantly improved.
> > Therefore, I will raise my rating accordingly.

---

> > > ### Author Response · Authors · 2025-08-09
> > >
> > > Thank you very much for the support!

---

### Note · Authors · 2025-08-14

We sincerely express our gratitude to all the reviewers for their constructive comments throughout the rebuttal period. We appreciate all your comments highlighting the strengths of our work, including:

$\textbf{Methodology novelty}$: We presented a novel transfer learning framework, CoRT, which leverages the co-training process to better transfer the beneficial knowledge from source domains. This alleviates failure of fine-tuning due to the largely biased pre-trained models in the two-step paradigms.

$\textbf{Theoretical foundations}$: We provided theoretical analyses for the proposed CoRT framework, which show clear and provable advantages compared to the existing two-step paradigm in high-dimensional regimes. The technical tools we developed in this work can also be beneficial for other co-training schemes.

During rebuttal period, we include enhancements including
- Explanatory statements: Clarified the framework in high-dimensional GLMs as  (a) it allows head-to-head comparisons with existing seminal works and (b) it plays an important role in various fields of studies.
- Mathematical statements: Clarified the CoRT framework can be naturally extended from GLMs to more general models, and the underlying principles that make CoRT effective.
- Additional experiments: Evaluated the runtime complexity to confirm the scalability under large dimensionality and source number.

We appreciate the recognition of our contributions: The authors' efforts to provide mathematical interpretations as evidence and theoretical support for the proposed method are noteworthy (Reviewer 8wQE); The proposed CoRT framework is novel and avoids the bias. It presents solid theoretical foundations and the provided convergence bounds compare favorably with the existing methods (Reviewer CHM6); Strong statistical analysis is provided to show the clear advantage of the proposed CoRT (Reviewer 1KFr); The theoretical analysis provides guarantee to the convergence of the optimization process (Reviewer MAPn).

$\textbf{Updates of the manuscript}$

- We added experiment results to demonstrate the computation complexity and Reviewer CHM6 was glad with the results.
- We explained when our CoRT is more effective and why NNs may not work with small sample data. Reviewer 8wQE agreed to raise the score.
- We explained the design of our method in detail and Reviewer MAPn agreed to raise the score.

We believe these explanations and clarifications fully addressed reviewers’ comments.

---

### Decision · Program_Chairs · 2025-09-17

**Decision:**

Accept (poster)

**Comment:**

The paper proposes a joint one-step framework for multi-source transfer learning in high-dimensional GLMs. The method jointly estimates the target parameters and the source parameter deviations. Theory is provided for the case of GLMs and they appear to be solid. Experiments on both simulation and real-world COVID data were provided. One main criticism of the paper is that the problem addressed is going against the current trend in pre-training large models and fine-tuning on new domain, which limits the real-world applicability of the problem setup. However, within the considered scope, the paper's contribution is solid.